# Learning Causal Dynamics Models in Object-Oriented Environments

## Abstract

Causal Dynamics Models (CDMs) have demonstrated significant potential in addressing various challenges in reinforcement learning. Recent studies have incorporated causal discovery to capture the causal dependencies among environmental variables in the learning of CDMs. However, the learning of CDMs is still confined to small-scale environments due to computational complexity and sample efficiency constraints. This paper aims to extend CDMs to large-scale object-oriented environments, which consist of a multitude of objects classified into different categories. We introduce the Object-Oriented CDM (OOCDM) that shares causalities and parameters among objects belonging to the same class. Furthermore, we propose a learning method for OOCDM that enables it to adapt to a varying number of objects. Experimental results from large-scale tasks indicate that OOCDM outperforms existing CDMs in terms of causal discovery, prediction accuracy, generalization, and computational efficiency.

## 1 Introduction

Reinforcement learning (RL) (Sutton & Barto, 2018) and causal inference (Pearl, 2000) have separately made much progress over the past decades. Recently, the combination of both fields has led to a series of successes (Zeng et al., 2023). In these studies, the use of *causal dynamics models* (CDMs) proves a promising direction. CDMs capture the causal structures of environmental dynamics and have been applied to address a wide range of challenges in RL, including learning efficiency, explainability, generalization, state representation, subtask decomposition, and transfer learning (see the literature review in Section 2.1). For example, a major function of CDMs is to reduce spurious correlations (Ding et al., 2022; Wang et al., 2022), which are particularly prevalent in the non-i.i.d. data produced in sequential decision-making.

Early research of CDMs exploits given causal structures of environments (Boutilier et al., 2000; Guestrin et al., 2003a; Madumal et al., 2020b), which may not be available in many applications. Therefore, some recent studies have proposed to develop CDMs using causal discovery techniques to learn such causal structures, i.e. causal graphs (CGs), from the data of history interactions (Volodin, 2021; Wang et al., 2021; 2022; Zhu et al., 2022). These approaches have been successful in relatively small environments consisting of a few variables. Unfortunately, some RL tasks involve many objects (e.g., multiple agents and environment entities in multi-agent domains (Malysheva et al., 2019)), which together contribute to a large set of environment variables. The applicability of CDMs in such large-scale environments remains questionable — the excessive number of potential causal dependencies (i.e., edges in CGs) makes causal discovery extremely expensive, and more samples and effort are required to correctly discriminate causal dependencies.

Interestingly, humans seem to effortlessly extract correct causal dependencies from vast amounts of real-world information. One possible explanation for this is that we intuitively perceive tasks through an object-oriented (OO) perspective (Hadar & Leron, 2008) — we decompose the world into objects and categorize them into classes, allowing us to summarize and share rules for each class. For example, "exercise causes good health of each person" is a shared rule of the class "Human", and "each person" represents any instance of that class. This OO intuition has been widely adopted in modern programming languages, referred to as object-oriented programming (OOP), to organize and manipulate data in a more methodical and readable fashion (Stroustrup, 1988).

This work aims to extend CDMs to large-scale OO environments. Inspired by OOP, we investigate how an OO description of the environment can be exploited to facilitate causal discovery and dynamics learning. We propose the *Object-Oriented Causal Dynamics Model* (OOCDM), a novel type of CDM that allows the sharing of causalities and model parameters among objects based on sound theories of causality. To implement causal discovery and learning for OOCDM, we present a modified version of Causal Dynamics Learning (CDL) (Wang et al., 2022) that can accommodate varying numbers of objects. We apply OOCDM to several OO domains and demonstrate that it outperforms state-of-the-art CDMs in terms of causal graph accuracy, prediction accuracy, generalization ability, and computational efficiency, especially for large-scale tasks. To the best of our knowledge, OOCDM is the first dynamics model to combine causality with the object-oriented settings in RL.

## 2 RELATED WORKS

### 2.1 CAUSALITY AND REINFORCEMENT LEARNING

Causality (see basics in Appendix B) formulates dependencies among random variables and is used across various disciplines (Pearl, 2000; Pearl et al., 2016; Pearl & Mackenzie, 2019). One way to combine causality with RL is to formulate a known causal structure among *macro* elements (e.g., the state, action, and reward) of the Markov Decision Process (MDP), thereby deriving algorithms with improved robustness and efficiency (Buesing et al., 2018; Lu et al., 2018; Zhang et al., 2020; Liao et al., 2021). This paper follows another direction focusing on the *micro* causality that exists among specific components of the environment. Modular models prove capable of capturing such causality using independent sub-modules, leading to better generalization and learning performance (Ke et al., 2021; Mittal et al., 2020; 2022). A popular setting for the micro causality is *Factored MDP* (FMDP) (Boutilier et al., 2000), where the transition dynamics is modeled by a CDM. Knowledge to this CDM benefits RL in many ways, including 1) efficiently solving optimal policies (Guestrin et al., 2003a; Osband & Van Roy, 2014; Xu & Tewari, 2020), 2) sub-task decomposition (Jonsson & Barto, 2006; Peng et al., 2022), 3) improving explainability (Madumal et al., 2020a;b; Volodin, 2021; Yu et al., 2023), 4) improving generalization of policies (Nair et al., 2019) and dynamic models (Ding et al., 2022; Wang et al., 2022; Zhu et al., 2022), 5) learning task-irrelevant state representations (Wang et al., 2021; 2022), and 6) policy transfer to unseen domains (Huang et al., 2022).

### 2.2 OBJECT-ORIENTED REINFORCEMENT LEARNING

It is common in RL to describe environments using multiple objects. Researchers have largely explored object-centric representation (OCR), especially in visual domains, to facilitate policy learning (Zambaldi et al., 2018; Zadaianchuk et al., 2020; Zhou et al., 2022; Yoon et al., 2023) or dynamic modeling (Zhu et al., 2018; 2019; Kipf et al., 2020; Locatello et al., 2020). However, OCR typically uses homogeneous representations of objects and struggles to capture the diverse nature of objects. Goyal et al. (2020; 2022) overcome this problem by extracting a set of dynamics templates (called *schemata* or *rules*) that are matched with objects to predict next states. Prior to our work, Guestrin et al. (2003b) and Diuk et al. (2008) investigated OOP-style MDP representations using predefined classes of objects. Relational Causal Discovery Maier et al. (2010); Marazopoulou et al. (2015) operates categorized objects and reveals the shared causality within different inter-object relations, which carries a similar idea of causality sharing. However, our work focuses on the FMDP settings where relations are implicit and unknown, which may contribute to more general use.

## 3 PRELIMINARIES

A random variable (one scalar or a combination of multiple scalars) is denoted by a capital letter (e.g., $X_1$ and $X_2$). Parentheses may combine variables or subgroups into a *group* (an ordered set) denoted by a bold letter, e.g. $\mathbf{X} = (X_1, X_2)$ and $\mathbf{Z} = (\mathbf{X}, Y_1, Y_2)$. We use $p$ to denote a distribution.

### 3.1 CAUSAL DYNAMICS MODELS FOR FACTORED MARKOV DECISION PROCESS

We consider the FMDP setting where the state and action consist of multiple random variables, denoted as $\mathbf{S} = (S_1, \cdots, S_{n_s})$ and $\mathbf{A} = (A_1, \cdots, A_{n_a})$, respectively. $S_i'$ (or $\mathbf{S}'$) denotes the state vari-

able(s) in the next step. The transition probability $p(\mathbf{S}'|\mathbf{S}, \mathbf{A})$ is modeled by a CDM (see Definition 1), which is also referred to as a *Dynamics Bayesian Network* (DBN) (Dean & Kanazawa, 1989) adapted to the context of RL. For clarity, we illustrate a simple deterministic CDM in Appendix C.4.

**Definition 1.** A *causal dynamics model* is a tuple $\langle \mathcal{G}, p \rangle$. $\mathcal{G}$ is the *causal graph*, i.e. a directed acyclic graph (DAG) on $(\mathbf{S}, \mathbf{A}, \mathbf{S}')$, defining the parent set $Pa(\mathbf{S}'_j)$ for each $\mathbf{S}'_j$ in $\mathbf{S}'$. Then $p$ is a transition distribution on $(\mathbf{S}, \mathbf{A}, \mathbf{S}')$ such that $p(\mathbf{S}'|\mathbf{S}, \mathbf{A}) = \prod_{j=1}^{n_s} p(\mathbf{S}_j|Pa(\mathbf{S}'_j))$.

We assume that $\mathcal{G}$ is unknown and must be learned from the data. Some studies learn CGs using sparsity constraints, which encourage models to predict the next state variable using fewer inputs (Volodin, 2021; Wang et al., 2021). However, there exists no theoretical guarantee that sparsity can lead to sound causality. Another way to discover CGs is to use conditional independence tests (CITs) (Eberhardt, 2017), as suggested by several recent studies (Wang et al., 2022; Ding et al., 2022; Zhu et al., 2022; Yu et al., 2023). Theorem 1 presents a prevalent approach that leads to sound CGs (see proof in Appendix C.3).

**Theorem 1** (Causal discovery for CDMs). *Assuming that state variables transit independently, i.e. $p(\mathbf{S}'|\mathbf{S}, \mathbf{A}) = \prod_{j=1}^{n_s} p(\mathbf{S}'_j|\mathbf{S}, \mathbf{A})$, then the ground-truth causal graph $\mathcal{G}$ is bipartite. That is, all edges start in $(\mathbf{S}, \mathbf{A})$ and end in $\mathbf{S}'$; if $p$ is a faithful probability function consistent with the dynamics, then $\mathcal{G}$ is uniquely identified by*

$$\mathbf{X}_i \in Pa(\mathbf{S}'_j) \Leftrightarrow \neg(\mathbf{X}_i \perp\!\!\!\perp_p \mathbf{S}'_j \,|\, (\mathbf{S}, \mathbf{A}) \smallsetminus \{\mathbf{X}_i\}), \quad \text{for } \mathbf{X}_i \in (\mathbf{S}, \mathbf{A}), \, \mathbf{S}'_j \in \mathbf{S}', \tag{1}$$

Here, "$\smallsetminus$" means set-subtraction and "$\perp\!\!\!\perp_p$" denotes the conditional independence under $p$. The independence "$\perp\!\!\!\perp_p$" here can be determined by CITs, which utilize samples drawn from $p$ to evaluate whether the conditional independence holds. There are many tools for CITs, such as Fast CIT (Chalupka et al., 2018), Kernel-based CIT (Zhang et al., 2012), and Conditional Mutual Information (CMI) used in this work. Read Appendix B.4 for more information about CITs and CMI. Performing CITs according to Eq. 1 leads to sound CGs, yet is hardly scalable. On the one hand, the computation is extremely expensive. Letting $n := n_a + n_s$ denote the total number of environment variables, the time complexity of mainstream approaches reaches up to $O(n^3)$, since $O(n^2)$ edges must be tested, each costing $O(n)$. On the other hand, a larger $n$ impairs sampling efficiency, as CITs require more samples to recover the joint distribution of condition variables.

## 3.2 OBJECT-ORIENTED MARKOV DECISION PROCESS

Following Guestrin et al. (2003b), we formulate the task as an *Object-Oriented MDP* (OOMDP) containing a set $\mathcal{O} = \{O_1, \cdots, O_N\}$ of *objects*. Each object $O_i$ corresponds to a subset of variables (called its *attributes*), written as $\mathbf{O}_i = (O_i.\mathbf{S}, O_i.\mathbf{A})$, where $O_i.\mathbf{S} \subseteq \mathbf{S}$ and $O_i.\mathbf{A} \subseteq \mathbf{A}$ respectively are its state attributes and action attributes. The objects are divided into a set of classes $\mathcal{C} = \{C_1, \cdots, C_K\}$. We call $O_i$ an *instance* of $C_k$ if $O_i$ belongs to some class $C_k$, denoted as $O_i \in C_k$. $C_k$ specifies a set $\mathcal{F}[C_k]$ of *fields*, which determine the attributes of $O_i$ as well as other instances of $C_k$. Each field in $\mathcal{F}[C_k]$, typically written as $C_k.U$ (where $U$ can be replaced by any identifier), signifies an attribute $O_i.\mathbf{U} \in \mathbf{O}_i$ for each $O_i \in C_k$. Note that italic identifiers are used to represent fields (e.g., $C_k.U$), yet attributes use corresponding Roman letters (e.g., $O_i.\mathbf{U}$) to highlight that attributes are random variables. The dynamics of the OOMDP satisfy that *the state variables of objects from the same class transit according to the same (unknown) class-level transition function*:

$$p(O_i.\mathbf{S}'|\mathbf{S}, \mathbf{A}) = p_{C_k}(O_i.\mathbf{S}'|\mathbf{O}_i; \mathbf{O}_1, \cdots, \mathbf{O}_{i-1}, \mathbf{O}_{i+1}, \cdots, \mathbf{O}_N), \quad \text{for } \forall O_i \in C_k, \tag{2}$$

which we refer to as the *result symmetry*. Diuk et al. (2008) further formulates the dynamics by a set of logical rules, which is not necessarily required in our setting. All notations used in this paper are listed in Appendix A, and a more rigorous definition of OOMDP is given in Appendix D.1. This OOMDP representation is available in many simulation platforms (which are inherently built using OOP) and can be intuitively specified from human experience. Therefore, we consider the OOMDP representation as prior knowledge and leave its learning to future work. To illustrate our setting, we present Example 1 as the OOMDP for a StarCraft environment.

**Example 1.** In a StarCraft scenario shown in Figure 1, the set of objects is $\mathcal{O} = \{M_1, M_2, Z_1, Z_2, Z_3\}$ and the set of classes is $\mathcal{C} = \{C_M, C_Z\}$. $C_M$ is the class for marines $M_1$ and $M_2$. Similarly, $C_Z$ is the class for zerglings $Z_1$, $Z_2$, and $Z_3$. The fields for both $C = C_M, C_Z$ are given by $\mathcal{F}[C] = \{C.H, C.P, C.A\}$ — the **H**ealth, **P**osition, and **A**ction (e.g., move or attack). Therefore, for example, $M_1.\mathrm{H}$ is the health of marine $M_1$, and $\mathbf{M}_1 = (M_1.\mathrm{H}, M_1.\mathrm{P}, M_1.\mathrm{A})$.

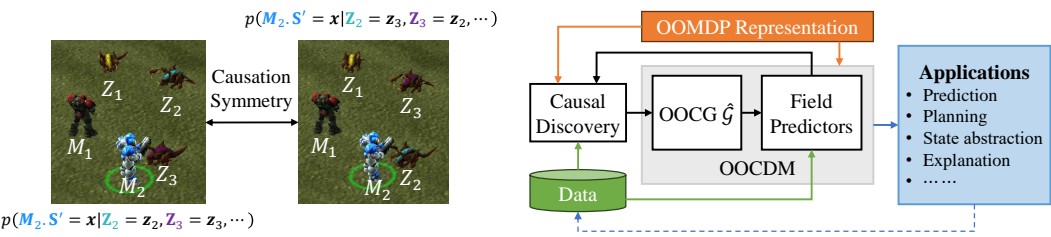

Figure 1: Visualizations and the causation symmetry in Example 1.

Figure 2: Workflow overview.

## 4  METHOD

The core of an OOCDM is the *Object-Oriented Causal Graph* (OOCG), which allows for class-level causality sharing based on the dynamic similarity between objects of the same class (see Section 4.1). Equation 2 has illustrated this similarity with respect to the result terms of the transition probabilities. Furthermore, we introduce an assumption 1 concerning the condition terms, called *causation symmetry*. It provides a natural notion that objects of the same class produce symmetrical effects on other objects. Figure 1 illustrates this assumption using the StarCraft scenario described above — swapping all attributes between two zerglings $Z_2$ and $Z_3$ makes no difference to the transition of other objects such as the marine $M_2$. We also assume that all state variables (attributes) transit independently in accordance with FMDPs (Guestrin et al., 2003a).

**Assumption 1** (Causation Symmetry). *Suppose $O_i \in C_k$. Then for any $a, b \neq i$, $O_a$ and $O_b$ are* **interchangeable** *to the transition of $O_i$, if they both belong to some class $C_l$:*

$$p(O_i.\mathbf{S}'|\mathbf{O}_a = \boldsymbol{a}, \mathbf{O}_b = \boldsymbol{b}, \cdots) = p(O_i.\mathbf{S}'|\mathbf{O}_a = \boldsymbol{b}, \mathbf{O}_b = \boldsymbol{a}, \cdots), \qquad O_a, O_b \in C_l. \qquad (3)$$

The workflow for using an OOCDM is illustrated in Figure 2. First, we use domain knowledge about the task to construct its OOMDP representation (Section 3.2). Subsequently, we initialize the OOCDM inclusive of field predictors (Section 4.2) and an OOCG estimation $\hat{G}$. This estimation is updated by performing causal discovery on the transition data and the predictors (Section 4.3), and these predictors are optimized using the current OOCG estimation and the stored data (Section 4.4). The learned OOCDM can then be applied to problems that require a CDM or causal graph (some basic applications are tested in Section 5). The soundness of the proposed approach relies on the dynamic symmetries (Eqs 2 and 3), which may sometimes be violated. However, it is usually feasible to ensure the symmetries by adding auxiliary attributes. Appendix I provides a simple solution for OOCDM to handle asymmetric environments, supported by theory and additional experiments.

### 4.1  OBJECT-ORIENTED CAUSAL GRAPH

According to Theorem 1, the ground-truth CG of an OOMDP follows a *bipartite causal graph* (BCG) structure, where no lateral edge is present in $\mathbf{S}'$. In order to simplify the process of causal discovery, we impose a restriction on the structure of $\mathcal{G}$ and introduce a special form of CGs that allows class-level causal sharing.

**Definition 2.** Let $\mathcal{F}_s[C_k] \subseteq \mathcal{F}[C_k]$ be the set of **state** fields of class $C_k$. An *Object-Oriented Causal Graph* is a BCG where **all** causal edges are given by a series of class-level causalities:

1. A *class-level local causality* for class $C_k$ from field $C_k.U \in \mathcal{F}[C_k]$ to state field $C_k.V \in \mathcal{F}_s[C_k]$, denoted as $C_k.U \to V'$, means that $O.U \in Pa(O.V')$ for every instance $O \in C_k$.
2. A *class-level global causality* from field $C_l.U \in \mathcal{F}[C_l]$ to state field $C_k.V \in \mathcal{F}_s[C_k]$, denoted as $C_l.U \to C_k.V'$, means that $O_j.U \in Pa(O_i.V')$ for every $O_i \in C_k$ and every $O_j \in C_l$ ($j \neq i$).

Definition 2 enables causality sharing by two types of class-level causalities, which are invariant with the number of instances of each class. Similar to relational causal discovery (Marazopoulou et al., 2015), this causality sharing greatly simplifies causal discovery and improves the readability of CGs. The *local* causality describes shared structures within individual objects of the same class,

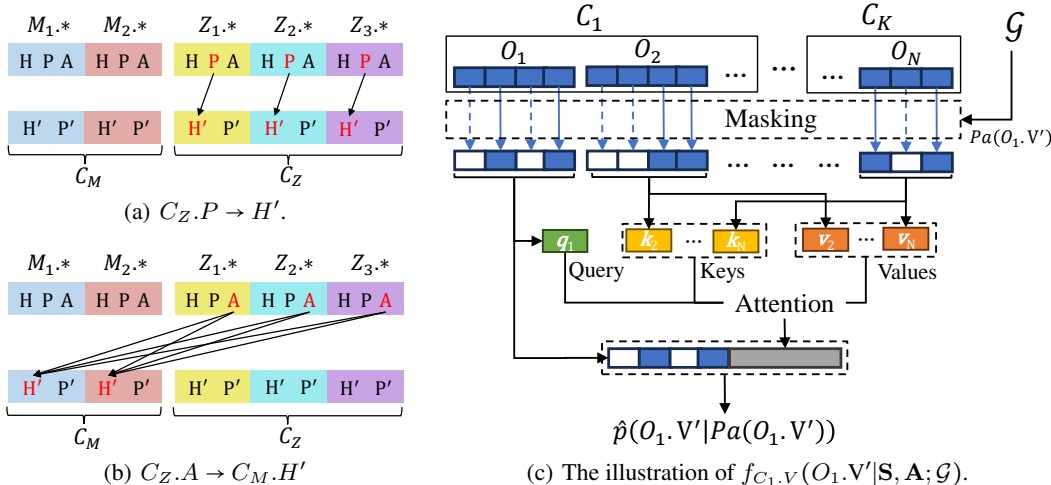

Figure 3: The class-level causalities in Example 1 and the implementation of a field predictor.

as illustrated in Figure 3(a). The *global* causality accounts for shared structures of object pairs, as illustrated in Figure 3(b). Note that the global causality $C_k.U \to C_k.V'$ (i.e., when $k = l$) is different from the local causality $C_k.U \to V'$ by definition. For clarity, the global and local causalities here are different from those considered by Pitis et al. (2020), where "local" means that $(\mathbf{S}, \mathbf{A})$ is confined in a small region in the entire space. Additionally, Theorem 2 shows the applicability of OOCGs, with proof in Appendix D.2.

**Theorem 2.** *The ground-truth CG of any OOMDP where Assumption 1 holds is exactly an OOCG.*

### 4.2 OBJECT-ORIENTED CAUSAL DYNAMICS MODEL

**Definition 3.** An *object-oriented causal dynamics model* is a CDM $\langle \mathcal{G}, \hat{p} \rangle$ (see Definition 1) such that 1) $\mathcal{G}$ is an OOCG, and 2) $\hat{p}$ satisfies Eqs. 2 and 3.

Based on OOCGs, we are able to define CDMs in an object-oriented manner (see Definition 3). In conventional CDMs, there exists an independent predictor for each next-state attribute (variable) in $\mathbf{S}'$. However, Equation 2 offers an opportunity to reduce the number of predictors by class-level sharing. That is, a shared *field predictor* $f_{C.V}$ is used for each state field $C.V \in \mathcal{F}[C]$ to predict the corresponding attribute $O.V'$ for every instance $O \in C$.

We now briefly describe how an OOCDM is implemented in our work. Inspired by Wang et al. (2022), we let an OOCG $\mathcal{G}$ be an argument of the predictor $f_{C.V}$, making it adaptable to various graph structures. Therefore, in our implementation, it follows that

$$\hat{p}(O.V' | Pa_{\mathcal{G}}(O.V')) = f_{C.V}(O.V' | \mathbf{S}, \mathbf{A}; \mathcal{G}) \qquad \text{for } O \in C, \tag{4}$$

where $Pa_{\mathcal{G}}(O.V')$ is the parent set of $O.V'$ in $\mathcal{G}$. We ensure that $f_{C.V}$ adheres to $\mathcal{G}$ by masking off the non-parental variables. In addition, we adopt key-value attention (Vaswani et al., 2017) to ensure causation symmetry (Eq. 3) and enable adaptation to varying numbers of objects. A simple illustration of our implementation of $f_{C.V}$ is given as Figure 3(c), and detail is in Appendix E.

### 4.3 OBJECT-ORIENTED CAUSAL DISCOVERY

Theorem 2 indicates that causal discovery in an OOMDP with Assumption 1 becomes looking for an OOCG. If the numbers of instances are fixed, checking each class-level causality in the OOCG only requires one CIT (see Appendix D.3), where most CIT tools are applicable.

Further, to perform CITs in environments with changeable instance numbers, we introduce an adaptation of CDL using the class-level conditional mutual information. Assume that we have a dataset $\mathcal{D} = \{(s_t, a_t, s_{t+1})\}_{t=1}^{T}$, where $s_t$, $a_t$ and $s_{t+1}$ are the observed values of $\mathbf{S}$, $\mathbf{A}$ and $\mathbf{S}'$ at step $t$, respectively. We use $O.v_{t+1}$ to denote the observed $O.V$ in $s_{t+1}$ for each state field $C_k.V$ and instance

$O \in C_k$. Some OOCGs are helpful to the estimation of CMI: **I)** $\mathcal{G}_1$ is the **full** bipartite CG containing all causalities, which is also an OOCG by definition; **II)** $\mathcal{G}_{C.U \not\to V'}$ contains all causalities except for $C.U \to V'$; and **III)** $\mathcal{G}_{C_k.U \not\to C.V'}$ contains all causalities except for $C_k.U \to C.V'$. Letting $C_k^t$ denotes the set of instances of class $C_k$ at step $t$, with the predictors introduced in Section 4.2, we respectively write the CMIs for class-level local and global causalities as

$$CMI_{\mathcal{D}}^{C_k.U \to V'} := \frac{1}{\sum_{t=1}^{T} |C_k^t|} \sum_{t=1}^{T} \sum_{O \in C_k^t} \log \frac{f_{C_k.V}(O.v_{t+1}|\boldsymbol{s}_t, \boldsymbol{a}_t; \mathcal{G}_1)}{f_{C_k.V}(O.v_{t+1}|\boldsymbol{s}_t, \boldsymbol{a}_t; \mathcal{G}_{C_k.U \not\to V'})}, \tag{5}$$

$$CMI_{\mathcal{D}}^{C_l.U \to C_k.V'} := \frac{1}{\sum_{t=1}^{T} |C_k^t|} \sum_{t=1}^{T} \sum_{O_j \in C_k^t} \log \frac{f_{C_k.V}(O.v_{t+1}|\boldsymbol{s}_t, \boldsymbol{a}_t; \mathcal{G}_1)}{f_{C_k.V}(O.v_{t+1}|\boldsymbol{s}_t, \boldsymbol{a}_t; \mathcal{G}_{C_l.U \not\to C_k.V'})}. \tag{6}$$

Then, each class-level causality (denoted as $\varsigma$) is confirmed if $CMI_{\mathcal{D}}^{\varsigma} > \varepsilon$, where $\varepsilon$ is the threshold parameter. In other words, $CMI_{\mathcal{D}}^{\varsigma}$ compare the predictions made with and without the concerned parents within $\varsigma$, and we confirm the causality if the difference is significant. Theoretically, if we have an infinite number of samples and an oracle estimation of $p$, then $\varepsilon$ can be set to 0. In practice, we set $\varepsilon > 0$. In this way, no extra models are needed for causal discovery. Finally, the whole OOCG is obtained by checking CMIs for all possible causalities (see Appendix E.2 for the pseudo-code).

Our approach greatly reduces the computational complexities of causal discovery, from a magnitude (asymptotic boundary) of $n^3$ to a magnitude of $Nmn$, where $m$ denotes the overall number of fields and $n$ denotes the overall number of variables in $(\mathbf{S}, \mathbf{A})$. See proofs and more conclusions about computational complexities in Appendix F.

### 4.4 MODEL LEARNING

Dynamics models are usually optimized through Maximum Likelihood Estimation. To better adapt to the varying numbers of instances, we define the *average instance log-likelihood* (AILL) function on a transition dataset $\mathcal{D}$ of $T$ steps for any CDM $\langle \mathcal{G}, \hat{p} \rangle$ as

$$\mathcal{L}_{\mathcal{G}}(\mathcal{D}) = \sum_{k=1}^{K} \frac{1}{\sum_{t=1}^{T} |C_k^t|} \sum_{t=1}^{T} \sum_{C_k.V \in \mathcal{F}_s[C_k]} \sum_{O \in C_k^t} \log \hat{p}(O.V'|Pa_{\mathcal{G}}(O.V'))_t, \tag{7}$$

where $\hat{p}(\cdot)_t$ is the estimated probability when variables take the values observed at step $t$ in $\mathcal{D}$.

The learning target of an OOCDM mimics that of CDL. First, we optimize the AILL function under a random OOCG denoted as $\mathcal{G}_{\lambda}$ (re-sampled when each time used) where the probability of each class-level causality item is $\lambda$. This will make our model capable of handling incomplete information and adaptable to different OOCGs including those like $\mathcal{G}_{C.U \not\to V'}$ or $\mathcal{G}_{C_k.U \not\to C.V'}$. Furthermore, we also hope to strengthen our model in two particular OOCGs: 1) the estimation of ground-truth $\hat{G}$ obtained by causal discovery, where CMIs are estimated by the current model, and 2) the full OOCG $\mathcal{G}_1$ to better estimate CMIs in Eqs. 5 and 5. Therefore, two additional items, $\mathcal{L}_{\mathcal{G}_1}(\mathcal{D})$ and $\mathcal{L}_{\hat{\mathcal{G}}}(\mathcal{D})$, respectively weighted by $\alpha$ and $\beta$, are considered in the overall target function:

$$J(\mathcal{D}) = \mathcal{L}_{\mathcal{G}_{\lambda}}(\mathcal{D}) + \alpha \mathcal{L}_{\mathcal{G}_1}(\mathcal{D}) + \beta \mathcal{L}_{\hat{\mathcal{G}}}(\mathcal{D}), \tag{8}$$

which is optimized by gradient ascent. Pseudo-code of the learning algorithm is in Appendix E.3. During the test phase, all predictions of our OOCDM are made using the discovered OOCG $\hat{\mathcal{G}}$.

## 5 EXPERIMENTS

OOCDM was compared with several state-of-the-art CDMs. **CDL** uses pooling-based predictors and also adopts CMIs for causal discovery. **CDL-A** is the attention-based variant of CDL, used to make a fair comparison with our model. **GRADER** (Ding et al., 2022) employs Fast CIT for causal discovery and Gated Recurrent Units as predictors. **TICSA** (Wang et al., 2021) utilizes score-based causal discovery. Meanwhile, OOCDM was compared to non-causal baselines, including a widely used multi-layer perceptron (**MLP**) in model-based RL (MBRL) and an object-aware Graph Neural Network (**GNN**) that uses the architecture of Kipf et al. (2020) to learn inter-object relationships.

Table 1: The accuracy (in percentage) of discovered causal graphs. $n$ indicates the number of environmental variables.

| Env | $n$ | GRADER | CDL | CDL-A | TICSA | **OOCDM** |
|---|---|---|---|---|---|---|
| $Block_2$ | 12 | $94.8_{\pm 1.3}$ | $99.4_{\pm 0.3}$ | $99.2_{\pm 1.3}$ | $97.0_{\pm 0.4}$ | $\mathbf{99.7}_{\pm 0.6}$ |
| $Block_5$ | 24 | $94.0_{\pm 1.5}$ | $97.5_{\pm 1.5}$ | $99.3_{\pm 0.6}$ | $96.3_{\pm 0.6}$ | $\mathbf{100.0}_{\pm 0.0}$ |
| $Block_{10}$ | 44 | $92.3_{\pm 0.9}$ | $97.6_{\pm 0.3}$ | $99.5_{\pm 0.3}$ | $97.7_{\pm 0.5}$ | $\mathbf{100.0}_{\pm 0.0}$ |
| Mouse | 28 | $90.5_{\pm 0.8}$ | $90.4_{\pm 3.2}$ | $94.7_{\pm 0.2}$ | $94.1_{\pm 0.2}$ | $\mathbf{100.0}_{\pm 0.0}$ |

Additionally, we assessed the performance of the dense version of our OOCDM, namely **OOFULL**, which employs the full OOCG $\mathcal{G}_1$ and is trained by optimizing $\mathcal{L}_{\mathcal{G}_1}$.

As mentioned in Section 2.1, CDMs are used for various purposes, and this work does not aim to specify the use of OOCDMs. Therefore, we evaluate the performance of causal discovery and the predicting accuracy, as most applications can benefit from such criteria. As a common application in MBRL, we also evaluate the performance of planning using dynamics models. Our experiments aim to 1) demonstrate that the OO framework greatly improves the effectiveness of CDMs in large-scale environments, and 2) investigate in what occasions causality brings significant advantages. Results are presented by the means and standard variances of 5 random seeds. Experimental details are presented in Appendix H.

## 5.1 ENVIRONMENTS

We conducted experiments in 4 environments. The **Block** environment consists of several instances of class $Block$ and one instance of class $Total$. The attributes of each $Block$ object transit via a linear transform; and the attributes of the $Total$ object transit based on the maximums of attributes of the $Block$ objects. The **Mouse** environment is an $8 \times 8$ grid world containing an instance of class $Mouse$, and several instances of class $Food$, $Monster$, and $Trap$. The mouse can be killed by hunger or monsters, and its goal is to survive as long as possible. The Collect-Mineral-Shards (**CMS**) and Defeat-Zerglings-Baineling (**DZB**) environments are StarCraftII mini-games (Vinyals et al., 2017). In CMS, the player controls two marines to collect 20 mineral shards scattered on the map, and in DZB the player controls a group of marines to kill hostile zerglings and banelings. Read Appendix G for detailed descriptions of these environments.

The Block and Mouse environments are ideal OOMDPs as they guarantee Eqs. 2 and 3. In addition, we intentionally insert spurious correlations in them to verify the effectiveness of causal discovery. In CMS and DZB environments, we intuitively formulate the objects and classes based on the units and their types in StarCraftII. They account for more practical cases where tasks are not perfect OOMDPs, as the StarCraftII engine may not guarantee Eqs. 2 and 3.

## 5.2 PERFORMANCE OF CAUSAL DISCOVERY

We measured the performance of causal discovery using offline data in Block and Mouse environments. Since non-OO baselines only accept a fixed number of variables, the number of instances of each class is fixed in these environments. Especially, we use "$Block_k$" to denote the Block environment where the number of $Block$ instances is fixed to $k$. We exclude CMS and DZB here as their ground-truth CGs are unknown (see learned OOCGs in Appendix H.6).

The accuracy of discovered CGs (measured by Structural Hamming Distance within the edges from $\mathbf{U}$ to $\mathbf{S}'$) is presented in Table 1. OOCDM outperforms other CDMs in all 4 tested environments. Meanwhile, it correctly recovers ground-truth CGs in 3 out of 4 environments. These results demonstrate the great sample efficiency of OO causal discovery – which is even improved by the larger number of instances. Furthermore, Table 2 shows the computation time used by causal discovery. We note that such results may be influenced by implementation detail and hardware conditions, yet the OOCDM excels baselines with a significant gap beyond these extraneous influences. In addition, Appendix H.5 shows that OOCDM achieves better performance with a relatively smaller size (i.e. fewer model parameters).

Table 2: The time used in causal discovery, presented in the form "(seconds)/(number of samples used)". $m$ denotes the number of all fields. TICSA is excluded from comparison as it does not involve an explicit causal discovery phase.

| | $n$ | $m$ | GRADER | CDL | CDL-A | **OOCDM** |
|---|---|---|---|---|---|---|
| Block-2 | 12 | 8 | ( 114.0±1.5 )/10k | ( **1.4**±0.1 )/10k | ( 2.1±0.4 )/10k | ( 2.1±0.2 )/10k |
| Block-5 | 24 | 8 | ( 927.0±69.3 )/10k | ( 5.2±0.6 )/10k | ( 8.5±2.8 )/10k | ( **2.1**±0.2 )/10k |
| Block-10 | 44 | 8 | ( 7.0e3±217.7 )/10k | ( 15.6±0.7 )/10k | ( 22.8±5.3 )/10k | ( **2.2**±0.2 )/10k |
| Mouse | 29 | 10 | ( 1.7e4±138.4 )/50k | ( 57.8±2.4 )/50k | ( 45.3±2.6 )/50k | ( **17.7**±4.0 )/50k |
| CMS | 44 | 4 | ( 5.5e4±397.1 )/100k | ( 209.8±18.9 )/100k | ( 252.5±27.3 )/100k | ( **7.4**±0.5 )/100k |
| DZB | 66 | 10 | ( 2.7e4±387.1 )/20k | ( 715.6±10.9 )/200k | ( 1.1e3±274.7 )/200k | ( **66.1**±0.6 )/200k |

Table 3: The average instance log-likelihoods of the dynamics models on various datasets. We do not show the standard variances for obviously over-fitting results (less than −100.0, highlighted in brown), as their variances are all extremely large.

| Env | data | GRADER | CDL | CDL-A | TICSA | GNN | MLP | OOFULL | **OOCDM** |
|---|---|---|---|---|---|---|---|---|---|
| | train | 21.1±0.3 | 20.9±1.5 | 19.3±1.9 | 17.4±2.2 | 18.8±0.6 | 16.5±1.2 | 21.5±0.9 | **22.4**±0.7 |
| Block$_2$ | i.d. | 17.1±2.5 | 20.2±1.8 | 10.4±16.8 | 16.4±1.9 | 17.9±0.7 | 10.1±4.4 | −568.2 | **22.2**±0.7 |
| | o.o.d. | −65.4 | 11.5±6.7 | −6.0e5 | −60.1±2.8 | −5.0±23.4 | −7.2e4 | −4.6e4 | **21.3**±1.9 |
| | train | 19.1±3.4 | 16.5±2.1 | 18.9±0.7 | 12.0±0.7 | 14.9±14.4 | 12.6±0.5 | **20.4**±1.7 | 19.6±1.7 |
| Block$_5$ | i.d. | 6.7±4.3 | −45.3±113.2 | −1.4e7 | 10.8±0.7 | 14.4±0.4 | −2.2±6.3 | **19.8**±1.7 | 19.5±1.7 |
| | o.o.d. | −95.6±41.7 | −5.3e6 | −1.1e9 | −5.5e3 | −13.4±3.4 | −1.5e7 | −4.0e7 | **13.5**±4.3 |
| | train | 19.3±0.6 | 12.9±0.8 | 16.0±0.6 | 11.1±1.3 | 13.3±0.15 | 8.9±0.6 | 20.3±0.6 | **21.2**±0.3 |
| Block$_{10}$ | i.d. | −26.7±8.4 | 6.9±6.4 | −9.2±42.5 | −10.4±39.8 | 12.9±0.2 | −75.3±20.0 | 20.2±0.6 | **21.1**±0.3 |
| | o.o.d. | −119.1 | −4.2e6 | −1.9e8 | −139.4 | −17.3±17.3 | −780.9 | −5.4e3 | **15.6**±5.4 |
| | train | 24.2±0.6 | 13.9±1.8 | 22.3±1.4 | 13.6±3.5 | 25.6±1.8 | 5.7±0.4 | 30.0±1.4 | **32.2**±1.1 |
| Mouse | i.d. | −3.2e3 | −2.0e5 | −3.6e4 | −1.5e4 | −2.7e4 | −1.6e7 | −65.0±153.3 | **26.8**±6.7 |
| | o.o.d. | −7.1e4 | −1.1e10 | −2.0e10 | −2.5e7 | −6.3e10 | −8.0e10 | −1.5e9 | **11.2**±17.2 |
| CMS | train | −1.2±0.1 | 3.6±0.8 | 4.1±1.5 | 2.8±1.6 | 6.4±6.2 | −2.0±1.5 | 8.5±1.1 | **9.0**±0.5 |
| | i.d. | −1.3±0.1 | −1.0e6 | 4.1±1.5 | −16.3±7.4 | 6.3±0.1 | −6.4e9 | 8.5±1.1 | **8.9**±0.5 |
| DZB | train | 11.0±1.0 | 4.2±2.5 | 12.1±0.1 | 13.2±1.2 | 18.0±10.0 | −0.9±0.8 | **29.0**±0.6 | 27.2±2.5 |
| | i.d. | −14.9±21.8 | −3.3±6.6 | 5.3±5.3 | −2.4e5 | 13.0±12.8 | −1.6e12 | 22.6±5.6 | **24.4**±5.9 |

## 5.3 PREDICTING ACCURACY

We use the AILL functions (see Eq. 7) to measure the predicting accuracy of dynamics models. The models are learned using offline **train**ing data. Then, the AILL functions of these models are evaluated on the **i.d.** (in-distribution) test data sampled from the same distribution as the training data. Especially, in Block and Mouse environments, we can modify the distribution of the starting state of each episode (see Appendix H.3) and obtain the **o.o.d.** (out-of-distribution) test data, which contains samples that are unlikely to appear during training. The i.d. and o.o.d. test data measure two levels of generalization, respectively considering situations that are alike and unalike to those in training. We do not collect the o.o.d. data for CMS and DZB, as modifying the initialization process is difficult with limited access to the StarCraftII engine in PySC2 platform (Vinyals et al., 2017).

The results are shown in Table 3. In small-scale environments like Block-2, causal models show better generalization ability than dense models on both i.d. and o.o.d. test data. However, in larger-scale environments, the performance of non-OO models declines sharply, and OO models (OOFULL and OOCDM) obtain the highest performance on the i.d. data. In addition, our OOCDM exhibits the best generalization ability on the o.o.d. data; in contrast, the performance of OOFULL is extremely low on such data. These results demonstrate that OO models are more effective in large-scale environments, and that causality greatly improves the generalization of OO models.

## 5.4 COMBINING MODELS WITH PLANNING

In this experiment, we trained dynamics models using offline data (collected through random actions). Given a reward function, we used these models to guide decision-making using Model Predictive Control (Camacho & Bordons, 1999) combined with Cross-Entropy Method (Botev et al.,

Table 4: The average return of episodes when models are used for planning. In the Mouse environment, "o.o.d." indicates the initial states are sampled from a new distribution.

| Env | GRADER | CDL | CDL-A | TICSA | GNN | MLP | OOFULL | **OOCDM** |
|---|---|---|---|---|---|---|---|---|
| Mouse | $-1.2_{\pm1.9}$ | $3.9_{\pm3.0}$ | $-5.0_{\pm1.3}$ | $-0.8_{\pm0.7}$ | $6.6_{\pm3.2}$ | $0.6_{\pm2.0}$ | $77.9_{\pm18.1}$ | $\mathbf{80.1}_{\pm16.9}$ |
| o.o.d. | $-0.4_{\pm1.7}$ | $1.8_{\pm2.5}$ | $-0.9_{\pm1.1}$ | $-1.2_{\pm0.6}$ | $0.6_{\pm0.2}$ | $-1.3_{\pm0.7}$ | $62.2_{\pm8.7}$ | $\mathbf{75.1}_{\pm17.5}$ |
| CMS | $-9.5_{\pm1.1}$ | $-9.8_{\pm1.1}$ | $-8.8_{\pm0.4}$ | $-9.3_{\pm0.9}$ | $-9.8_{\pm0.7}$ | $-8.8_{\pm0.5}$ | $-4.1_{\pm3.3}$ | $\mathbf{3.4}_{\pm6.3}$ |
| DZB | $202.9_{\pm12.3}$ | $217.3_{\pm12.4}$ | $171.7_{\pm18.2}$ | $188.9_{\pm8.5}$ | $233.8_{\pm19.8}$ | $205.4_{\pm6.7}$ | $\mathbf{269.8}_{\pm21.5}$ | $266.2_{\pm11.4}$ |

Table 5: Results on various tasks in the Mouse environment. "seen" and "unseen" respectively indicate the performances measured in seen and unseen tasks.

| Model | average instance log-likelihood | | | episodic return | |
|---|---|---|---|---|---|
| | train | seen | unseen | seen | unseen |
| OOCDM | $26.9_{\pm3.5}$ | $\mathbf{25.4}_{\pm2.8}$ | $\mathbf{24.8}_{\pm2.8}$ | $\mathbf{94.8}_{\pm29.7}$ | $\mathbf{88.8}_{\pm34.8}$ |
| OOFULL | $\mathbf{30.7}_{\pm1.9}$ | $22.5_{\pm3.2}$ | $7.9_{\pm29.8}$ | $77.0_{\pm24.6}$ | $70.8_{\pm22.4}$ |

2013) (see Appendix E.4), which is widely used in MBRL. The Block environment is not included here as it does not involve rewards. In the Mouse environment, the o.o.d. initialization mentioned in Section 5.3 is also considered. The average returns of episodes are shown in Table 4, showing that OOFULL and OOCDM are significantly better than non-OO approaches.

Between the OO models, OOCDM obtains higher returns than OOFULL in 3 of 4 environments, which demonstrates that OOCDM better generalizes to the unseen state-action pairs produced by planning. Taking CMS for example, the agent collects only a few mineral shards in the training data. When the agent plans, it encounters unseen states where most mineral shards have been collected. However, we note that OOFULL performs slightly better than OOCDM in DZB. One reason for this is that DZB possesses a joint action space of 9 marines, which is too large to conduct effective planning. Therefore, planning does not lead to states that are significantly different from those in training, prohibiting the advantage of generalization from converting to the advantage of returns. Additionally, the true CG of DZB is possibly less sparse than those in other environments, making OOFULL contain less spurious edges. Therefore, CDMs would be more helpful, if the true CG is sparse, and there exists a large divergence between the data distributions in training and testing.

## 5.5 HANDLING VARYING NUMBERS OF INSTANCES

In the Mouse environment, we tested whether OOCDM and OOFULL are adaptable to various tasks with different numbers of $Food$, $Moster$, and $Trap$ instances. We randomly divide tasks into the *seen* and *unseen* tasks (see Appendix H.4). Dynamics models are first trained in *seen* tasks and then transferred to the *unseen* without further training. We measured the log-likelihoods on the training data, the i.d. test data on seen tasks, and the test data on unseen tasks. The average episodic returns of planning were also evaluated, separately on seen and unseen tasks. As shown in Table 5, our results demonstrate that 1) OO models can be learned using data from different tasks, 2) OO models perform a zero-shot transfer to unseen tasks with a mild reduction of performance, and 3) the overall performance is improved when combing the model with causality.

## 6 CONCLUSION

This paper proposes OOCDMs that capture the causal relationships within OOMDPs. Our main innovations are the OOCGs that share class-level causalities and the use of attention-based field predictors. Furthermore, we present a CMI-based method that discovers OOCGs in environments with changing numbers of objects. Theoretical and empirical data indicate that OOCDM greatly enhances the computational efficiency and accuracy of causal discovery in large-scale environments, surpassing state-of-the-art CDMs. Moreover, OOCDM well generalizes to unseen states and tasks, yielding commendable planning outcomes. In conclusion, this study provides OOCDM as a promising solution to learn and apply CDMs in large object-oriented environments.

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

Table 6: The meanings of acronyms that appear in the paper.

| Acronym | Explanation |
|---------|-------------|
| AILL | Average instance log-likelihood. |
| BCG | Bipartite causal graph. |
| BN | Bayesian Network. |
| CDM | Causal dynamics model. |
| CDL | A baseline proposed by Wang et al. (2022). |
| CG | Causal graph. |
| CIT | Conditional independence test. |
| CLCE | Class-level causality expression. |
| CMS | Collect-Mineral-Shards, a StarCraftII minigame. |
| CMI | Conditional mutual information. |
| DAG | Directed acyclic graph. |
| DBN | Dynamics Bayesian Network. |
| DZB | Defeat-Zerglings-Banelings, a StarCraftII minigame. |
| GNN | A baseline based on a graph neural network (Kipf et al., 2020). |
| i.d. | In-distribution. |
| MBRL | Model-based reinforcement learning. |
| MDP | Markov decision process. |
| MLP | Multi-layer perceptron. |
| OO | Object-oriented. |
| OOCDM | Object-oriented causal dynamics model |
| OOCG | Object-oriented causal graph. |
| o.o.d. | Out-of-distribution. |
| OOFULL | Object-oriented full model (a variant of OOCDM that uses full OOCGs). |
| OOMDP | Object-oriented Markov decision process. |
| RL | Reinforcement learning. |
| TICSA | A baseline proposed by Wang et al. (2021). |

## A  ACRONYMS AND SYMBOLS

The acronyms that appear in our paper are explained in Table 6. The meanings of symbols used in the paper or will be used in the appendices are described in Table 7 unless otherwise specified.

Table 7: Symbols used in the paper and appendices

| Symbol(s) | Explanation |
|-----------|-------------|
| $S_i$ | The $i$-th state variable in a FMDP. |
| $S_i'$ | The $i$-th next-state variable in a FMDP. |
| $\mathbf{S}$ | The group of state variables in a FMDP. |
| $\mathbf{S}'$ | The group of next-state variables in a FMDP. |
| $A_i$ | The $i$-th action variable in a FMDP. |
| $\mathbf{A}$ | The group of action variables in a FMDP. |
| $\mathbf{\Delta}$ | The group of all variables in a transition, i.e. $(\mathbf{S}, \mathbf{A}, \mathbf{S}')$. |
| $n_s$ | The number of state variables in a FMDP. |
| $n_a$ | The number of action variables in a FMDP. |
| $p$ | The probability distribution of random variables. |
| $\hat{p}$ | The estimated distribution for $p$ in a dynamics model. |
| $\mathcal{G}$ | The DAG of a causal model (e.g., a Bayesian network, CDM, or OOCDM). |
| $X \rightarrow Y$ | Variable X is a parent of variable Y in some given DAG. |
| $Pa_{\mathcal{G}}(X)$ | The parent set of variable X in DAG $\mathcal{G}$. |
| $Pa(X)$ | The parent set of variable X in the ground-truth causal graph. |
| $C$ (or $C_i$) | A (or the $i$-th) class in an OOMDP. |
| $\mathcal{C}$ | The set of classes in an OOMDP. |
| $\mathcal{F}[C]$ | The set of fields of class $C$, i.e. $\mathcal{F}_s[C] \cup \mathcal{F}_a[C]$. |

| Symbol(s) | Explanation |
|---|---|
| $\mathcal{F}_s[C]$ | The set of state fields of class $C$. |
| $\mathcal{F}_a[C]$ | The set of action fields of class $C$. |
| $\mathcal{F}$ | The set of all fields in an OOMDP, i.e. $\bigcup_{C \in \mathcal{C}} \mathcal{F}[C]$. |
| $\mathcal{F}_s$ | The set of all state fields in an OOMDP, i.e. $\bigcup_{C \in \mathcal{C}} \mathcal{F}_s[C]$. |
| $C.U$ | Some filed of $C$ in $\mathcal{F}[C]$. |
| $C.V$ | Some state field of $C$ in $\mathcal{F}_s[C]$. |
| $Dom_{C.U}$ | The domain of some field $C.U$. |
| $O, O_i$ | An object in an OOMDP. |
| $N$ | The number of objects in an OOMDP. |
| $K$ | The number of classes in an OOMDP. |
| $O \in C$ | Object $O$ is an instance of class $C$. |
| $O.U$ | An attribute of $O$ (derived from the field $C.U \in \mathcal{F}[C]$ where $O \in C$). |
| $O.V$ | A state attribute of $O$ (derived from the field $C.S \in \mathcal{F}_s[C]$ where $O \in C$). |
| $O.\mathbf{S}$ | The group of all state attributes of $O$. |
| $O.\mathbf{A}$ | The group of all action attributes of $O$. |
| $\mathbf{O}$ | All attributes of $O$, i.e. $(O.\mathbf{S}, O.\mathbf{A})$. |
| $O.V'$ | The variable of state attribute $O.V$ in the next-step. |
| $O.\mathbf{S}'$ | The group of state variables $O.\mathbf{S}$ in the next-step. |
| $O_a \sim O_b$ | $O_a$ and $O_b$ are instances of the same class. |
| $C.U \to V'$ | A local causality expression from $C.U$ to $C.V$. |
| $C_l.U \to C_k.V'$ | A global causality expression from $C_l.U$ to $C_k.V$. |
| $\mathcal{D}$ | A dataset of transition samples. |
| $C_k^t$ | The set of instances of class $C_k$ at step $t$. |
| $\hat{p}(\cdot)_t$ | The estimation of $p$ when variables take the observed values at step $t$. |
| $CMI_{\mathcal{D}}^{\varsigma}$ | The CMI for class-level causality $\varsigma$ on data $\mathcal{D}$. |
| $f_{C.V}$ | The predictor for the state field $C.V$ in the OOCDM. |
| $\mathcal{L}_{\mathcal{G}}(\mathcal{D})$ | The AILL function of data $\mathcal{D}$ under the CG $\mathcal{G}$. |
| $J(\mathcal{D})$ | The overall target function for model learning. |

Table 7 ends

# B  BASICS OF CAUSALITY

## B.1  CAUSAL MODELS

In this section, we present some of the basic concepts and theorems of causality, which form the foundation of our theory. We first introduce Markov Compatibility (Pearl, 2000), which defines whether a graph can correctly reflect the relationships among variables given a probability function.

**Definition* 1** (Markov Compatibility). Assume $\mathcal{G}$ is an acyclic directional graph (DAG) on a group of random variables $\mathbf{X} = (X_1, ..., X_n)$. Given any probability function $p$ of these variables, if the *rule of production decomposition* holds:

$$p(X_1, ..., X_n) = \prod_{j=1}^{n} p(X_j | Pa_{\mathcal{G}}(X_j)),\qquad(9)$$

then we say that $p$ is *compatible* with $\mathcal{G}$, or that $\mathcal{G}$ *represents* $p$.

Causality (the DAG) is a universal concept. The following theorem shows, that no matter what the probability function is, the dependencies between variables can always be represented by some DAG. This leads to a general form of a causal model called the Bayesian Network (BN).

**Theorem* 1** (Existence of causal graphs). *For any probability function $p$ of variables $\mathbf{X} = (X_1, \cdots, X_n)$, there always exists a DAG $\mathcal{G}$ that $p$ is compatible with.*

*Proof.* Using the chain rule of probability functions, we have

$$p(X_1, ..., X_n) = p(X_1)p(X_2|X_1)p(X_3|X_1, X_2)\cdots p(X_n|X_1, ..., X_{n-1}).\qquad(10)$$

Letting $Pa(X_j) \subseteq \{X_1, ..., X_{j-1}\}$ denote the minimal subset such that $p(X_j|X_1, ..., X_{j-1}) = p(X_j|Pa(X_j))$ (the Markovian parents (Pearl, 2000)) for $j = 1, ..., n$, we obtain Eq. 9. $\qquad\square$

**Definition\* 2** (Bayesian Network). A *Bayesian Netowrk* is a tuple $\langle \mathcal{G}, p \rangle$, where $\mathcal{G}$ is a directed acyclic graph (DAG) on a set of random variables $\mathbf{X} = (X_1, ..., X_n)$, and $p$ is a probability function of $\mathbf{X}$ such that $p$ is compatible with $\mathcal{G}$.

Especially, according to Laplacian's conception, most stochastic phenomenons in nature are due to deterministic functions combined with unobserved disturbances. This conception leads to a special type of BN called the Structural Causal Model (SCM), which is the most popular model in causal inference.

**Definition\* 3** (Structural Causal Model). A *Structural Causal Model* (SCM) is a tuple $\langle \mathcal{G}, p, \mathbf{U}, \mathcal{F} \rangle$, where $\langle \mathcal{G}, p \rangle$ forms a Bayesian Network on variables $\mathbf{X} = (X_1, ..., X_n)$. $\mathbf{U} = (U_1, ..., U_n)$ is a set of disturbance variables that are independent of each other. $\mathcal{F} = \{f_1, f_2, \cdots, f_n\}$ is a set of structural equations, such that

$$X_i = f_i(Pa(X_i); U_i). \tag{11}$$

### B.2 D-SEPARATION

The concept of *d-seperation* plays an important role in causal inference. Given a CG $\mathcal{G}$, the criterion of d-separation provides an effective way to determine on what condition two groups of variables are independent.

**Definition\* 4** (d-separation). Assume $\mathcal{G}$ is a DAG on a set of variables $\mathbf{V}$. Assume $\mathbf{X}$, $\mathbf{Y}$, and $\mathbf{Z}$ are three disjoint groups of variables in $\mathbf{V}$. We say an un-directional path between $\mathbf{X}$ and $\mathbf{Y}$ is *blocked* by $\mathbf{Z}$ if one of the following requirements is met: 1) The path contains a chain $A \to B \to C$ or a fork $A \leftarrow B \to C$ such that $B \in \mathbf{Z}$; or 2) the path contains a collider $A \to B \leftarrow C$ such that $\mathbf{Z}$ contains no descendent of B. We say $\mathbf{X}$ and $\mathbf{Y}$ are *d-separated* by $\mathbf{Z}$, if $\mathbf{Z}$ blocks all un-directional paths between $\mathbf{X}$ and $\mathbf{Y}$ in $\mathcal{G}$, denoted as

$$\mathbf{X} \perp\!\!\!\perp_{\mathcal{G}} \mathbf{Y} \mid \mathbf{Z}. \tag{12}$$

**Theorem\* 2** (d-separation criterion). *Assume $\mathcal{G}$ is a DAG on a set of variables $\mathbf{V}$. Assume $\mathbf{X}$, $\mathbf{Y}$, and $\mathbf{Z}$ are three disjoint groups of variables in $\mathbf{V}$. We have:*
*1) if $p$ is any probability function compatible with $\mathcal{G}$, then*

$$(\mathbf{X} \perp\!\!\!\perp_{\mathcal{G}} \mathbf{Y} \mid \mathbf{Z}) \Rightarrow (\mathbf{X} \perp\!\!\!\perp_{p} \mathbf{Y} \mid \mathbf{Z}), \tag{13}$$

*where $\perp\!\!\!\perp_{p}$ means conditional independence under $p$, namely $p(\mathbf{Y}|\mathbf{Z}) = p(\mathbf{Y}|\mathbf{X}, \mathbf{Z})$;*
*2) if $(\mathbf{X} \perp\!\!\!\perp_{p} \mathbf{Y} \mid \mathbf{Z})$ holds for all $p$ that is compatible with $\mathcal{G}$, then $(\mathbf{X} \perp\!\!\!\perp_{\mathcal{G}} \mathbf{Y} \mid \mathbf{Z})$ also holds.*

Using the d-separation criterion, the following rule is proven by Pearl (2000).

**Theorem\* 3** (Causal Markov Condition). *Assume $\mathcal{G}$ is a DAG on a set of variables $\mathbf{V}$. Let $p$ denote a probability function for these variables. Then $p$ is compatible with $\mathcal{G}$ if and only if $(X \perp\!\!\!\perp_{p} Y | Pa_{\mathcal{G}}(X))$ holds for any $X, Y \in \mathbf{V}$ such that $Y$ is not a descendant of $X$.*

### B.3 CAUSAL DISCOVERY

Consider that $\mathbf{V}$ is a set of variables, and that $p$ is a probability function of these variables. The goal of causal discovery is to recover a DAG $\mathcal{G}$ that is compatible with $p$ from a set of observation data (sampled from $p$) of these variables. However, a probability function $p$ may be compatible with more than one DAG. For example, consider two SCMs on variables $\{X, Y, Z\}$ where $X$ is the only exogenous variable:

$$\mathcal{M}_1 : X = U_X, \ Y = X^2 + U_Y, \ Z = X + X^2 + U_Z; \tag{14}$$

$$\mathcal{M}_2 : X = U_X, \ Y = X^2 + U_Y, \ Z = X + Y + U_Z. \tag{15}$$

If the distributions of disturbances are the same in both SCMs and $U_Y \equiv 0$, then the two SCMs lead to identical probability functions. Therefore, this probability function is compatible with two different DAGs: In $\mathcal{M}_1$, we have $Pa(Z) = \{X\}$; in $\mathcal{M}_2$, we have $Pa(Z) = \{X, Y\}$.

Since there exists more than one DAG that $p$ may be compatible with, Definition\* 6 suggests that we may look for the minimal DAG that can represent the fewest probability functions, i.e. the DAG that focuses most on $p$. It is worth mentioning that in the original definitions of Pearl (2000), the observability of variables is considered, which is ignored here since all variables are observable in our work.

**Definition\* 5** (Structural preference and equivalence). Assume $\mathcal{G}_1$ and $\mathcal{G}_2$ are DAGs on the same set of variables. If any probability function $p$ compatible with $\mathcal{G}_1$ is also compatible with $\mathcal{G}_2$, we say $\mathcal{G}$ is preferred to $\mathcal{G}_2$, denoted as $\mathcal{G}_1 \preceq \mathcal{G}_2$. If we have $\mathcal{G}_1 \preceq \mathcal{G}_2$ and $\mathcal{G}_2 \preceq \mathcal{G}_1$, we say $\mathcal{G}_1$ and $\mathcal{G}_2$ are equivalent, denoted as $\mathcal{G}_1 \equiv \mathcal{G}_2$.

**Definition\* 6** (Minimal structure). Assume $\boldsymbol{G}$ is a family of DAGs defined on the same set of variables. We say $\mathcal{G} \in \boldsymbol{G}$ is the minimal DAG among $\boldsymbol{G}$ if every $\mathcal{G}' \in \boldsymbol{G}$ satisfies that $\mathcal{G} \preceq \mathcal{G}'$.

Faithfulness (also known as stability) is an important concept for causal discovery. We say $p$ is faithful to a DAG $\mathcal{G}$ if all the conditional independence relationships in $p$ are "stored" in the structure of $\mathcal{G}$. In other words, the independent relationships in $p$ stem purely from the causal structure $\mathcal{G}$ rather than coincidence. In addition, faithfulness offers a stronger condition than minimality, as it implies a unique minimal structure. Therefore, the faithfulness condition becomes a vital assumption for causal discovery, which makes the structure of the DAG identifiable. If $p$ is faithful to $\mathcal{G}$, then $\mathcal{G}$ precludes all spurious correlations. By assuming that the probability $p$ of data follows a stable distribution, we can use the Causal Faithfulness Property (Theorem\* 4) to identify the CG $\mathcal{G}$ that $p$ is compatible with and faithful to.

**Definition\* 7** (Faithfulness and stable distribution). Assume $\mathcal{G}$ is a DAG and probability function $p$ is compatible with $\mathcal{G}$. If we have

$$(\mathbf{X} \perp\!\!\!\perp_p \mathbf{Y}|\mathbf{Z}) \Rightarrow (\mathbf{X} \perp\!\!\!\perp_\mathcal{G} \mathbf{Y}|\mathbf{Z}) \tag{16}$$

for any disjoint variable groups $\mathbf{X}$, $\mathbf{Y}$, and $\mathbf{Z}$, we say $p$ is *faithful* to the DAG $\mathcal{G}$. Consider $p$ a probability function of a set of variables. If there exists a DAG $\mathcal{G}$ on these variables such that $p$ is compatible with and faithful to $\mathcal{G}$, we say $p$ follows a *stable distribution*.

**Theorem\* 4** (Causal faithfulness property). *Assume $\mathcal{G}$ is a DAG. If a probability function $p$ is compatible with and faithful to $\mathcal{G}$, we have*

$$(\mathbf{X} \perp\!\!\!\perp_\mathcal{G} \mathbf{Y} \mid \mathbf{Z}) \Leftrightarrow (\mathbf{X} \perp\!\!\!\perp_p \mathbf{Y} \mid \mathbf{Z}) \tag{17}$$

*for any disjoint variable groups $\mathbf{X}$, $\mathbf{Y}$, and $\mathbf{Z}$ in $\mathcal{G}$.*

The above Theorem\* 4 can be easily derived from Theorem\* 2. The following theorem (Peters et al., 2017) shows that faithfulness is a stronger requirement than minimality.

**Theorem\* 5** (Faithfulness implicates an unique minimal structure). *Assume $p$ is a probability function of a set of variables and $\boldsymbol{G}$ is the set of DAGs that $p$ is compatible with. If $p$ is faithful to $\mathcal{G} \in \boldsymbol{G}$, then $\mathcal{G}$ is the unique minimal DAG in $\boldsymbol{G}$.*

### B.4 CONDITIONAL INDEPENDENCE TESTS

According to Theorem\* 4, the discovery of the graph structure is converted into the determination of the conditional independence relations under a faithful probability $p$. However, the exact formulation of $p$ is usually unknown, and we have to make the judgment using samples drawn from $p$.

The technique for testing whether variables are conditionally independent is called the Conditional Independence Test (CIT). In other words, CIT uses a dataset $\{(x_i, y_i, z_i)\}_{i=1}^N$ drawn from $p$ to estimate whether the hypothesis $(\mathbf{X} \perp\!\!\!\perp_p \mathbf{Y} \mid \mathbf{Z})$ holds. A simple way to implement a CIT is to learn two linear regression models, $\hat{y} = f(x, z)$ and $\hat{y} = g(z)$, using the given data. We then define the square errors of both models:

$$\epsilon_f(x_i, y_i, z_i) = \frac{1}{N} \sum (y_i - f(x_i, z_i))^2, \tag{18}$$

$$\epsilon_g(x_i, y_i, z_i) = \frac{1}{N} \sum (y_i - g(z_i))^2. \tag{19}$$

If conditional independence holds, then $\mathbf{X}$ does not carry any information about $\mathbf{Y}$, and thus the argument $x$ will not change the regression error. Therefore, a Student t-test can be used to check whether $\epsilon_f/\epsilon_g$ is expected to be 1, which confirms the independence. Additionally, there are many more advanced tools to perform this test, such as Fast CIT Chalupka et al. (2018) and Kernel-based CIT Zhang et al. (2012).

Another way to perform the CIT is to estimate the conditional mutual information (CMI). The CMI between X and Y conditional on Z is defined as

$$CMI(\mathrm{X}, \mathrm{Y} \mid \mathrm{Z}) := \mathbb{E}_{\mathrm{X,Y,Z}} \left[ \log \frac{p(\mathrm{X, Y}|\mathrm{Z})}{p(\mathrm{X}|\mathrm{Z})p(\mathrm{Y}|\mathrm{Z})} \right] = \mathbb{E}_{\mathrm{X,Y,Z}} \left[ \log \frac{p(\mathrm{Y}|\mathrm{X,Z})}{p(\mathrm{Y}|\mathrm{Z})} \right]. \tag{20}$$

**Theorem\* 6.** $CMI(X, Y \mid Z) \geq 0$, *where equality holds if and only if* $(X \perp\!\!\!\perp_p Y \mid Z)$.

Using the theory above, we can determine whether conditional independence holds by checking whether the CMI is $0$. As suggested by Wang et al. (2022), we can estimate the CMI using the neural approximates of $p(Y|X, Z)$ and $p(Y|Z)$.

## C    THE THEORY OF CAUSAL DYNAMICS MODELS

We assume the state in a Factored Markov Decision Process (FMDP) is composed of $n_s$ state variables written as $\mathbf{S} = (S_1, \cdots, S_{n_s})$. Similarly, we have $\mathbf{A} = (A_1, \cdots, A_{n_a})$. In this section, we show 1) that there always exists a causal dynamics model (a.k.a., dynamics Bayesian network) to formulate the dynamics of an FMDP, 2) that this causal dynamics model has bipartite causal graph if state variables transit independently, and 3) how the ground-truth causal graph of an FMDP can be uniquely identified. The following discussion is based on the general form of FMDPs. However, the definitions, analysis, and conclusion are also applicable in OOMDPs, where the attributes are merely variables organized in an OO framework.

### C.1    CAUSAL STRUCTURE OF FACTORED MARKOV DECISION PROCESS

The following theorem describes the general causal structure for in transition $\mathbf{\Delta}$ of an FMDP. We first define the concept of consistency, which means that the probability function of $\mathbf{\Delta}$ follows the transition function of the FMDP whereas the state distribution $p(\mathbf{S})$ and policy $p(\mathbf{A}|\mathbf{S})$ can be arbitrary.

**Definition\* 8** (Consistent probability function). Assume $\mathbf{\Delta} = (\mathbf{S}, \mathbf{A}, \mathbf{S}')$ is the set of transition variables of an FMDP. Suppose that $p$ is a probability function of variables $\mathbf{\Delta}$. We say it is *consistent* with the dynamics, if

$$p(\mathbf{S}, \mathbf{A}, \mathbf{S}') = p(\mathbf{S}'|\mathbf{S}, \mathbf{A})p(\mathbf{A}|\mathbf{S})p(\mathbf{S}), \tag{21}$$

and $p(\mathbf{S}'|\mathbf{S}, \mathbf{A})$ is exactly the transition function of the concerned FMDP.

**Theorem\* 7.** *Assume $p$ is any probability function of a variables $\mathbf{\Delta} = (\mathbf{S}, \mathbf{A}, \mathbf{S}')$. If it is consistent with an FMDP, then there exists a DAG $\mathcal{G}$ on $\mathbf{\Delta}$ such that:*

1. *$p$ is compatible with $\mathcal{G}$;*

2. *$Pa(S_i) \subseteq \mathbf{S}$ for every $S_i \in \mathbf{S}$;*

3. *$Pa(A_i) \subseteq (\mathbf{S}, \mathbf{A})$ for every $A_i \in \mathbf{A}$*

4. *$Pa(S'_j) \subseteq (\mathbf{S}, \mathbf{A}, \mathbf{S}')$ for every $S'_j \in \mathbf{S}'$;*

5. *$\mathcal{G}$ contains no backward edge like $S'_j \to S_i$ or $A_j \to S_i$.*

*Proof.* We have

$$p(\mathbf{\Delta}) = p(\mathbf{S})p(\mathbf{A}|\mathbf{S})p(\mathbf{S}'|\mathbf{S}, \mathbf{A}). \tag{22}$$

It is easy to see that $Pa(A_i) \subseteq \mathbf{S}$ for every $A_i \in \mathbf{A}$ and $Pa(S'_j) \in (\mathbf{S}, \mathbf{A})$ for every $S'_j \in \mathbf{S}'$ if we decompose the probabilities using the chain rule. For $p(\mathbf{S}'|\mathbf{S}, \mathbf{A})$, we can write

$$p(\mathbf{S}'|\mathbf{S}, \mathbf{A}) = \prod_{j=1}^{n_s} p(S_j|\mathbf{S}, \mathbf{A}, S'_1, ..., S'_{j-1}), \tag{23}$$

and define $Pa(S'_j) \subseteq (\mathbf{S}, \mathbf{A}, S_1, ..., S_{j-1})$ as the minimal subset such that $p(S'_j|\mathbf{S}, \mathbf{A}, S'_1, ..., S'_{j-1}) = p(S'_j|Pa(S'_j))$. For $p(\mathbf{S})$ and $p(\mathbf{A}|\mathbf{S})$, we can perform similar decomposition. Therefore, the above conclusions are easy to draw. $\square$

The definition of CDMs has been provided in the paper's Definition 1. From the above proof, we can see that the parenthood of next-state variables $\mathbf{S}'$ is not affected by the choice of policy $p(\mathbf{A}|\mathbf{S})$ and the prior distribution of state $p(\mathbf{S})$. Therefore, Causal Dynamics Models (CDMs) only care about the causality of $\mathbf{S}'$. Like Definition\* 1, we define whether the causal graph can represent the dynamics of an FMDP using the product decomposition.

**Definition\* 9** (Represented dynamics). Assume $\mathcal{M} = \langle \mathcal{G}, p \rangle$ is a CDM for an FMDP. If $\mathcal{G}$ satisfies that

$$p^*(\mathbf{S}'|\mathbf{S}, \mathbf{A}) = \prod_{j=1}^{n_s} p^*(\mathrm{S}'_j|Pa(\mathrm{S}'_j)) \qquad (24)$$

for every probability function $p^*$ of $(\mathbf{S}, \mathbf{A}, \mathbf{S}')$ that is consistent with the FMDP, then we say $\mathcal{G}$ *represents* the FMDP's dynamics. Further, if we also have that $p(\mathbf{S}'|\mathbf{S}, \mathbf{A}) = p^*(\mathbf{S}'|\mathbf{S}, \mathbf{A})$ for every consistent probability function $p^*$, we say the CDM $\mathcal{M}$ *matches* the dynamics of the FMDP.

It is important to note that a CDM is **not** a causal model (Bayesian network). It does not specify the causality of $\mathbf{S}$ and $\mathbf{A}$ but focuses on the causality of the next state $\mathbf{S}'$. In other words, a CDM hopes to capture the universal rule of transitions, no matter what policy the agent uses, how each episode begins, and how each episode terminates. It is concretized to a real causal model when the policy $p(\mathbf{A}|\mathbf{S})$ and the state distribution $p(\mathbf{S})$ are given.

**Theorem\* 8** (Concretization). *Assume $\mathcal{G}$ is a causal graph that represents the dynamics of an FMDP. Then for every probability function $p$ consistent with the FMDP's dynamics, there exists a DAG $\mathcal{G}_p$ on $(\mathbf{S}, \mathbf{A}, \mathbf{S}')$ such that 1) $\mathcal{G}_p$ satisfies all propositions in Theorem\* 7, 2) $\mathcal{G}$ is a sub-graph of $\mathcal{G}_p$, and 3) $\mathcal{G}_p$ and $\mathcal{G}$ share the same parent set $Pa(\mathrm{S}'_j)$ for each next-sate variable $\mathrm{S}'_j \in \mathbf{S}'$. We call this DAG $G_P$ as a concretization of $\mathcal{G}$ under $p$.*

*Proof.* Because $\mathcal{G}$ represents the dynamics of an FMDP, we have

$$p(\mathbf{S}'|\mathbf{S}, \mathbf{A}) = \prod_{j=1}^{n_s} p(\mathrm{S}'_j|Pa(\mathrm{S}'_j)).$$

Now, we use $Pa_{\mathcal{G}_p}(\mathrm{X})$ to denote the parent set of variable X in $\mathcal{G}_p$. Since $p$ is consistent with the FMDP, we have

$$p(\mathbf{S}, \mathbf{A}, \mathbf{S}') = p(\mathbf{S}'|\mathbf{S}, \mathbf{A})p(\mathbf{A}|\mathbf{S})p(\mathbf{S}).$$

Using the chain rule to decompose $p(\mathbf{A}|\mathbf{S})$ and $p(\mathbf{S})$, we have

$$p(\mathbf{S}, \mathbf{A}, \mathbf{S}') = p(\mathbf{S}'|\mathbf{S}, \mathbf{A}) \prod_{j=1}^{n_a} p(\mathrm{A}_j|\mathbf{S}, \mathrm{A}_1, ..., \mathrm{A}_{j-1}) \prod_{i=1}^{n_s} p(\mathrm{S}_i|\mathrm{S}_1, ..., \mathrm{S}_{i-1}).$$

Then there exists $\mathcal{G}_p$ where $Pa_{\mathcal{G}_p}(\mathrm{A}_j) \subseteq (\mathbf{S}, \mathrm{A}_1, ..., \mathrm{A}_{j-1})$, $Pa_{\mathcal{G}_p}(\mathrm{S}_i) \subseteq \{\mathrm{S}_1, ..., \mathrm{S}_{i-1}\}$, and $Pa_{\mathcal{G}_p}(\mathrm{S}'_k) = Pa(\mathrm{S}'_k)$, such that

$$p(\mathbf{S}, \mathbf{A}, \mathbf{S}') = p(\mathbf{S}'|\mathbf{S}, \mathbf{A}) \prod_{j=1}^{n_a} p(\mathrm{A}_j|Pa_{\mathcal{G}_p}(\mathrm{A}_j)) \prod_{i=1}^{n_s} p(\mathrm{S}_i|Pa_{\mathcal{G}_p}(\mathrm{S}_i))$$

$$= \prod_{k=1}^{n_s} p(\mathrm{S}'_k|Pa_{\mathcal{G}_p}(\mathrm{S}'_k)) \prod_{j=1}^{n_a} p(\mathrm{A}_j|Pa_{\mathcal{G}_p}(\mathrm{A}_j)) \prod_{i=1}^{n_s} p(\mathrm{S}_i|Pa_{\mathcal{G}_p}(\mathrm{S}_i))$$

$$= \prod_{k=1}^{n_s} p(\mathrm{S}'_k|Pa(\mathrm{S}'_k)) \prod_{j=1}^{n_a} p(\mathrm{A}_j|Pa_{\mathcal{G}_p}(\mathrm{A}_j)) \prod_{i=1}^{n_s} p(\mathrm{S}_i|Pa_{\mathcal{G}_p}(\mathrm{S}_i)).$$

Then the theorem is proven with the above equations. $\qquad\square$

There may exist more than one causal graph that represents the dynamics of the dynamics. However, not all these graphs are "good" as they may contain redundant edges. In order to remove spurious correlations and improve the generalization of dynamics models, we want the CG to capture as many independent relationships as possible. Most importantly, these independent relationships should be universal. That is, they hold for every other probability function that is consistent with the dynamics so that they will not be destroyed if the agent changes its policy or we change the distribution of states. Therefore, the desired property of the causal graph is given in the following definition.

**Definition\* 10** (Dynamical faithfulness). Assume $\mathcal{G}$ is a causal graph of a CDM and $p$ is a probability function of $(\mathbf{S}, \mathbf{A}, \mathbf{S}')$. We say $p$ is *dynamically faithful* to $\mathcal{G}$, if there exists a DAG $\mathcal{G}_*$ such that 1) $p$ is compatible with and faithful to $\mathcal{G}_*$, and 2) $\mathcal{G}_*$ is a concretization of $\mathcal{G}$ under $p$.

**Definition\* 11** (Ground-truth causal graph). Assume $\mathcal{G}$ is a causal graph of a CDM for an FMDP. We say $\mathcal{G}$ a *ground-truth* causal graph of the FMDP's dynamics if it is the unique causal graph inferred from all dynamically faithful probability functions. That is, for every consistent probability function $p$ of $(\mathbf{S}, \mathbf{A}, \mathbf{S}')$ and any causal graph $\mathcal{G}'$, we have

$$p \text{ is dynamically faithful to } \mathcal{G}' \implies \mathcal{G}' = \mathcal{G}.$$

**Theorem\* 9.** *A necessary and sufficient condition for a CG $\mathcal{G}$ to be the ground-truth causal graph of the FMDP dynamics is that, for every consistent probability function $p$ and any DAG $\mathcal{G}_*$ on $(\mathbf{S}, \mathbf{A}, \mathbf{S}')$, we have*

$$p \text{ is compatible with and faithful to } \mathcal{G}_* \implies \mathcal{G}_* \text{ is a concretization of } \mathcal{G} \text{ under } p.$$

*Proof.* (Necessity) Let $\mathcal{G}'$ denotes a sub-graph of $\mathcal{G}_*$ such that $\mathcal{G}_*$ is a concretization of $\mathcal{G}'$ under $p$. We have that

$$p \text{ is compatible with and faithful to } \mathcal{G}_*$$
$$\implies p \text{ is dynamically faithful to } \mathcal{G}'$$
$$\implies \mathcal{G}' = \mathcal{G}$$
$$\implies \mathcal{G}_* \text{ is a concretization of } \mathcal{G} \text{ under } p.$$

(Sufficiency) Let $\mathcal{G}'_p$ be some concretization of $\mathcal{G}'$ under $p$. We have that

$$p \text{ is dynamically faithful to } \mathcal{G}'$$
$$\implies \exists \mathcal{G}'_p \text{ which } p \text{ is compatible with and faithful to}$$
$$\implies \exists \mathcal{G}'_p \text{ is a concretization of } \mathcal{G}$$
$$\implies \mathcal{G} = \mathcal{G}'.$$

$\square$

## C.2 BIPARTITE CAUSAL GRAPHS

Not all MDPs are suitable to be modeled by causality. For example, if the state variables are raw pixels of an image, then transitions of variables are densely correlated, leading to a dense causal graph. In this case, CDMs are deprecated, unless certain abstraction and representation techniques are performed to simplify the causal structure (not included in our work). We will make decent assumptions about the FMDP, which greatly simplifies the structure of the causal graph.

**Assumption\* 1** (Independent transition). *The transition function of the FMDP follows that*

$$p(\mathbf{S}'|\mathbf{S}, \mathbf{A}) = \prod_{j=1}^{n_s} p(\mathbf{S}'_j|\mathbf{S}, \mathbf{A}). \tag{25}$$

Several studies have assumed that the causal graph of a CDM is bipartite (Volodin, 2021; Wang et al., 2021; 2022; Ding et al., 2022). We formally define a bipartite causal graph (BCG) below. If the transition is independent (Assumption\* 1), we argue that: 1) we can use BCGs as they always exist, and 2) we should use BCGs as they are necessary for faithfulness.

**Definition\* 12** (Bipartite causal graph). Consider that $\mathcal{G}$ is the CG of a CDM. If we have $Pa(\mathbf{S}'_j) \subseteq (\mathbf{S}, \mathbf{A})$ for every $\mathbf{S}'_j \in \mathbf{S}$, we say $\mathcal{G}$ is a *bipartite causal graph* (BCG). In other words, no lateral edge like $\mathbf{S}'_i \to \mathbf{S}'_j$ exists among $\mathbf{S}'$.

**Theorem\* 10** (Existence of BCGs). *If an FMDP follows Assumption\* 1 then it is matched by some CDM whose causal graph is a BCG. In addition, in this BCG we have*

$$p(\mathbf{S}'_j|Pa(\mathbf{S}'_j)) = p(\mathbf{S}'_j|\mathbf{S}, \mathbf{A}), \quad \text{for } \mathbf{S}'_j \in \mathbf{S}'. \tag{26}$$

*Proof.* Assuming $p$ gives the transition function of the SCM. We can define the CDM as $\langle \mathcal{G}, p \rangle$. In $\mathcal{G}$, we let $Pa(\mathbf{S}'_j)$ be any subset of $(\mathbf{S}, \mathbf{A})$ such that $p(\mathbf{S}'_j|Pa(\mathbf{S}'_j)) = p(\mathbf{S}'_j|\mathbf{S}, \mathbf{A})$. Such a subset always exists since it may directly be $(\mathbf{S}, \mathbf{A})$. Using Assumption\* 1, we have

$$p(\mathbf{S}'|\mathbf{S}, \mathbf{A}) = \prod_{j=1}^{n_s} p(\mathbf{S}'_j|\mathbf{S}, \mathbf{A}) = \prod_{j=1}^{n_s} p(\mathbf{S}'_j|Pa(\mathbf{S}'_j)).$$

Then, we let $p(S'_j|Pa(S'_j))$ be equal to $p(S'_j|Pa(S'_j))$. As a result, the dynamics are matched by the CDM and $\mathcal{G}$ is a BCG. $\qquad\square$

**Theorem\* 11** (Faithfulness for BCGs). *Assume that the dynamics of an FMDP are represented by $\mathcal{G}$ and $p$ is a probability function consistent with the FMDP. If Assumption\* 1 holds, then a necessary condition of that $p$ is dynamically faithful to $\mathcal{G}$ (see Definition\* 10) is that $\mathcal{G}$ is a BCG.*

*Proof.* Assume $\mathcal{G}_p$ is the concretization of $\mathcal{G}$ under $p$ such that $p$ is faithful to $\mathcal{G}$. If $\mathcal{G}$ is not bipartite, there exist $j, k$ such that $S'_j \to S'_k$ in $\mathcal{G}_p$. In this case, we have $(S'_j \not\perp\!\!\!\perp_\mathcal{G} S'_k|\mathbf{S})$. According to Assumption 1, we have $(S'_j \perp\!\!\!\perp_p S'_k|\mathbf{S})$. Therefore, we have that

$$(S'_j \perp\!\!\!\perp_p S'_k|\mathbf{S}) \not\Leftrightarrow (S'_j \perp\!\!\!\perp_{\mathcal{G}_p} S'_k|\mathbf{S}).$$

That is, $p$ is not faithful to $\mathcal{G}_p$. Using reduction to absurdity, we prove that $\mathcal{G}$ is a BCG. $\qquad\square$

Humans decompose the world into components based on independence. Therefore, it is rational to assume that state variables transit independently (Assumption\* 1), which brings many benefits: 1) The ground-truth causal graph is a BCG so that the complexity of causal discovery is reduced; 2) The ground-truth causal graph can be uniquely identified by conditional independent tests, and 3) The computation of CDM can be implemented in parallel using GPUs.

Instead of BCGs, we note that there exists research that considers learning arbitrary CGs for CDMs (Zhu et al., 2022), where the requirement of independent transition can be released. However, this kind of CDM can not be computed in parallel, and the procedure of causal discovery is much more complicated. Learning CGs is already very expensive even though we consider only BCGs. Therefore, we suggest that Assumption 1 is vital to make causal discovery applicable in large-scale environments.

## C.3 CAUSAL DISCOVERY FOR CAUSAL DYNAMICS MODELS

The approach to identifying the CG representing the dynamics of the FMDP is already introduced in the paper's Theorem 1. However, the expression of the theorem is rather vague. Given the above definitions, we now rewrite the theorem in a more rigorous way.

**Theorem\* 12** (Causal Discovery for FMDPs). *Consider that probability function $p$ is consistent (see Definition\* 8) with the dynamics of an FMDP, where Assumption\* 1 holds. Then, there exists a causal graph $\mathcal{G}$ that represents the dynamics of the FMDP (see Definition\* 9). Assuming that $p$ is dynamically faithful to $\mathcal{G}$ (see Definition\* 10), we have*

1. *$\mathcal{G}$ is a bipartite causal graph (see Definition\* 12),*

2. *$\mathcal{G}$ is the ground-truth causal graph (see Definition\* 11) of the dynamics, and*

3. *$\mathcal{G}$ is uniquely identified by the rule that*

$$X_i \in Pa(S'_j) \Leftrightarrow (X_i \not\perp\!\!\!\perp_P S'_j | (\mathbf{S}, \mathbf{A}) \smallsetminus \{X_i\}) \tag{27}$$

   *for every $X_i \in (\mathbf{S}, \mathbf{A})$ and every $S'_j \in \mathbf{S}'$.*

*Proof.* Since $p$ is consistent with the FMDP, then the transition function is $p(\mathbf{S}'|\mathbf{S}, \mathbf{A})$. Using the chain rule, we have

$$p(\mathbf{S}'|\mathbf{S}, \mathbf{A}) = \prod_{j=1}^{n_s} p(S'_j|\mathbf{S}, \mathbf{A}, S'_1, ..., S'_{j-1}).$$

By defining $Pa(S'_j) \subseteq (\mathbf{S}, \mathbf{A}, S'_1, ..., S'_{j-1})$ as any subset such that

$$p(S'_j|\mathbf{S}, \mathbf{A}, S'_1, ..., S'_{j-1}) = p(S'_j|Pa(S'_j)).$$

we have that $\mathcal{G}$ represents the transition dynamics of the FMDP.

We use $\mathcal{G}_p$ to denote the concretization of $\mathcal{G}$ under $p$. According to Theorem\* 8, $p$ is compatible with $\mathcal{G}_p$. Having assumed that $p$ is dynamically faithful to $\mathcal{G}$, we can further assume that $p$ is also faithful to $\mathcal{G}_p$. According to Theorem\* 4, we have

$$(\mathbf{X} \perp\!\!\!\perp_p \mathbf{Y}|\mathbf{Z}) \Leftrightarrow (\mathbf{X} \perp\!\!\!\perp_{\mathcal{G}_p} \mathbf{Y}|\mathbf{Z})$$

for any disjoint variable groups $\mathbf{X}, \mathbf{Y}, \mathbf{Z}$ in $(\mathbf{S}, \mathbf{A}, \mathbf{S}')$. In addition, we have that $G$ is a BCG according to Theorem* 11.

Assume that $X_i$ is a variable in $(\mathbf{S}, \mathbf{A})$. According to the definition of d-separation, if $X_i \in Pa(S'_j)$, $X_i$ and $S'_j$ can not be d-separated by any group $\mathbf{Z}$ of variables such that $X_i, S'_j \notin \mathbf{Z}$. Letting $\mathbf{Z} = (\mathbf{S}, \mathbf{A}) \smallsetminus \{X_i\}$, we have

$$X_i \in Pa(S'_j) \Rightarrow (X_i \not\perp_{\mathcal{G}_p} S'_j | \mathbf{Z}).$$

Noticing that $\mathcal{G}$ is a BCG and $\mathcal{G}$ is a sub-graph of $\mathcal{G}_p$ (according to 8), every path from $X_i$ to $S'_j$ in $\mathcal{G}_p$ is blocked by $\mathbf{Z}$ unless $X_i \in Pa(S'_j)$. Therefore, we have

$$X_i \notin Pa(S'_j) \Rightarrow (X_i \perp\!\!\!\perp_{\mathcal{G}_p} S'_j | \mathbf{Z}).$$

In other words, we have

$$(X_i \not\perp_{\mathcal{G}_p} S'_j | \mathbf{Z}) \Rightarrow X_i \in Pa(S'_j).$$

Combing the above conclusions, we prove that

$$X_i \in Pa(S'_j) \Leftrightarrow (X_i \not\perp_{\mathcal{G}_p} S'_j | (\mathbf{S}, \mathbf{A}) \smallsetminus \{X_i\}) \Leftrightarrow (X_i \not\perp_P S'_j | (\mathbf{S}, \mathbf{A}) \smallsetminus \{X_i\}).$$

Therefore, we have that the causal graph $\mathcal{G}$ representing the dynamics of the FMDP is uniquely identified using the above rule.

Now we consider replacing $p$ with any other probability $p^*$ such that $p^*$ is faithful to some concretization $\mathcal{G}^*_{p^*}$. We use $Pa^*(S_i)$ to denote the parent set of $S'_j \in \mathbf{S}'$ in $\mathcal{G}^*_{p^*}$.

Consider that $X_i$ is a variable in $(\mathbf{S}, \mathbf{A})$ and define $\mathbf{Z} := (\mathbf{S}, \mathbf{A}) \smallsetminus \{X_i\}$. If $X_i \in Pa(S'_j)$ and $X_i \notin Pa^*(S'_j)$, using the above rule we have

$$(X_i \not\perp_p S'_j | \mathbf{Z}) \wedge (X_i \perp\!\!\!\perp_{p^*} S'_j | \mathbf{Z}).$$

In other words, we have

$$p(S'_j | \mathbf{Z}) \neq p(S'_j | \mathbf{Z}, X_i),$$
$$p^*(S'_j | \mathbf{Z}) = p^*(S'_j | \mathbf{Z}, X_i).$$

Noting that $(\mathbf{Z}, X_i) = (\mathbf{S}, \mathbf{A})$, $p^*(S'_j | \mathbf{Z}, X_i)$ is identical to $p(S_i | \mathbf{Z}, X_i)$ for every $S_i \in \mathbf{S}'$ as they are both given by the transition function of the FMDP. This leads to that

$$p(S'_j | \mathbf{Z}) \neq p^*(S'_j | \mathbf{Z}).$$

However, we can also write that

$$
\begin{aligned}
p(S'_j | \mathbf{Z}) &= \int_x p(S'_j | \mathbf{Z}, X_i = x) p(X_i = x | \mathbf{Z}) \\
&= \int_x p^*(S'_j | \mathbf{Z}, X_i = x) p(X_i = x | \mathbf{Z}) \\
&= \int_x p^*(S'_j | \mathbf{Z}) p(X_i = x | \mathbf{Z}) \\
&= p^*(S'_j | \mathbf{Z}) \int_x p(X_i = x | \mathbf{Z}) \\
&= p^*(S'_j | \mathbf{Z}).
\end{aligned}
$$

From the above equations, we obtain the paradox that

$$p(S'_j | \mathbf{Z}) = p^*(S'_j | \mathbf{Z}).$$

Using reduction to absurdity, $X_i \in Pa(S'_j)$ implies that $X_i \in Pa^*(S'_j)$. Similarly, we can prove the opposite direction of this implication. As a result, we have

$$X_i \in Pa(S'_j) \Leftrightarrow X_i \in Pa^*(S'_j),$$

which shows that $G^* = G$. Therefore, we have proven that $\mathcal{G}$ is the ground-truth causal graph.

$\square$

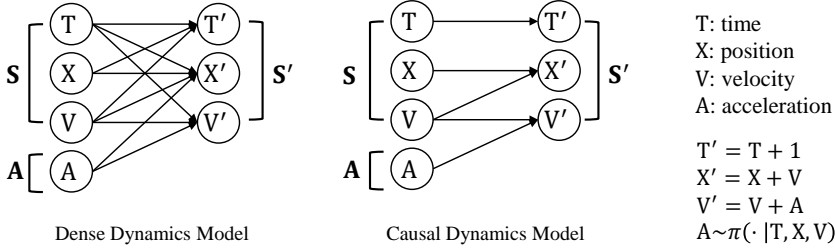

Figure 4: The dense dynamics model and causal dynamics model for a simple kinetic system.

### C.4 AN EXMPLE OF CAUSAL DYNAMICS MODEL

Consider a simple kinetic system where variables include T (time), X (position), V (velocity), and A (acceleration), where A is the action determined by the agent. Their dynamics are given in Figure 4. The dense dynamics model predicts the next-state variables using the entire input $(T, X, V, A)$. However, the CDM predicts the next-state variables using only causal parents. In Figure 4, we present the CDM with a ground-truth causal graph, and a dense dynamics model has a fully-connected structure.

Suppose that X, V, and T all start with 0. Assume that the agent approximately uses a deterministic policy:

$$\pi(X, V, T) \approx \begin{cases} 0, & V = 1, \\ 1, & V < 1, \\ -1, & V > 1. \end{cases}$$

Then V is likely to be around 1 except for the initial step. Then, a spurious correlation that $X' \approx T$ would emerge in the data. In a dense dynamics model, this spurious correlation will possibly be learned, leading to serious generalization errors when the agent changes its policy. However, the CDM does not have such a problem, as T is not a parent of $X'$.

## D THE THEORY OF OBJECT-ORIENTED MDPS

In the paper's Section 2.2 we have introduced the concept of OOMDP, where variables are composed of the attributes of objects which are described by several classes. Now, we provide more information about OOMDPs, including rigorous definitions and the proof of the paper's Theorem 2.

### D.1 RIGOROUS DEFINITIONS

**Definition\* 13** (Class). A *class*, usually denoted as $C$, is a tuple $\langle \mathcal{F}_s[C], \mathcal{F}_a[C], \boldsymbol{Dom}_C \rangle$. Here, $\mathcal{F}_s[C]$ and $\mathcal{F}_a[C]$ are disjoint sets of *fields*, where $\mathcal{F}_s[C]$ is for *state fields* and $\mathcal{F}_a[C]$ is for *action fields*; the set of all fields $C$ is defined as $\mathcal{F}[C] = \mathcal{F}_a[C] \cup \mathcal{F}_s[C]$; Each *field* in $\mathcal{F}[C]$ is a tuple like $\langle C, U \rangle$ (written as $C.U$ for short), where $C$ is exactly the class symbol $C$, and $U$ is the identifier of the field. $\boldsymbol{Dom}_C = \{Dom_{C.U}\}_{C.U \in \mathcal{F}[C]}$ gives the set of domains for each field.

**Definition\* 14** (Instance and attributes). Consider that $\mathbf{O} \subseteq (\mathbf{S}, \mathbf{A})$ is a sub-group of variables at the current step of an FMDP, and that $C = \langle \mathcal{F}_s[C], \mathcal{F}_a[C], \boldsymbol{Dom}_C \rangle$ is a class. If there exist:

1. a bijection $\beta^s : \mathcal{F}_s[C] \to \mathbf{O} \cap \mathbf{S}$ such that the domain of $\beta^s(C.U)$ is exactly $Dom_{C.U}$ for every state field $C.U \in \mathcal{F}_s[C]$,

2. and a bijection $\beta^a : \mathcal{F}_a[C] \to \mathbf{O} \cap \mathbf{A}$ such that the domain of $\beta^a(C.U)$ is exactly $Dom_{C.U}$ for every action field $C.U \in \mathcal{F}_a[C]$,

then we say that the FMDP contains an *instance* $O$ (we use the corresponding, non-bold letter) of $C$, denoted as $O \in C$. Variables in $\mathbf{O}$ are called the *attributes* of $O$, denoted by attribute symbols: $O.U := \beta^s(C.U)$ for every $C.U \in \mathcal{F}_s[C]$, or $O.U := \beta^a(C.U)$ for every $C.U \in \mathcal{F}_a[C]$.

**Definition\* 15** (Object-oriented decomposition for FMDP). Consider that $\mathcal{C} = \{C_1, \cdots, C_K\}$ is a set of classes. If $(\mathbf{S}, \mathbf{A})$ can be devided into $N$ sub-groups $(\mathbf{O}_1, \cdots, \mathbf{O}_N)$ and each $\mathbf{O}_i$ forms the

attributes of an instance $O_i$ of some class in $\mathcal{C}$, we say the FMDP is *decomposed* by $\mathcal{C}$ and call each instance $O_i$ as an *object*.

With the paper's assumptions about the result symmetry and the causation symmetry, we finally give the definition of an OOMDP below.

**Definition\* 16** (Object-oriented FMDP). We say that an FMDP is an *object-oriented FMDP* (OOMDP) on a set of classes $\mathcal{C} = \{C_1, \cdots, C_K\}$, if

1. the state variables transit independently (see Assumption\* 1),

2. the FMDP is decomposed by $\mathcal{C}$, and

3. Eqs. 2 and 3 hold under this decomposition.

### D.2  CAUSAL GRAPH FOR OOMDP

First, we prove that there always exists an OOCG to represent the dynamics of any OOMDP.

**Theorem\* 13.** *In an OOMDP, there always exists a causal graph $\mathcal{G}$ such that $\mathcal{G}$ represents the dynamics of the OOMDP (see Definition\* 9) and $\mathcal{G}$ is an OOCG.*

*Proof.* The proof is direct. Because variables transit independently, we have

$$p(\mathbf{S}'|\mathbf{S}, \mathbf{A}) = \prod_{C \in \mathcal{C}} \prod_{O \in C} \prod_{C.V \in \mathcal{F}_s[C]} p(O.V'|\mathbf{S}, \mathbf{A}).$$

Therefore, the full OOCG, where $Pa(O.V') = (\mathbf{S}, \mathbf{A})$ for each next-state attribute $O.V'$, will always represent the dynamics of the OOMDP. □

Now we prove the paper's Theorem 2 that the ground-truth causal graph (see Difinition\* 11) of an OOMDP is always an OOCG.

*Proof of the paper's Theorem 2.* Assume that $\mathcal{G}$ is the ground-truth causal graph of the OOMDP. Based on Assumption\* 1 and Theorem\* 12, we have that $\mathcal{G}$ exists, is a bipartite causal graph (BCG), and can be uniquely identified. Consider any consistent probability function $p$ and a DAG $\mathcal{G}_p$ that $p$ is compatible with and faithful to. We know that this DAG is a concretization of $\mathcal{G}$ since $\mathcal{G}$ is the ground-truth causal graph.

Assume $O_a$ and $O_b$ are both instances of class $C$, and $C.U \in \mathcal{F}[C], C.V \in \mathcal{F}_s[C]$ are fields of $C$. According to Theorem\* 4, we have that $(O_a.U \perp\!\!\!\perp_P O_a.V'|(\mathbf{S}, \mathbf{A}) \smallsetminus \{O_a.U\})$ if $O_a.U \notin Pa(O_a.V')$. In other words, if $O_a.U \notin Pa(O_a.V')$ we have

$$p(O_a.V'|\mathbf{O}_1, \cdots, \mathbf{O}_a, \cdots, \mathbf{O}_b, \cdots, \mathbf{O}_N) = p(O_a.V'|\mathbf{O}_1, \cdots, \mathbf{O}_a^{-U}, \cdots, \mathbf{O}_b, \cdots, \mathbf{O}_N),$$

where $\mathbf{O}_a^{-U}$ denotes $\mathbf{O}_a \smallsetminus \{O_a.U\}$. We define another consistent probability function $q$ such that

$$q(\mathbf{O}_1, \cdots, \mathbf{O}_a = \boldsymbol{x}, \cdots, \mathbf{O}_b = \boldsymbol{y}, \cdots, \mathbf{O}_N) := p(\mathbf{O}_1, \cdots, \mathbf{O}_a = \boldsymbol{y}, \cdots, \mathbf{O}_b = \boldsymbol{x}, \cdots, \mathbf{O}_N),$$
$$q(\mathbf{S}'|\mathbf{S}, \mathbf{A}) := p(\mathbf{S}'|\mathbf{S}, \mathbf{A}),$$

where $\boldsymbol{x}$ and $\boldsymbol{y}$ are vectors of values assigned to the objects' attributes (If $p = q$ we enforce $\boldsymbol{x} = \boldsymbol{y}$). We use $\boldsymbol{y}_{-U}$ to denote the vector where the value for the field $U \in \mathcal{F}[C]$ is missing, so that $\boldsymbol{y} = (\boldsymbol{y}_{-U}, y_U)$ where $y_U$ is the value for $U$. Then, we have

$$q(O_b.V'|\mathbf{O}_1, \cdots, \mathbf{O}_a = \boldsymbol{x}, \cdots, \mathbf{O}_b^{-U} = \boldsymbol{y}_{-U}, \cdots, \mathbf{O}_N)$$
$$= \int_{y_U} q(O_b.V'|\cdots, \mathbf{O}_a = \boldsymbol{x}, \cdots, \mathbf{O}_b = \boldsymbol{y}, \cdots) q(O_b.U = y_U|\cdots, \mathbf{O}_a, \cdots, \mathbf{O}_b^{-U} = \boldsymbol{y}_{-U}, \cdots)$$
$$= \int_{y_U} p(O_b.V'|\cdots, \mathbf{O}_a = \boldsymbol{x}, \cdots, \mathbf{O}_b = \boldsymbol{y}, \cdots) p(O_a.U = y_U|\cdots, \mathbf{O}_a^{-U} = \boldsymbol{y}_{-U}, \cdots, \mathbf{O}_b = \boldsymbol{x}, \cdots).$$

Using the result symmetry (the paper's Eq. 2), we have (continuing from the above equations)

$$= \int_{y_U} p(O_a.\mathrm{V}'|\cdots, \mathbf{O}_a = \boldsymbol{y}, \cdots, \mathbf{O}_b = \boldsymbol{x}, \cdots) p(O_a.\mathrm{U} = y_U|\cdots, \mathbf{O}_a^{-U} = \boldsymbol{y}_{-U}, \cdots, \mathbf{O}_b = \boldsymbol{x}, \cdots)$$

$$= p(O_a.\mathrm{V}'|\mathbf{O}_1, \cdots, \mathbf{O}_a = \boldsymbol{y}, \cdots, \mathbf{O}_b = \boldsymbol{x}, \cdots, \mathbf{O}_N)$$

$$= q(O_a.\mathrm{V}'|\mathbf{O}_1, \cdots, \mathbf{O}_a = \boldsymbol{y}, \cdots, \mathbf{O}_b = \boldsymbol{x}, \cdots, \mathbf{O}_N).$$

Using the result symmetry again, we have

$$q(O_a.\mathrm{V}'|\mathbf{O}_1, \cdots, \mathbf{O}_a = \boldsymbol{y}, \cdots, \mathbf{O}_b = \boldsymbol{x}, \cdots, \mathbf{O}_N) = q(O_b.\mathrm{V}'|\mathbf{O}_1, \cdots, \mathbf{O}_a = \boldsymbol{x}, \cdots, \mathbf{O}_b = \boldsymbol{y}, \cdots, \mathbf{O}_N).$$

Combining the above formulae, we have

$$q(O_b.\mathrm{V}'|\mathbf{O}_1, \cdots, \mathbf{O}_a = \boldsymbol{x}, \cdots, \mathbf{O}_b^{-U} = \boldsymbol{y}_{-U}, \cdots, \mathbf{O}_N) = q(O_b.\mathrm{V}'|\mathbf{O}_1, \cdots, \mathbf{O}_a = \boldsymbol{x}, \cdots, \mathbf{O}_b = \boldsymbol{y}, \cdots, \mathbf{O}_N),$$

which says $(O_b.\mathrm{U} \perp\!\!\!\perp_q O_b.\mathrm{V}' \mid (\mathbf{S}, \mathbf{A}) \smallsetminus \{O_b.\mathrm{U}\})$.

Since $p$ is faithful to $\mathcal{G}_p$, it is easy to prove that there exists a concretization $\mathcal{G}_q$ that $q$ is faithful to. According to Theorem* 4, we have $(O_b.\mathrm{U} \perp\!\!\!\perp_{\mathcal{G}_q} O_b.\mathrm{V}'|(\mathbf{S}, \mathbf{A}) \smallsetminus \{O_b.\mathrm{U}\})$. This leads to the corollary that $O_b.\mathrm{U} \notin Pa(O_b.\mathrm{V}')$. Therefore, we have proven that $O_a.\mathrm{U} \notin Pa(O_a.\mathrm{V}') \Rightarrow O_b.\mathrm{U} \notin Pa(O_b.\mathrm{V}')$. Similarly, we can prove that $O_a.\mathrm{U} \notin Pa(O_a.\mathrm{V}') \Leftarrow O_b.\mathrm{U} \notin Pa(O_b.\mathrm{V}')$. As a result, it is obvious that

$$O_a.\mathrm{U} \in Pa(O_a.\mathrm{V}') \Leftrightarrow O_b.\mathrm{U} \in Pa(O_b.\mathrm{V}') \tag{28}$$

So far, we have proven the shared local causality in the CG. Now, we follow a similar methodology to prove the shared global causality (we will skip some of the similar details). Assume $O_a$ and $O_b$ are both instances of $C_k$; Assume $O_i$ and $O_j$ are both instances of $C$, where $\{i, j\} \cap \{p, q\} = \varnothing$.

According to Theorem* 4, we have that $(O_a.\mathrm{U} \perp\!\!\!\perp_P O_i.\mathrm{V}'|(\mathbf{S}, \mathbf{A}) \smallsetminus \{O_a.\mathrm{U}\})$ if $O_a.\mathrm{U} \notin Pa(O_i.\mathrm{V}')$ In other words, if $O_a.\mathrm{U} \notin Pa(O_i.\mathrm{V}')$ we have

$$p(O_i.\mathrm{V}'|\mathbf{O}_a, \mathbf{O}_b, \mathbf{O}_i, \mathbf{O}_j, \cdots) = p(O_i.\mathrm{V}'|\mathbf{O}_a^{-U}, \mathbf{O}_b, \mathbf{O}_i, \mathbf{O}_j, \cdots).$$

We re-define probability function $q$ such that

$$q(\mathbf{O}_a = \boldsymbol{x}, \mathbf{O}_b = \boldsymbol{y}, \cdots) := p(\mathbf{O}_a = \boldsymbol{y}, \mathbf{O}_b = \boldsymbol{x}, \cdots),$$

$$q(\mathbf{S}'|\mathbf{S}, \mathbf{A}) := p(\mathbf{S}'|\mathbf{S}, \mathbf{A}).$$

where $\boldsymbol{x}, \boldsymbol{y}$ are vectors of values assigned to the objects' attributes. We have

$$q(O_i.\mathrm{V}'|\mathbf{O}_a = \boldsymbol{x}, \mathbf{O}_b^{-U} = \boldsymbol{y}_{-U}, \cdots)$$

$$= \int_{y_U} q(O_i.\mathrm{V}'|\mathbf{O}_a = \boldsymbol{x}, \mathbf{O}_b = \boldsymbol{y}, \cdots) q(O_b.\mathrm{U} = y_U|\mathbf{O}_a = \boldsymbol{x}, \mathbf{O}_b^{-U} = \boldsymbol{y}_{-U}, \cdots)$$

$$= \int_{y_U} p(O_i.\mathrm{V}'|\mathbf{O}_a = \boldsymbol{x}, \mathbf{O}_b = \boldsymbol{y}, \cdots) p(O_b.\mathrm{U} = y_U|\mathbf{O}_a = \boldsymbol{y}_{-U}, \mathbf{O}_b = \boldsymbol{x}, \cdots).$$

Using the causation symmetry (the paper's Eq. 3), we have (continuing the above equations)

$$= \int_{y_U} p(O_i.\mathrm{V}'|\mathbf{O}_a = \boldsymbol{y}, \mathbf{O}_b = \boldsymbol{x}, \cdots) p(O_b.\mathrm{U} = y_U|\mathbf{O}_a = \boldsymbol{y}_{-U}, \mathbf{O}_b = \boldsymbol{x}, \cdots)$$

$$= p(O_i.\mathrm{V}'|\mathbf{O}_a = \boldsymbol{y}, \mathbf{O}_b = \boldsymbol{x}, \cdots)$$

$$= q(O_i.\mathrm{V}'|\mathbf{O}_a = \boldsymbol{y}, \mathbf{O}_b = \boldsymbol{x}, \cdots).$$

Using the causation symmetry again, we obtain

$$q(O_i.\mathrm{V}'|\mathbf{O}_a = \boldsymbol{x}, \mathbf{O}_b^{-U} = \boldsymbol{y}_{-U}, \cdots) = q(O_i.\mathrm{V}'|\mathbf{O}_a = \boldsymbol{x}, \mathbf{O}_b = \boldsymbol{y}, \cdots),$$

which says $(O_b.\mathrm{U} \perp\!\!\!\perp_q O_i.\mathrm{V}'|(\mathbf{S}, \mathbf{A}) \smallsetminus \{O_b.\mathrm{U}\})$. This leads to that $O_b.\mathrm{U} \notin Pa(O_i.\mathrm{V}')$. Therefore, we can prove that $O_a.\mathrm{U} \notin Pa(O_i.\mathrm{V}') \Rightarrow O_b.\mathrm{U} \notin Pa(O_i.\mathrm{V}')$. Similarly, we can easily prove the other direction, leading to that

$$O_a.\mathrm{U} \notin Pa(O_i.\mathrm{V}') \Leftrightarrow O_b.\mathrm{U} \notin Pa(O_i.\mathrm{V}').$$

Using the result symmetry (the paper's Eq. 2), it is easy to get that

$$q(O_j.\mathbf{V}'|\mathbf{O}_a = \boldsymbol{x}, \mathbf{O}_b^{-U} = \boldsymbol{y}_{-U}, \mathbf{O}_i = \boldsymbol{z}, \mathbf{O}_j = \boldsymbol{w}, \cdots)$$
$$=q(O_i.\mathbf{V}'|\mathbf{O}_a = \boldsymbol{x}, \mathbf{O}_b^{-U} = \boldsymbol{y}_{-U}, \mathbf{O}_i = \boldsymbol{w}, \mathbf{O}_j = \boldsymbol{z}, \cdots)$$
$$=q(O_i.\mathbf{V}'|\mathbf{O}_a = \boldsymbol{x}, \mathbf{O}_b = \boldsymbol{y}, \mathbf{O}_i = \boldsymbol{w}, \mathbf{O}_j = \boldsymbol{z}, \cdots)$$
$$=q(O_j.\mathbf{V}'|\mathbf{O}_a = \boldsymbol{x}, \mathbf{O}_b = \boldsymbol{y}, \mathbf{O}_i = \boldsymbol{z}, \mathbf{O}_j = \boldsymbol{w}, \cdots).$$

which says $(O_b.\mathrm{U} \perp\!\!\!\perp_q O_j.\mathbf{V}'|(\mathbf{S}, \mathbf{A}) \smallsetminus \{O_b.\mathrm{U}\})$. This leads to that $O_b.\mathrm{U} \notin Pa(O_j.\mathbf{V}')$. Combining with the conclusion that we have just drawn, we have $O_a.\mathrm{U} \notin Pa(O_i.\mathbf{V}') \Rightarrow O_b.\mathrm{U} \notin Pa(O_j.\mathbf{V}') \Rightarrow O_a.\mathrm{U} \notin Pa(O_j.\mathbf{V}')$, and the other direction is proven similarly.

Finally, we obtain that

$$O_a.\mathrm{U} \notin Pa(O_i.\mathbf{V}') \Leftrightarrow O_b.\mathrm{U} \notin Pa(O_i.\mathbf{V}') \Leftrightarrow O_a.\mathrm{U} \notin Pa(O_j.\mathbf{V}') \Leftrightarrow O_b.\mathrm{U} \notin Pa(O_j.\mathbf{V}'). \quad (29)$$

Eqs. 28 and 29 together indicate that the causal graph is an OOCG, according to Definition 2. $\quad\square$

### D.3 OBJECT-ORIENTED CAUSAL DISCOVERY

In the main paper, we suggest using CMI for CITs, as it allows for varying numbers of instances and integrates causal discovery with model learning. The following Theorem* 14 describes how class-level causalities can be identified using CITs, providing the theoretic basis of our causal discovery. In Eqs. 30 and 31, the independence relationships in the right can be jointly tested through only one CIT, by merging the data of all concerned objects. We also note that CIT tools other than CMI are also applicable if the environment has a fixed number of instances for each class.

**Theorem* 14** (Causal discovery for OOMDPs). *The ground-truth CG $\mathcal{G}$ of an OOMDP is uniquely identified under any faithful probability function $p$ by the following rules:*

$$C.U \to V' \iff \forall O \in C\big(O.\mathrm{U} \not\perp\!\!\!\perp_P O.\mathbf{V}' \mid (\mathbf{S}, \mathbf{A}) \smallsetminus \{O.\mathrm{U}\}\big), \quad (30)$$

$$C_k.U \to C.V' \iff \forall O_j \in C\big(\mathbf{U}_{C_k.U|j} \not\perp\!\!\!\perp_P O_j.\mathbf{V}' \mid \mathbf{U}_{-C_k.U|j}\big), \quad (31)$$

*where $\mathbf{U}_{C_k.U|j} := \{O_r.\mathrm{U}\,|\,O_r \in C_k, r \neq j\}$ and $\mathbf{U}_{-C_k.U|j} := (\mathbf{S}, \mathbf{A}) \smallsetminus \mathbf{U}_{C_k.U|j}$.*

*Proof.* Using Theorem* 12 and the paper's Definition 2, it is obvious that

$$C.U \to V' \Leftrightarrow \forall O \in C(O.\mathrm{U} \in Pa(O.\mathbf{V}'))$$
$$\Leftrightarrow \forall O \in C\big(O.\mathrm{U} \not\perp\!\!\!\perp_P O.\mathbf{V}' \mid (\mathbf{S}, \mathbf{A}) \smallsetminus \{O.\mathrm{U}\}\big).$$

From the paper's Definition 2 we have

$$C_k.U \to C.V' \Leftrightarrow \forall O_r \in C_k \forall O_j \in C(r = j \vee O_r.\mathrm{U} \in Pa(O_j.\mathbf{V}')).$$

From the paper's Theorem 2, we know that $\mathcal{G}$ is an OOCG, which guarantees d-separations in $\mathcal{G}$:

$$\forall O_r \in C_k \forall O_j \in C(r = j \vee O_r.\mathrm{U} \in Pa(O_j.\mathbf{V}')) \iff \forall O_j \in C\big(\mathbf{U}_{C_k.U|j} \not\perp\!\!\!\perp_{\mathcal{G}} O_j.\mathbf{V}' \mid \mathbf{U}_{-C_k.U|j}\big).$$

Using Theorem* 4 then we have

$$\forall O_j \in C\big(\mathbf{U}_{C_k.U|j} \not\perp\!\!\!\perp_{\mathcal{G}} O_j.\mathbf{V}' \mid \mathbf{U}_{-C_k.U|j}\big)$$
$$\iff \forall O_j \in C\big(\mathbf{U}_{C_k.U|j} \not\perp\!\!\!\perp_p O_j.\mathbf{V}' \mid \mathbf{U}_{-C_k.U|j}\big).$$

That is, we have

$$C_k.U \to C.V' \Leftrightarrow \forall O \in C\big(\mathbf{U}_{C_k.U|j} \not\perp\!\!\!\perp_P O.\mathbf{V}' \mid \mathbf{U}_{-C_k.U|j}\big).$$

$\quad\square$

# E DETAILS OF IMPLEMENTATION

## E.1 STRUCTURE OF OBJECT-ORIENTED CAUSAL DYNAMICS MODELS

In an OOMDP, each attribute (variable) may contain one or several scalars. To handle the heterogeneous nature of different attributes, the OOCDM uses an *attribute encoder* $\text{AttrEnc}_{C.U}$ : $Dom_{C.U} \rightarrow \mathbb{R}^{d_e}$ for each field $C.U \in \mathcal{F} = \bigcup_{C \in \mathcal{C}} \mathcal{F}[C]$. It maps the attribute $O.U$ of every instance $O \in C$ to a $d_e$ dimensional *attribute-encoding vector*. All attribute encoders are implemented by a multi-layer perceptron where we use ReLU as the activation function.

Consider that $f_{C.V}$ is the predictor for the state field $C.V \in \mathcal{F}_s$ in an OOCDM. To compute $f_{C.V}(O_j.V'|\mathbf{O}_j; \mathbf{U}_{-O_j}; \mathcal{G})$ for any $O_j \in C$, we first use the above encoders to encode all observed variables. Assume that the value of the attribute $O_i.U$ of an object $O_i \in \mathcal{O}$ is observed to be $O_i.u$ (the corresponding lower-case letter is used) and the class of $O_i$ is $C_k$, Then, this attribute is encoded into the attribute-encoding vector denoted as:

$$O_i.\boldsymbol{u} := \text{AttrEnc}_{C_k.U}(O_i.u) \in \mathbb{R}^{d_e}, \quad U \in \mathcal{F}[C_k].$$

We now mask off the encoding vector if the attribute is not a parent variable for $O_j.V'$ based on the OOCG $\mathcal{G}$. That is, we define the *masked attribute-encoding vector* of attribute $O_i.U$ for $O_j.V'$ as:

$$[O_i.\boldsymbol{u}]_{O_j.V'} := \begin{cases} \mathbf{0}, & \text{if } j \neq i \text{ and } C_k.U \rightarrow C.V' \\ \mathbf{0}, & \text{if } j = i \text{ and } C.U \rightarrow V' \\ O_i.\boldsymbol{u}, & \text{otherwise.} \end{cases}$$

We concatenate all masked attribute-encoding vectors of $O_i$, and then we obtain a $(|\mathcal{F}[C_k]|d_e)$-dimensional vector called the *object-encoding vector* of $O_i$, denoted as $\boldsymbol{x}_i$:

$$\boldsymbol{x}_i = Concat\left([O_i.\boldsymbol{u}]_{O_j.V'} \text{ for } C_k.U \in \mathcal{F}[C_k]\right).$$

Then, we apply a *query encoder*, denoted as $\text{QEnc}_{C.V}$ that maps $\boldsymbol{x}_j$ to the *query vector* $\boldsymbol{q}$:

$$\boldsymbol{q} = \text{QEnc}_{C.V}(\boldsymbol{x}_j) \in \mathbb{R}^{d_k}.$$

For every other object $O_i$ such that $j \neq i$ (we denote the class of $O_i$ as $C_k$), we apply a *key encoder* $\text{KEnc}_{C_k \rightarrow C.V}$ and a *value encoder* $\text{VEnc}_{C_k \rightarrow C.V}$ that respectively map $\boldsymbol{x}_i$ to a key-vector $\boldsymbol{k}_i$ and a value-vector $\boldsymbol{v}_i$:

$$\boldsymbol{k}_i = \text{KEnc}_{C_k \rightarrow C.V}(\boldsymbol{x}_i) \in \mathbb{R}^{d_k},$$
$$\boldsymbol{v}_i = \text{VEnc}_{C_k \rightarrow C.V}(\boldsymbol{x}_i) \in \mathbb{R}^{d_v}.$$

Then, we perform the key-value attention (Vaswani et al., 2017):

$$\alpha_i = \frac{\exp\left(\boldsymbol{q}^T \boldsymbol{k}_i / \sqrt{d_k}\right)}{\sum_{r \neq i} \exp\left(\boldsymbol{q}^T \boldsymbol{k}_r / \sqrt{d_k}\right)},$$
$$\boldsymbol{h} := (\boldsymbol{q}, \sum_{j \neq i} \alpha_i \boldsymbol{v}_i) \in \mathbb{R}^{d_k + d_v},$$

where $\boldsymbol{h}$ is called the *distribution embedding* of $O_j.V'$.

Finally, we use a distribution decoder $\text{Dec}_{C.V}$ to map $\boldsymbol{h}$ into the distribution of $\hat{p}(O_j.V'|Pa(O_j.V'))$. If $Dom_{C.V}$ is continuous, it outputs the mean and standard variance of a normal distribution:

$$(\mu, \sigma) = Dec_{C.V}(\boldsymbol{h}); \quad p(O_j.V'|Pa(O_j.V')) \sim \mathcal{N}(\mu, \sigma).$$

If $Dom_{C.V}$ is discrete (we assume that $Dom_{C.V}$ has $m$ elements), then $\text{Dec}_{C.V}$ outputs the probability of each choice:

$$(p_1, \cdots, p_m) = Dec_{C.V}(\boldsymbol{h}); \quad p(O_j.V'|Pa(O_j.V')) \sim \text{Categorical}(p_1, \cdots, p_m).$$

The illustration of the structure of such a predictor is presented in Figure 3(c) of the main paper, where $i = 1$. So far, we have described the structure of one single predictor $f_{C.V}$, and other predictors follow the same design as $f_{C.V}$. In addition, it is possible to compute $\hat{p}(O_j.V'|Pa(O_j.V'))$ for all $O_j \in C$ in parallel. Therefore, the predictor $f_{C.V}$ actually outputs $\hat{p}(O_j.V'|Pa(O_j.V'))$ for all $O_j \in C$ once-for-all in our implementation (read our code for more detail).

E.2 THE ALGORITHM OF OBJECT-ORIENTED CAUSAL DISCOVERY

We define the following notations:

1. $s_t$, $a_t$, and $s_{t+1}$ are the observed values of $\mathbf{S}$, $\mathbf{A}$, and $\mathbf{S}'$ at step $t$.

2. $O_j.v_{t+1}$ to denote the observed value of attribute $O_j.\mathrm{V}$ at step $t+1$.

3. $C^t$ denotes the set of instances of class $C$ at step $t$.

Then, the pseudo-code of object-oriented causal discovery is provided in Algorithm 1.

---

**Algorithm 1** Object-oriented causal discovery

---

**Require:** The dataset $\mathcal{D} = \{(s_t, a_t, a_{t+1})\}_{t=1}^T$, predictors $\{f_{C.V}\}_{C \in \mathcal{C}, C.V \in \mathcal{F}_s[C]}$, and $\varepsilon \geq 0$.
1: Initialize $\mathcal{G} \longleftarrow$ empty OOCG.
2: **for** $C.V$ in $\bigcup_{C \in \mathcal{C}} \mathcal{F}_s[C]$ **do**
3:      $\mathcal{L} \leftarrow \sum_{t=1}^T \sum_{O_j \in C^t} \log f_{C.V}(O_j.v_{t+1} | s_t, a_t; \mathcal{G}_1)$.
4:      **for** $C.U$ in $\mathcal{F}[C]$ **do**
5:          $\tilde{\mathcal{L}} \leftarrow \sum_{t=1}^T \sum_{O_j \in C^t} \log f_{C.V}(O_j.v_{t+1} | s_t, a_t; \mathcal{G}_{C.U \not\to V'})$.
6:          $CMI_{\mathcal{D}}^{C.U \to V'} \leftarrow \frac{1}{\sum_{t=1}^T |C^t|}(\mathcal{L} - \tilde{\mathcal{L}})$.
7:          Add $C.U \to V'$ into $\mathcal{G}$ if $CMI_{\mathcal{D}}^{C.U \to V'} > \varepsilon$.
8:      **for** $C_k.U$ in $\bigcup_{C_k \in \mathcal{C}} \mathcal{F}[C_k]$ **do**
9:          $\tilde{\mathcal{L}} \leftarrow \sum_{t=1}^T \sum_{O_j \in C^t} \log f_{C.V}(O_j.v_{t+1} | s_t, a_t; \mathcal{G}_{C_k.U \not\to C.V'})$.
10:         $CMI_{\mathcal{D}}^{C_k.U \to C.V'} \leftarrow \frac{1}{\sum_{t=1}^T |C^t|}(\mathcal{L} - \tilde{\mathcal{L}})$.
11:         Add $C_k.U \to C.V'$ into $\mathcal{G}$ if $CMI_{\mathcal{D}}^{C_k.U \to C.V'} > \varepsilon$.
12: **return** $\mathcal{G}$

---

E.3 THE ALGORITHM OF MODEL LEARNING

The model is learned by optimizing the target function defined in the paper's Eq. 8 $J(\mathcal{D})$, with a given data-set $\mathcal{D}$. However, it is impractical and expensive to compute $J(\mathcal{D})$ if $\mathcal{D}$ contains too many samples. Therefore, we use stochastic gradient ascent, in which we repeatedly sample a batch $\mathcal{B} \subset \mathcal{D}$ and maximize $J(\mathcal{B})$. The pseudo-code of learning our OOCDM is provided in Algorithm 2. In this algorithm, we consider both online and offline settings, although in our experiments we only adopt offline learning to best exploit the advantage of generalization.

---

**Algorithm 2** Learning Object-oriented Causal Dynamics Model

---

**Require:** The dataset $\mathcal{D}$, number $n_{iter}$ of iterations, and number $n_{batch}$ of batches in each iteration.
1: Initialize predictors $f_{C.V}$ for every $C.V \in \bigcup_{C \in \mathcal{C}} \mathcal{F}_s[C]$.
2: **for** $i_{iter} = 1, \cdots, n_{iter}$ **do**
3:      Obtain $\hat{\mathcal{G}}$ using causal discovery (Algorithm 1).
4:      **for** $i_{batch} = 1, \cdots, n_{batch}$ **do**
5:          Sample batch $\mathcal{B} \subset \mathcal{D}$.
6:          Perform gradient ascent on $\mathcal{J}(\mathcal{B})$ defined in the paper's Eq. 8.
7:      Optionally, collect new data into $\mathcal{D}$ using the latest policy.        ▷ for online learning only
8: **return** predictors $\{f_{C.V}\}_{C \in \mathcal{C}, C.V \in \mathcal{F}_s[C]}$ and $\hat{\mathcal{G}}$.

---

E.4 PLANNING WITH DYNAMICS MODELS

We combine dynamics models with Model Predictive Control (MPC) (Camacho & Bordons, 1999), where the Cross-Entropy Method (CEM) (Botev et al., 2013) is used as the planning algorithm to determine the agents' actions. Given a planning horizon $H$, the following process is repeated several times: 1) First, we sample $k$ action sequences with lengths of $H$ from a distribution $p_{\boldsymbol{\Theta}}(\mathbf{A}_1, \cdots, \mathbf{A}_H)$ parameterized by $\boldsymbol{\Theta}$; 2) then, we use the dynamics models to perform counterfactual reasoning

with these action sequences, which generates $k$ $H$-step trajectories; 3) among these trajectories, we choose the top-$k^*$ (we have $k^* < K$) trajectories with the highest returns to update the parameter $\boldsymbol{\Theta}$. In the final iteration, we return the first action in the trajectory that produces the highest return.

Since our work only focuses on the dynamics, we assume that true reward function $R(\mathbf{S}, \mathbf{A}, \mathbf{S}')$ of the environment is given so that an extra reward model is not required. This makes sure that no reward bias is introduced in our comparison between different kinds of dynamics models. We present the pseudo-code of planning in Algorithm 3.

---

**Algorithm 3** Planning with Cross Entropy Method

---

**Require:** The dynamics model $\hat{p}$, the reward function $R$, the current state $\boldsymbol{s}$, the planning horizon $H$, the number $n_{plan}$ of iterations, the number $k$ of samples, the number $k^*$ of elite samples, and the discount factor $\lambda$.

1: Initialize the parameter $\boldsymbol{\Theta}$.
2: **for** $i = 1, \cdots, n_{plan}$ **do**
3:     **for** $j = 1, \cdots, k$ **do**
4:         Sample the $j$-th $H$-step action sequences $(\boldsymbol{a}_1^{(j)}, \cdots, \boldsymbol{a}_H^{(j)})$ with $p_{\boldsymbol{\Theta}}(\mathbf{A}_1, \cdots, \mathbf{A}_H)$.
5:         $\boldsymbol{s}_1^{(j)} \leftarrow \boldsymbol{s}$.
6:         **for** $t = 1, \cdots, H$ **do**
7:             Sample $\boldsymbol{s}_{t+1}^{(j)}$ using $\hat{p}(\mathbf{S}'|\mathbf{S} = \boldsymbol{s}_t^{(j)}, \mathbf{A} = \boldsymbol{a}_t^{(j)})$.
8:             Compute the reward $r_t^{(j)} \leftarrow R(\boldsymbol{s}_t, \boldsymbol{a}_t, \boldsymbol{s}_{t+1})$.
9:         Compute the return $r_i \leftarrow \sum_{t=1}^{H} \gamma^{t-1} r_t^{(j)}$.
10:     **if** $i < n_{plan}$ **then**
11:         $\boldsymbol{E} \leftarrow$ the set of top-$k^*$ action sequences with the highest return $r_i$ ($j \in \{1, \cdots, k\}$).
12:         $\boldsymbol{\Theta} \leftarrow$ Maximum-Likelihood-Estimation($\boldsymbol{E}$).
13:     **else**
14:         $j^* \leftarrow \arg\max_{j} r_i$.
15:         **return** $\boldsymbol{a}_1^{(j^*)}$.

---

## F COMPLEXITY ANALYSIS

In this section, we only consider one OOMDP so that the numbers of the instances of classes are fixed. The following symbols are used in this section:

1. $N_i$ denotes the number of instances of the $i$-th class $C_i$.

2. $K$ denotes the number of classes.

3. $N := \sum_{i=1}^{K} N_i$ denotes the number of objects.

4. $m_i := |\mathcal{F}[C_i]|$ denotes the number of fields of the $i$-th class $C_i$.

5. $m := \sum_{i=1}^{K} m_i$ denotes the overall number of fields in the OOMDP.

6. $n := \sum_{i=1}^{K} N_i m_i$ denotes the number of variables (attributes) at each step in the OOMDP.

7. $k$ denotes the number of samples used in predicting, causal discovery, or planning.

8. $k^*$ denotes the number of elite samples used in the Cross-Entropy Method (CEM) for planning.

9. $H$ denotes the planning horizon in Model Predictive Control (MPE) for planning.

10. $l$ denotes the number of iterations in CEM.

11. Most importantly, $O$ becomes the symbol for an asymptotic boundary rather than an object (in this section only).

It is obvious that $n \geq N$ and $n \geq m$ hold in all OOMDPs. Especially, in large-scale environments, we have $n >> m$. The theorems about the complexities of our OOCDM (implemented as described in Appendix E) are presented and proven in the following.

**Theorem\* 15.** *The time complexity of predicting the next states using our OOCDM is $O(nNk)$.*

*Proof.* Since attribute encoders are shared by encoders, then computing all attribute-encoding vectors costs $O(nk)$. Then, for every state field $S \in F_{C_i}^s$ of every class $C_i$, the predictor spends:

1. $O(nk)$ in applying masks to and concatenating attribute-encoding vectors into object-encoding vectors;

2. $O(Nk)$ in deriving key, value, and query vectors from object-encoding vectors;

3. $O(N_i(N - N_i)k + N_ik) = O(N_iNk)$ in the attention operation;

4. $O(N_ik)$ in decoding the distribution embedding.

Therefore, each state field in $F_{C_i}$ leads to a cost of $O(nk) + O(Nk) + O(N_iNk) + O(N_ik) = O(N_iNk)$. By summing up the costs of all state fields, the cost of predicting the next states is

$$O(nk) + \sum_{i=1}^{K} m_c \cdot O(N_iNk)$$
$$= O(nk) + O\left(N \sum_{i=1}^{K} m_C N_i k\right)$$
$$= O(nk) + O(nNk)$$
$$= O(nNk).$$

$\square$

**Theorem\* 16.** *The time complexity of causal discovery using our OOCDM is $O(nmNk)$.*

*Proof.* In the process of proving Theorem\* 15, we know that each predictor $f_{C_i.S}$ costs $O(N_iNk)$ to predict the attribute

First, we consider the local causality expressions. For each class $C_i$, we have $m_i^2$ local causalities. For each local causality expression $localcausC_iUV$, the predictor $f_{C_i.S}$ is used twice for each sample to estimate $CMI_{\mathcal{D}}^{localcausC_iUV}$. Therefore, the complexity of discovering all local causality expressions shared by $C_i$ is $O(m_i^2 N_i Nk)$

Then, we consider the global causality expressions. For each class $C_i$, we have $m_i m$ global causalities. For each global causality expression like $C_j.U \to C_i.V'$, the predictor $f_{C_i.S}$ is used twice for each sample to estimate $CMI_{\mathcal{D}}^{C_j.U \to C_i.V'}$. Therefore, the complexity of discovering all global causality expressions shared by $C_i$ is $O(m_i m N_i Nk)$

Combing the above results, all causality expressions (local and global) shared by $C_i$ cost
$$O(m_i^2 N_i Nk) + O(m_i m N_i Nk) = O(m_i m N_i Nk).$$
Finally, the time complexity for causal discovery is
$$\sum_{i=1}^{K} O(m_i m N_i Nk)) = O\left(mNk \sum_{i=1}^{K} O(m_i N_i)\right) = O(nmNk).$$

$\square$

**Theorem\* 17.** *The time complexity of planning for an action using our OOCDM is $O(lk(HnN + \log k^*))$.*

*Proof.* In each iteration, we have $k$ action sequences with lengths of $H$. Therefore, sampling the action sequences costs $O(kH)$. Then, using models simulating trajectories and computing returns cost $H \cdot O(nNk) = O(HnNk)$. Identifying the top-$k^*$ trajectories costs $O(k \log k^*)$. Re-estimating parameters costs $O(k * H)$. Therefore, the cost of each iteration is
$$O(kH) + O(HnNk) + O(k \log k^*) + O(Hk^*) = O(k(HnN + \log k^*)).$$
Finally, considering $l$ iterations, the time complexity of planning for an action of our OOCDM is $O(lk(HnN + \log k^*))$

$\square$

|  | **OOCDM** | CDL | FCIT+GRU |
|---|---|---|---|
| t.c. of predicting | $O(nNk)$ | $O(n^2k)$ | $O(n^2k)$ |
| t.c. of causal discovery | $O(nmNk)$ | $O(n^3k)$ | $O(n^3k\log k)$ |
| t.c. of planning | $O(lk(HnN + \log k^*))$ | $O(lk(Hn^2 + \log k^*))$ | $O(lk(Hn^2 + \log k^*))$ |
| s.c. of model weights | $O(m^2)$ | $O(n^2)$ | $O(n^2)$ |

Table 8: Comparison of computational complexity between our OOCDM and the state-of-the-art CDMs. Here "t.c." means "time complexity" and "s.c." means "space complexity".

**Theorem\* 18.** *The space complexity of model weights of our OOCDM is $O(m^2)$.*

*Proof.* The space complexity of attribute encoders is $O(m)$. In each predictor, there exists $K$ key encoders, $K$ value encoders, one query encoder, and one distribution decoder. Here, the space complexity of the key-encoder, value-encoder, or query-encoder for each class $C_i$ is $O(m_i)$; and the space complexity of the distribution decoder is $O(1)$.

Finally, the space complexity of the entire OOCDM is

$$\sum_{i=1}^{K} m_i \left( 2\sum_{j=1}^{K} O(m_j) + 2\cdot O(m_i) + O(1) \right) + O(m)$$

$$= \sum_{i=1}^{K} m_i O(m) + O(m)$$

$$= O(m^2).$$

$\square$

In Table 8, we further compare our OOCDM with the state-of-the-art CDMs in terms of the above-mentioned aspects of computational complexity. These baselines include 1) **CDL**, which learns the SCM underlying the environmental dynamics by estimating CMIs like us Wang et al. (2022), and 2) the **FCIT+GRU**, which uses FCIT to discover causalities and uses GRUs to fit structural equations (Ding et al., 2022). We can see that our OOCDM utilizes object-oriented information to share sub-models (predictors) and causality among objects of each class, leading to a great reduction of computational complexity, especially the scale of model weights and the time complexity of causal discovery. It is worth noting that all predictors are implemented in parallel in practice, making our OOCDM even more computationally efficient if GPUs are used.

## G Environments

### G.1 Block

The Block environment is a simple environment designed to validate the effectiveness of causal discovery for different numbers of environmental variables. It contains two classes: $\mathcal{C} = \{Block, Total\}$. The fields of these classes are given by

- $\mathcal{F}_s[Block] = \{Block.S_1, Block.S_2, Block.S_3\}$
- $\mathcal{F}_a[Block] = \{Block.A\}$,
- $\mathcal{F}_s[Total] = \{Total.S_1, Total.S_2, Total.S_3, Total.T\}$,
- $\mathcal{F}_a[Total] = \varnothing$.

The transition of each $O \in Block$ follows a linear transform:

$$\begin{pmatrix} O.S_1' \\ O.S_2' \\ O.S_3' \end{pmatrix} = \begin{bmatrix} 1 & 0 & 0 & -0.3 \\ 0.5 & 1.0 & 0 & 0 \\ 0 & 0.25 & 0.75 & 1.0 \end{bmatrix} \begin{pmatrix} O.S_1 \\ O.S_2 \\ O.S_3 \\ \tanh O.A \end{pmatrix} + \mathcal{N}(\mathbf{0}, 0.01^2 \boldsymbol{I}). \tag{32}$$

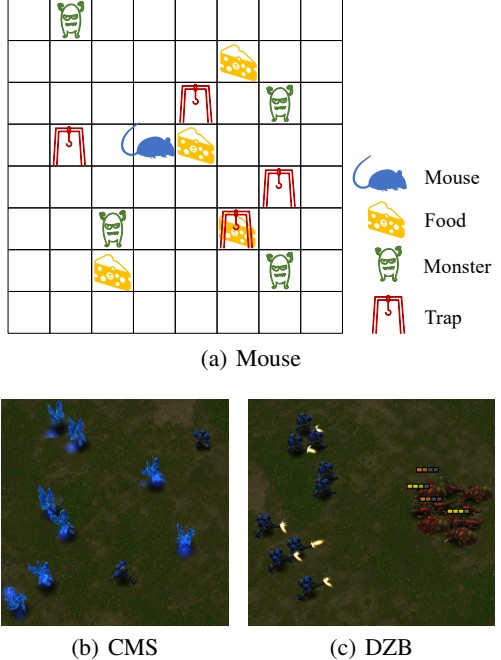

(a) Mouse

(b) CMS          (c) DZB

Figure 5: Illustrations of the (a) Mouse, (b) Collect-Mineral-Shards, and (c) Defeat-Zerglings-Banelings environments.

The transition of the instance of $Total$ follows that

$$O.S'_j = \frac{1}{2}O.S_j + \frac{1}{2}\max_{O_i \in Block} O_i.S_j, \qquad j = 1, 2, 3, \tag{33}$$

$$O.T' = O.T + 1 + \mathcal{N}(0, 0.01^2). \tag{34}$$

The Block environment contains no rewards. That is, $R(\mathbf{S}, \mathbf{A}, \mathbf{S}') \equiv 0$.

At the beginning of each episode, We initialize the attributes of each $Block$ object by

$$(O.S_1, \, O.S_2, \, O.S_3)^T \sim \mathcal{N}\left((1, 0, 0)^T, \mathrm{diag}\left(0.25, 1, 1\right)\right), \tag{35}$$

and initialize the $Total$ instance by

$$(O.S_1, \, O.S_2, \, O.S_3, \, O.T)^T \sim \mathcal{N}\left(\mathbf{0}, \mathrm{diag}\left(0.01^2, 0.01^2, 0.01^2, 0\right)\right). \tag{36}$$

We use a random policy (which produces Gaussian actions) to obtain the training data. Therefore, $O.S_1$ for every $O \in Block$ is likely to stay close to $1$. Further, this leads to spurious correlations such as $Total.T \to Block.S'_2$.

The ground-truth causal graph of the Block environment is an OOCG, which we visualize in Figure 6.

## G.2 MOUSE

The Mouse Environment aims to validate the performance of dynamics models in a more complicated OOMDP. It contains four classes: $\mathcal{C} = \{Mouse, Food, Monster, Trap\}$, whose fields are

- $\mathcal{F}_s[Mouse] = \{Mouse.Health, Mouse.Hunger, Mouse.Position\}$,
- $\mathcal{F}_a[Mouse] = \{Mouse.Move\}$,
- $\mathcal{F}_s[Food] = \{Food.Amount, Food.Position\}$,
- $\mathcal{F}_s[Monster] = \{Monster.Noise, Monster.Position\}$,

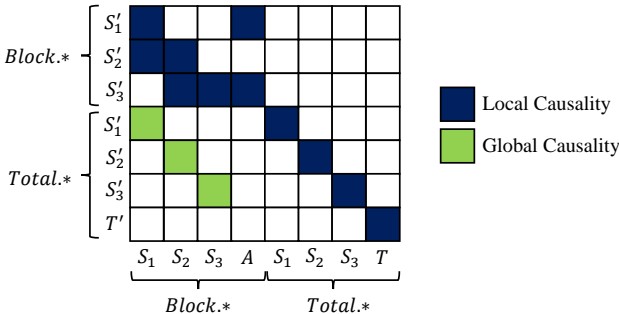

Figure 6: The visualization of the ground-truth OOCG (the adjacency matrix of class-level causalities) of the Block environment.

- $\mathcal{F}_s[Trap] = \{Trap.Duration, Trap.Position\}$,
- $\mathcal{F}_a[Food] = \mathcal{F}_a[Monster] = \mathcal{F}_a[Trap] = \varnothing$

All objects are located in an $8 \times 8$ grid world. That is, the domain of the field $Position$ of every class is in $\{0, 1, \cdots, 7\}^2$. Typically, the environment contains only one instance of $Mouse$ and arbitrary numbers of instances of other classes. We illustrate the Mouse environment in Figure 5(a).

The instance $O_{mouse}$ of $Mouse$ has an attribute $O_{mouse}.\text{Health} \leq 10$ and $O_{mouse}.\text{Hunger} \in [0, 100]$. The hunger point $O_{mouse}.\text{Hunger}$ is reduced by $1$ for each step unless the mouse reaches any instance of food. For each $O_{food} \in Food$ that is reached by the mouse (i.e., $O_{food}.\text{Position} = O_{mouse}.\text{Position}$), the mouse consumes all amount of the food ($O_{food}.\text{Amount}' \leftarrow 0$) and restores the equal amount of $O.Hunger$. If the mouse is starving ($O_{mouse}.\text{Hunger} < 25$), it loses one point of $O.\text{Health}$ for each step. However, if the mouse is full ($O_{mouse}.\text{Hunger} > 75$), it restores one point of $O.\text{Health}$ for each step. If the health $O_{mouse}.\text{Health}$ drops below $0$, the episode terminates because the mouse is dead.

The mouse has an action attribute $O_{mouse}.\text{Move}$, which can be chosen from $5$ choices: North, South, East, West, and Staying. Except for Staying, the mouse's position $O_{mouse}.\text{Position}$ changes to the nearby grid based on the chosen direction (unless it reaches the boundary of the world). However, if the mouse is trapped by a trap $O_{trap} \in Trap$ (i.e., $O_{trap}.\text{Position} = O_{mouse}.\text{Position}$ and $O_{trap}.\text{Duration} > 0$), then the mouse's position will not be changed no matter what $O_{mouse}.\text{Move}$ is chosen, and $O_{trap}.\text{Duration}$ is reduced by $1$.

The positions of $Food$ instances are fixed after being randomly initialized. The amount $O_{food}.\text{Amount}$ of an instance $O_{food}$ slowly accrues with time. That is $O_{food}.\text{Amount}' \leftarrow O_{food}.\text{Amount} + \mathcal{N}(1, 0.01)$ unless it is consumed by the mouse. We note that $O_{food}.\text{Amount}$ increases slower than that $O_{mouse}.\text{Hunger}$ decreases. Therefore, the mouse must constantly navigate from one food to another to prevent from starving.

An instance $O_{monster}$ of $Monster$ randomly wanders in the world. That is, its position randomly changes into a nearby grid at each step. If the mouse is reached by a monster (i.e., $O_{monster}.\text{Position} = O_{mouse}.\text{Position}$), the mouse directly loses $5$ points of $O_{mouse}.\text{Health}$. Each monster also contains an attribute $O_{monster}.\text{Noise}$ of noise, which is used to create spurious correlations and confuse non-causal dynamics models. The transition of $O_{monster}.\text{Noise}$ is given by $O_{monster}.\text{Noise}' = O_{monster}.\text{Noise} + \mathcal{N}(0, 0.01)$.

The goal of the agent is to make the mouse live as long as possible and stay away from starving. Therefore, the reward function is given by:

$$R(\mathbf{S}, \mathbf{A}, \mathbf{S}') = 0.01 \cdot O_{mouse}.\text{Hunger} + (O_{mouse}.\text{Health}' - O_{mouse}.\text{Health} \\ + 0.05 \cdot (O_{mouse}.\text{Hunger}' - O_{mouse}.\text{Hunger}). \tag{37}$$

The ground-truth causal graph of the Mouse environment is an OOCG, which we visualize in Figure 7.

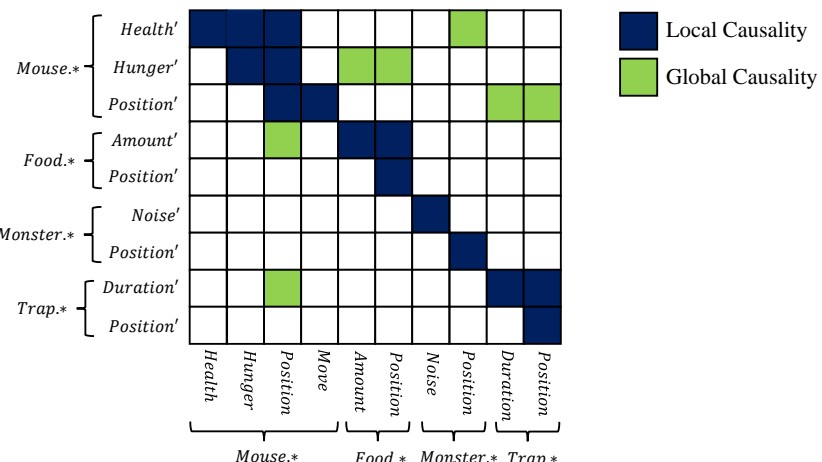

Figure 7: The visualization of the ground-truth OOCG (the adjacency matrix of class-level causalities) of the Mouse environment.

## G.3 STARCRAFT MINI-GAMES

Our experiments consider two StarCraft mini-games as environments. We formulate these environments as OOMDPs merely based on our intuition. That is, the objects correspond to units in the StarCraftII game and classes correspond to the type of the unit. The values of the attributes are observed through the PySC2 (Vinyals et al., 2017) interface.

The true dynamics of these environments are implemented in the StarCraftII engine. Being non-developers of the game, we do not know the precise dynamics of these environments. For example, we observe that if a marine chooses NOOP as its action, it automatically attacks a hostile unit (if any) in its attacking range. However, we have no clue based on what rule it chooses the unit that it attacks. Therefore, we do not know whether the Definition* 16 of OOMDP is strictly satisfied (possibly not), not to mention the ground-truth causal graph of these environments.

We believe that 1) humans factorize the world into components (variables) based on independent relationships, and 2) We discriminate and categorize objects based on structural and dynamical similarity. Therefore, we believe that the Definition* 16 is roughly satisfied, even though the object-oriented description is provided by non-experts. Through these StarCraft mini-games, we hope to show that our OOCDM is applicable to a wide range of RL problems.

### G.3.1 COLLECT-MINERAL-SHARDS

The Collect-Mineral-Shards (CMS) environment is a StarCraftII mini-game. The game contains 2 marines and 20 mineral shards. The player (agent) needs to control the movement of the marines to collect all mineral shards as fast as possible. We illustrate the CMS environment in Figure 5(b).

We decompose the environment by 2 classes $\mathcal{C} = \{Marine, Mineral\}$ such that

- $\mathcal{F}_a[Marine] = \{Marine.Position\}$, $F^a_{Marine} = \{Marine.Move\}$,
- $\mathcal{F}_s[Mineral] = \{Mineral.Position, Mineral.Collected\}$, $F^a_{Mineral} = \varnothing$,

where we set

- $Dom_{Marine.Position} = Dom_{Mineral.Position} = [-99, 99]^2$,
- $Dom_{Marine.Move} = \{North, East, South, West, Staying\}$,
- and $Dom_{Mineral.Collected} = \{True, False\}$.

We define the reward function as the number of collected mineral shards in each step:

$$R(\mathbf{S}, \mathbf{A}, \mathbf{S}') = \sum_{O \in Mineral} \begin{cases} 1, & O.\text{Collected}' \wedge \neg O.\text{Collected}, \\ 0, & \text{otherwise}. \end{cases} \tag{38}$$

### G.3.2 DEFEAT-ZERGLINGS-BANELINGS

The Defeat-Zerglings-Banelings (CMS) environment is also a StarCraftII mini-game. The game contains 9 marines (controlled by the player), 6 zerglings (hostile), and 4 banelings (hostile). The player needs to control the marines to deal as much damage as possible to the hostile zerglings and banelings. We illustrate the DZB environment in Figure 5(c).

We decompose the environment by 3 classes $\mathcal{C} = \{Marine, Zergling, Baneling\}$ such that

- $\mathcal{F}_s[C] = \{C.Position, C.Health, C.Alive\}$ for every $C \in \mathcal{C}$,
- $\mathcal{F}_a[Marine] = \{Marine.Move\}$.

where we set

- $Dom_{C.Position} = [-99, 99]^2$ for every $C \in \mathcal{C}$.
- $Dom_{C.Health} = [-1, 999]$ for every $C \in \mathcal{C}$.
- $Dom_{C.Alive} = \{\text{True}, \text{False}\}$ for every $C \in \mathcal{C}$.
- $Dom_{Marine.Move} = \{\text{North}, \text{East}, \text{South}, \text{West}, \text{NOOP}\}$.

We define the reward function as the total damage dealt to the hostile zerglings and banelings in each step:

$$R(\mathbf{S}, \mathbf{A}, \mathbf{S}') = \sum_{O \in Zergling} (O.\text{Health} - O.\text{Health}') + \sum_{O \in Zergling} (O.\text{Health} - O.\text{Health}'). \tag{39}$$

## H ADDITIONAL INFORMATION OF EXPERIMENTS

### H.1 EXPERIMENT SETTINGS

In all experiments, the dynamics models are trained using offline data that is collected by a random policy that produces uniform actions. However, it should be noted that data generated during the application of the OOCDM can also contribute to further training in practice (Ding et al., 2022).

All models are trained and evaluated using one GPU (NVIDIA TITAN XP). The only exception is the causal discovery for GRADER, which is implemented by an open-source toolkit called Fast CIT and computed on 4 CPUs in parallel in our experiments. All experiments were repeated 5 times using different random seeds; the means and standard variances of the performances are reported.

### H.2 IMPLEMENTATION OF BASELINES

**CDL** We perform causal discovery based on Theorem* 12 and the conditional independence tests are implemented using Conditional Mutual Information (CMI), which is estimated by the model. First, each variable in $(\mathbf{S}, \mathbf{A})$ is encoded into an encoding vector. Then, in the predictor of each state variable (attribute), the encoding vectors of parental variables are aggregated by an element-wise maximum operation. Finally, the aggregated encoding is mapped to the distribution of the next-state variable by an MLP.

**CDL-A** Most of the parts are identical to the original CDL. However, the encoding vectors of parental variables are aggregated by attention operation instead of max-pooling. Each input variable's encoding vector is transformed into a value and a key vector, and we learn a query vector for each output variable. Key-value attention is performed to obtain the aggregated encoding of the output variable (the attention weights of non-parental variables masked to 0).

**GRADER** We perform causal discovery based on Theorem* 12 and the conditional independent tests are implemented using Fast CIT (Chalupka et al., 2018). The model contains an individual predictor for each state variable (attribute), which aggregates all parents by a 2-directional Gated Recurrent Unit and then produces the distribution of the next-state variable.

**TICSA** This algorithm learns a probability matrix $M$ that stores the probability of each causal edge in the BCG. To infer the next state, the model first samples the causal graph from $M$. Then, it masks off non-parental input features and predicts next-state variables using an MLP architecture. To learn $M$, the loss function includes a sparsity penalty $\|M\|_1$, and the causal graphs are sampled using Gumbel softmax (Jang et al., 2017) during the training phase.

**MLP** In the MLP model, all input variables are concatenated into a vector. Then, we pass this vector into a 3-layer multi-layer perceptron and obtain an embedding vector $\boldsymbol{x}$. Finally, each variable (attribute) is decoded into the posterior distribution of $\hat{p}(O.S'|\boldsymbol{S}, \boldsymbol{A})$ by applying an individual transform on $\boldsymbol{x}$.

**GNN** The model architecture follows the design of Structural World Model (Kipf et al., 2020). We encode objects into *object state encodings* and *object action encodings* using individual encoders. Then, we transform these object encodings via a GNN based on a complete graph, where objects correspond to the nodes: 1) We compute the edge embeddings with the state encodings of each pair of objects; 2) we compute the node embedding for each object $O_i$, using its state encoding, its object action encoding, and all edge embeddings of in-degrees; 3) we decode the0 node embeddings of $O_i$ to the distributions of its next-state attributes.

**OOFULL** The model follows identical structure as described in Appendix E. However, the training loss only contains $\mathcal{L}_{\mathcal{G}_1}$ (See the paper's Eq. 7) and always uses the full OOCG $\mathcal{G}_1$ in evaluation.

To make our comparison fair, we do not want these baselines to perform badly in large-scale environments simply due to insufficient model capacity. Therefore, the number of hidden units in the non-object-oriented models (MLP, GRADER, and CDL) are adjusted according to the scale of the environment, making sure that the capacity of these models is pertinent to the complexity of the environments. However, our object-orient models (OOFULL and OOC) have a fixed number of parameters as long as the classes are fixed, no matter how many instances are in the environment.

## H.3 OUT-OF-DISTRIBUTION DATA

We construct o.o.d. data by changing the distribution of initial states of episodes, which is easy to implement in the Block and Mouse environments. However, the CMS and DZB environments are StarCraftII mini-games provided by the PySC2 platform (Vinyals et al., 2017). The platform offers limited access to the StarcraftII engine, and thus modifying the initialization process of episodes is very difficult. Therefore, we did not construct o.o.d. data for CMS and DZB.

**Block** To obtain the o.o.d. data, we initialize the attributes of each *Block* object from a new distribution at the beginning of each episode:

$$(O.S_1, O.S_2, O.S_3)^T \sim \mathcal{N}\left((0.5, 0, 0)^T, \text{diag}(0.25, 4, 4)\right). \tag{40}$$

**Mouse** In the i.d. data, the attributes from the field $Monster.Noise$ are initialized from a normal distribution $\mathcal{N}(0, 1)$ at the beginning of each episode. To construct the o.o.d. data, we increase the standard variance to 3. To confuse non-causal models, the initialization of $O_{food}.Amount$ is correlated to $O_{food}.Position$. During training, $O_{food}.Amount$ will be assigned with a larger value if $O_{food}.Position$ is in the east of the world; in the o.o.d. data, however, $O_{food}.Amount$ will be assigned with a larger value if $O_{food}.Position$ is in the north.

## H.4 UNSEEN TASKS

In the Mouse environment, we sample the numbers of food, monsters, and traps respectively from $[3, 6]$, $[1, 5]$, and $[1, 5]$. Thereby, we obtain a task pool containing $4 \times 5 \times 5 = 100$ tasks. We randomly split these tasks into 47 seen tasks and 53 unseen tasks. The dynamics models are trained

Table 9: The total sizes of model parameters. $n$ denotes the number of variables in the environment, and $m$ denotes the number of all fields. The model sizes of OOFULL are the same as those of OOCDM.

|  | $n$ | $m$ | GRADER | CDL | CDL-A | TICSA | GNN | MLP | **OOCDM** |
|---|---|---|---|---|---|---|---|---|---|
| Block-2 | 12 | 8 | 2.3MB | 319.7KB | 348.6KB | 184.8KB | 66.4KB | 99.4KB | 401.6KB |
| Block-5 | 24 | 8 | 6.0MB | 1.0MB | 1.1MB | 582.3KB | 100.0KB | 224.4KB | 401.6KB |
| Block-10 | 44 | 8 | 15.7MB | 3.1MB | 3.2MB | 2.0MB | 147.9KB | 538.0KB | 401.6KB |
| Mouse | 28 | 10 | 15.1 MB | 2.7MB | 3.0MB | 720.4KB | 432.3KB | 701.4KB | 546.6KB |
| CMS | 44 | 4 | 23.5MB | 4.3MB | 4.6MB | 1.4MB | 250.7KB | 1.1MB | 140.3KB |
| DZB | 66 | 10 | 38.8MB | 7.7MB | 8.2MB | 2.6MB | 616.3KB | 1.7MB | 549.8KB |

|  |  | Values for environments | | | |
|---|---|---|---|---|---|
| Symbol | Meaning | Block | Mouse | CMS | DZB |
| $d_e$ | The dimension of attribution-encoding vectors | 16 | 16 | 16 | 16 |
| $d_k$ | The dimension of key vectors | 32 | 32 | 32 | 32 |
| $d_v$ | The dimension of value vectors | 32 | 32 | 32 | 32 |
| $\varepsilon$ | The threshold of CMIs in causal discovery | 0.3 | 0.1 | 0.2 | 0.03 |
| $n_{plan}$ | The number of planning iteration in CEM | - | 5 | 5 | 5 |
| $H$ | The planning horizon in MPC | - | 20 | 20 | 20 |
| $\alpha$ | The weight of $\mathcal{L}_{\mathcal{G}_1}$ in the target function | 1 | 1 | 1 | 1 |
| $\beta$ | The weight of $\mathcal{L}_{\hat{\mathcal{G}}}$ in the target function | 1 | 1 | 1 | 1 |
| $\gamma$ | The discount of rewards in MPC | - | 0.95 | 0.95 | 0.95 |
| $\lambda$ | The probability of each OOC-expression when sampling $\mathcal{G}_\lambda$ | | | | |
|  | The number of samples in CEM | - | 500 | 500 | 500 |
|  | The number of elite samples in CEM | - | 100 | 100 | 100 |
| $n_{iter}$ | The number of iteration in training | 50 | 200 | 80 | 200 |
| $n_{batch}$ | The number of batches in each iteration | 1000 | 1000 | 1000 | 1000 |

Table 10: The main hyper-parameters used in our experiments.

using offline data collected in seen tasks and then transferred into unseen tasks without further training.

## H.5 COMPUTATIONAL COSTS

We provide additional results about the model size (see Table 9), and the computation time of causal discovery is shown in Table 2. These results show that our OOCDM greatly reduces the model complexity in large-scale environments. We stress that these results are for reference only, as they are affected by many factors, including the implementation details, software (we use Python and PyTorch here), and computation devices. We are also aware that some of the comparison made here is not perfectly fair, as these CDMs perform causal discovery using different devices. However, Our approach shows the advantages of several orders of magnitude compared to GRADER, reducing the computation time from multiple hours to several seconds. Such a huge gap in these results cannot be caused solely by the differences in devices. Combining the results in the paper's Table 1, we conclude that our OOCDM uses relatively fewer parameters and the least computation time to discover the most accurate causal graphs.

## H.6 LEARNED OOCGS OF STARCRAFTII MINI-GAMES

Since the ground-truth OOCG of the CMS and DZB environments are not known, we here present the OOCGs learned by our causal discovery algorithm in Figure 8. The learned OOCGs for CMS are identical for all seeds. However, The learned OOCGs for DZB are slightly different between seeds, and thus the OOCG of the seed that produces the highest likelihood is presented.

## H.7 HYPER-PARAMETERS

Main hyper-parameters are listed in Table 10, and more details are contained in our code.

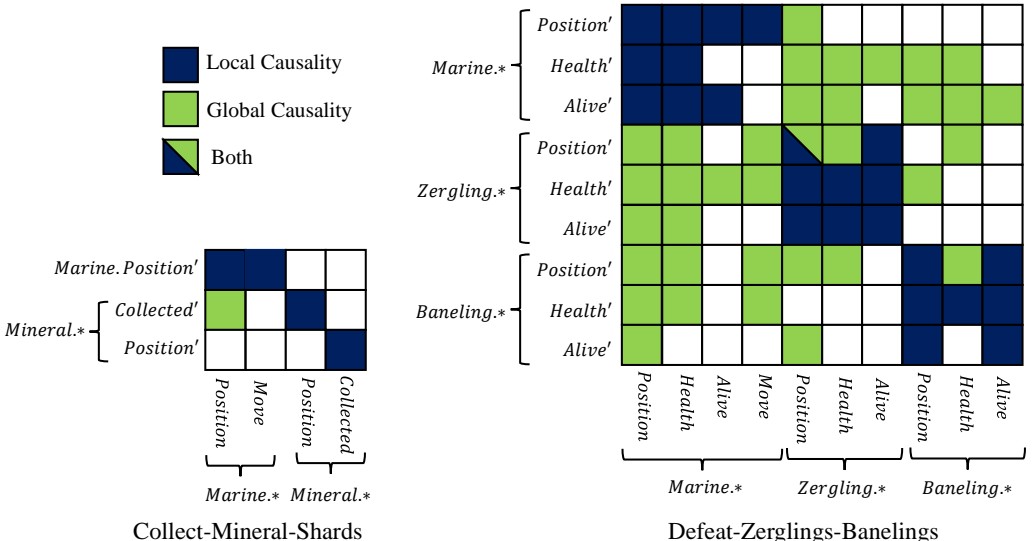

Figure 8: The visualization of the discovered OOCGs (the adjacency matrix of class-level causalities) of the CMS and DZB environments.

# I HANDLING ASYMMETRIC ENVIRONMENTS WITH OOCDMS

## I.1 ENSURING THE RESULT AND CAUSATION SYMMETRY OF OOMDPS

Result symmetry (Eq. 2) and causation symmetry (Eq. 3) may be too strong to hold true in some cases. In an *asymmetric environment* (where one of the symmetries does not hold), the ground-truth causal graph may not be an OOCG, and some objects might possess their unique causal connections and dynamics, which may greatly compromise the performance of OOCDMs that strictly comply with these symmetries.

However, the dynamics cannot be modeled symmetrically typically because the attributes of objects provide insufficient information to do so. The following theorem indicates that both result symmetry and causation symmetry can be guaranteed by adding additional state attributes for the objects.

**Theorem\* 19.** *Assume $\mathcal{M}$ is an OOMDP where the classes are $\{C_1, \cdots, C_K\}$, where the result symmetry (Eq. 2) and causation symmetry (Eq. 2) may be violated. There always exist an extended OOMDP $\widetilde{\mathcal{M}}$ such that the following statements hold ($\widetilde{*}$ denotes the corresponding item in $\widetilde{\mathcal{M}}$):*

1. *$\mathcal{F}_s[C_k] \subseteq \mathcal{F}_s[\widetilde{C}_k]$ and $\mathcal{F}_a[C_k] = \mathcal{F}_a[\widetilde{C}_k]$ for $k = 1, \cdots, K$.*
2. *There exist $\Phi$ that maps $(\widetilde{\mathbf{S}}, \widetilde{\mathbf{A}})$ into $(\mathbf{S}, \mathbf{A})$ and $\Psi$ that maps $\widetilde{\mathbf{S}}'$ into $\mathbf{S}'$.*
3. *$\widetilde{\mathcal{M}}$ mirrors the dynamics of $\mathcal{M}$. In other words, $\tilde{p}(\widetilde{\mathbf{S}}'|\widetilde{\mathbf{S}}, \widetilde{\mathbf{A}}) = p(\Psi(\widetilde{\mathbf{S}}')|\Phi(\widetilde{\mathbf{S}}, \widetilde{\mathbf{A}}))$.*
4. *Result symmetry and causation symmetry both hold in $\widetilde{\mathcal{M}}$.*

*Proof.* Let $N_k$ denote the number of instances of $C_k$. Then, we define $\mathcal{F}_s[\widetilde{C}_k] = \mathcal{F}_s[C_k] \cup \{C_k.Id\}$, where $Dom_{C_k.Id} = \{1, 2, \cdots, N_k\}$. The distribution of the start state in $\widetilde{\mathcal{M}}$ ensures that each instance $\widetilde{O}$ has a unique $\widetilde{O}.Id$ among all instances of $\widetilde{C}_k$.

We define $\phi(k, c)$ as the function that finds the index $i \in \{1, 2, \cdots, N\}$ such that $\widetilde{O}_{\phi(k,c)}$ is an instance of $\widetilde{C}_k$ and $\widetilde{O}_{\phi(k,c)}.Id = c$. In other words, $\phi(k, c)$ outputs the overall index of the $c$-th instance of $\widetilde{C}_k$ in $\widetilde{\mathcal{M}}$. Meanwhile, we use $RemoveId(\cdot)$ to remove all variables like $\widetilde{O}.Id$ in the given variables.

In $\mathcal{M}$, we assume $O_i$ is the $c_i$-th instance of class $C_{k_i}$. Then we may define $\Phi$ and $\Psi$ as:

$$(\mathbf{S}, \mathbf{A}) = \Phi(\widetilde{\mathbf{S}}, \widetilde{\mathbf{A}}) \coloneqq \big(RemoveId(\widetilde{\mathbf{O}}_{\phi(k_1, c_1)}), \cdots, RemoveId(\widetilde{\mathbf{O}}_{\phi(k_N, c_N)})\big)$$

$$\mathbf{S}' = \Psi(\widetilde{\mathbf{S}}') \coloneqq \big(RemoveId(\widetilde{O}_{\phi(k_1, c_1)}.\mathbf{S}'), \cdots, RemoveId(\widetilde{O}_{\phi(k_N, c_N)}.\mathbf{S}')\big)$$

Then we directly define the dynamics of $\widetilde{\mathcal{M}}$ by

$$\widetilde{O}_i.\text{Id}' \equiv \widetilde{O}_i.\text{Id}, \qquad i = 1, \cdots, N.$$

$$\tilde{p}\left(RemoveId(\widetilde{\mathbf{S}}')|\widetilde{\mathbf{S}}, \widetilde{\mathbf{A}}\right) \coloneqq p(\Psi(\widetilde{\mathbf{S}}')|\Phi(\widetilde{\mathbf{S}}, \widetilde{\mathbf{A}})),$$

Then we will have $\tilde{p}(\widetilde{\mathbf{S}}'|\widetilde{\mathbf{S}}, \widetilde{\mathbf{A}}) \coloneqq p(\Psi(\widetilde{\mathbf{S}}')|\Phi(\widetilde{\mathbf{S}}, \widetilde{\mathbf{A}}))$.

Therefore, assuming $\widetilde{O}_i \in \widetilde{C}_k$, we have

$$\tilde{p}(\widetilde{O}_i.\mathbf{S}'|\widetilde{\mathbf{S}}, \widetilde{\mathbf{A}}) = \tilde{p}_{\widetilde{C}_k}(\widetilde{O}_i.\mathbf{S}'|\widetilde{\mathbf{S}}, \widetilde{\mathbf{A}}) \coloneqq p\left(O_{\phi(k, \tilde{O}.\text{Id})}.\mathbf{S}'|\Phi(\widetilde{\mathbf{S}}, \widetilde{\mathbf{A}})\right).$$

Therefore, $\widetilde{\mathcal{M}}$ satisfies the result symmetry.

Moreover, it is easy to prove that $\widetilde{\mathcal{M}}$ satisfies the causation symmetry. Assuming $\widetilde{O}_x, \widetilde{O}_y \in \widetilde{C}_l$, swapping their attributes (which include $\widetilde{O}_x.\text{Id}$ and $\widetilde{O}_x.\text{Id}$) does not affect $\phi(l, c)$ for any $c \in \{1, \cdots, N_l\}$. Therefore, it does not affect the result of $\Phi(\widetilde{\mathbf{S}}, \widetilde{\mathbf{A}})$ and $\Psi(\widetilde{\mathbf{S}})$. Eventually, swapping the attributes of $\widetilde{O}_x$ and $\widetilde{O}_y \in C_l$ has no influence on $\tilde{p}(\widetilde{O}_i.\mathbf{S}'|\widetilde{\mathbf{S}}, \widetilde{\mathbf{A}})$. $\qquad\square$

The proof provides an easy way to ensure the result and causation symmetries – the OOMDP can simply include an *identity attribute* $O.\text{Id}$ which gives the unique index of the object among all instances of its class. This is always plausible since no additional feature must be observed. Meanwhile, these identity attributes are fixed throughout an episode, and thus we do not need to learn their predictors in the implementation of OOCDM.

## I.2 OOCDM FOR ASYMMETRIC ENVIRONMENTS

Appendix I.1 introduces a way to ensure the result and causation symmetries by adding additional attributes about the identity of objects. To extend OOCDM to asymmetric environments, we implement built-in identity attributes by slightly augmenting predictors in Appendix E.1. The augmented predictors are able to handle the asymmetric dynamics without explicitly modifying the representation of the OOMDP. The new architecture is illustrated in Figure 9. Each class contains an identity encoder $IdEnc_{C_k}$ that maps the indices of instances to identity-encoding vectors, which are integrated into the object-encoding vectors. Assuming $O_i$ is the $c$-th instance of $C_k$, then

$$\boldsymbol{x}_i = Concat\left([O_i.\boldsymbol{u}]_{O_j.\text{V}'} \text{ for } C_k.U \in \mathcal{F}[C_k]; \; IdEnc_{C_k}(c)\right).$$

The other parts are the same as the architecture in Appendix I.1. Since the identity attributes are used implicitly, we do not consider them into causal discovery. Therefore, in asymmetric environments, object-oriented causal discovery will identify the minimal OOCG that represents the transition dynamics, instead of the ground-truth causal graph.

However, by masking the identity-encoding vectors and by computing CMIs, we may also identify the causal descendants of the identity attributes. Theoretically, the ground-truth causal graph of the original OOMDP can be obtained by the following procedure:

1. Identify the class-level causal descendants of identity attributes using class-level CMIs.
2. For every state field $C_k.V$, if there is $C_l$ such that $C_l.Id \to C_k.V'$, examine the parents of $O_j.\text{V}'$ within class $C_l$ using non-object-oriented CMIs, i.e. the CDL method. Otherwise, if $C_l.Id \nrightarrow C_k.V'$, examine the parents of $O_j.\text{V}'$ within class $C_l$ using object-oriented CMIs.

## I.3 THE EXPERIMENT ON THE ASYMMETRIC BLOCK ENVIRONMENT

We performed an additional experiment to test the performance of OOCDM using the modified predictors in Appendix I.2. We designed the **AsymBlock** Environment with similar dynamics to the Block environment, yet result and causation symmetry are violated in AsymBlock. In AsymBlock$_k$, there will be $k$ instances of $Block$ and also $k$ instances of $Total$. For the $c$-th instance of $Total$, we have

$$O_c.\text{S}'_j = \frac{1}{2}O_c.\text{S}_j + \frac{1}{2}\max_{O_i \in Block, i \le c} O_i.\text{S}_j, \qquad j = 1, 2, 3; \; c = 1, \cdots, k.$$

The results of prediction accuracy and the computation time of causal discovery are shown in Table 11. In particular, we use the "-asym" suffix for OOFULL and OOCDM to signify that the

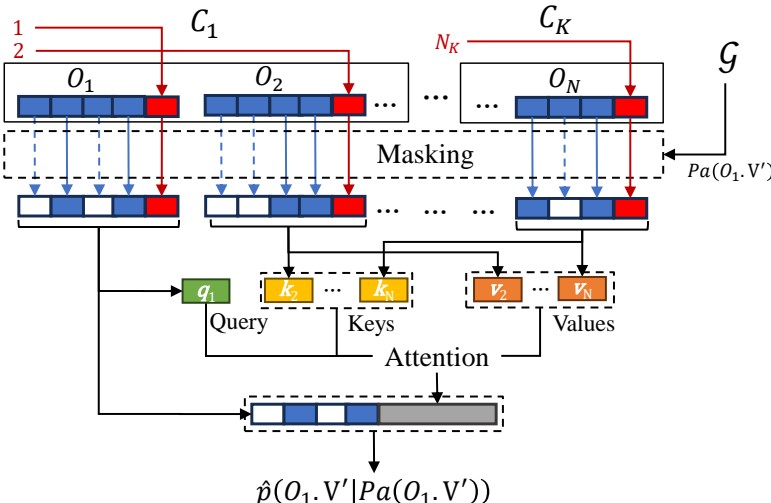

Figure 9: The illustration of $f_{C_1.V}(O_1.V'|\mathbf{S}, \mathbf{A}; \mathcal{G})$ with built-in identity attributes. $N_i$ denotes the number of instance in $C_i$ for $i = 1, \cdots, K$.

Table 11: The average instance log-likelihoods and causal discovery times (in seconds) on the AsymBlock environment, where the numbers of $Block$ and $Total$ are both 5. The "-asym" suffix for OOFULL and OOCDM means that the predictors are modified according to Appendix I.2 to handle the asymmetric dynamics. We do not show the standard variances for obviously over-fitting results (log-likelihoods less than $-100.0$, highlighted in brown).

|  | GRADER | CDL | CDL-A | TICSA | GNN |
|---|---|---|---|---|---|
| AILL train | $\mathbf{22.6}_{\pm 1.3}$ | $16.4_{\pm 1.4}$ | $22.0_{\pm 0.7}$ | $20.8_{\pm 0.7}$ | $16.1_{\pm 0.41}$ |
| AILL i.d. | $14.7_{\pm 1.7}$ | $-7.9_{\pm 12.2}$ | $-586.8$ | $16.7_{\pm 2.8}$ | $15.3_{\pm 0.42}$ |
| AILL o.o.d. | $-25.6_{\pm 12.4}$ | $-130.1$ | $-4.2e4$ | $-25.1_{\pm 34.5}$ | $-165.5$ |
| causal discovery time | $4789.6_{\pm 177.4}$ | $19.1_{\pm 5.7}$ | $19.1_{\pm 2.8}$ | - | - |
|  | MLP | OOFULL | OOCDM | OOFULL-asym | OOCDM-asym |
| AILL train | $10.8_{\pm 1.5}$ | $18.7_{\pm 0.3}$ | $16.4_{\pm 2.5}$ | $21.8_{\pm 1.0}$ | $22.0_{\pm 1.8}$ |
| AILL i.d. | $2.6_{\pm 5.8}$ | $15.7_{\pm 0.5}$ | $15.2_{\pm 2.4}$ | $2.2_{\pm 1.0}$ | $\mathbf{21.9}_{\pm 1.7}$ |
| AILL o.o.d. | $-170.1$ | $-14.2_{\pm 23.9}$ | $-34.0_{\pm 18.0}$ | $-700.6$ | $\mathbf{8.3}_{\pm 8.5}$ |
| causal discovery time | - | - | $\mathbf{2.1}_{\pm 0.1}$ | - | $2.3_{\pm 0.0}$ |

predictors are modified according to Appendix I.2 to handle the asymmetric dynamics. According to these results, the built-in identity attributes allow OOCDM-asym and OOFULL-asym to infer the unique dynamics of each object, leading to significant improvement against the symmetric version. Since AsymBlock does not satisfy the result and causation symmetry, the ground-truth causal graph is not an OOCG. Therefore, we do not compare the accuracy of causal graphs. However, we observe that the OOCDM with modified predictors successfully identified the minimal OOCG that represents the dynamics. As a result, OOCDM-asym shows better generalization ability than OOFULL-asym. These results demonstrate that through simple modification OOCDM-asym may handle environments where result and causation symmetries do not strictly hold. We also notice that the modification does not lead to a significant increase in the computation costs of causal discovery.

## J  WEAKNESSES AND FUTURE WORKS

A weakness of this work is the requirement of domain knowledge to formulate environments as OOMDPs. Although using objects and classes to describe the world is natural, intuitive formulation may violate result symmetry and causation symmetry. As mentioned in Section 2.2, many studies have investigated the learning of object-centric representation. However, extracting OOP-style representation (i.e. involving multiple classes) remains an open problem, especially when Eq. 3 needs to be satisfied. Therefore, future work will investigate how to extract properly-categorized objects

from raw observations. Meanwhile, more effective methods to release result and causation symmetries should be further explored, where modeling relational interactions from raw factorization may be a potential direction.

Another weakness of this work is that FMDP imposes strong constraints that may not hold in more complicated tasks involving confounders, partial observability, or non-Markovian dynamics. Addressing these challenges in an object-oriented framework is important to extend the applicability of our approach. Therefore, we propose to explore these directions using more rigorous tools of causality in the future.

In addition, we expect to explore more sophisticated object-oriented representations in the future. Class inheritance may be a promising direction. For example, fields, models, and causality of some classes (e.g., "Dog" and "Cat") can be derived from those of a base class (e.g., "Animal"). Such class inheritance could provide considerable generalization opportunities for object-oriented environments, leading to class inheritance and model inheritance.

