# OpenReview forum: "Learning Causal Dynamics Models in Object-Oriented Environments"
_ICLR.cc/2024/Conference — Submitted to ICLR 2024_

### Official Review · Reviewer_miEU · 2023-10-29

**Soundness:** 3 good
**Presentation:** 1 poor
**Contribution:** 2 fair
**Rating:** 6
**Confidence:** 4

**Summary:**

The paper introduces an object-oriented causal dynamics model (OOCDM) that leverages the hierarchical structure among state and action variables. Specifically, state and action variables are decomposed into a number of object variables, and the objects that belong to the same class share the same causality and transition function. Assuming that these object structures are given as prior knowledge, OOCDM learns the forward dynamics (i.e., transition function) and causal relationships between objects and fields. As OOCDM leverages the known hierarchical structure (i.e., objects and classes), it is more computationally efficient, and robust compared to previous CDMs.

**Strengths:**

- The paper tackles an important problem and the motivation is clear. It is known that causal dynamics models (CDMs) are more robust compared to traditional dense dynamics models and it is important to improve the scalability of CDMs for their wider applicability.
- The formulation of OOCDM which extends previous CDMs to object-oriented MDP representations seems novel, as far as I know.
- The authors provide experimental results on larger-scale environments compared to prior works.

**Weaknesses:**

First, the manuscript is hard to follow due to the heavy notations and too much jargon, which could be more simplified. Second, the assumptions need more explanation. The *result symmetry* (eq. 3) is understandable, but the *causation symmetry* (eq. 4) does not seem to always hold. It would be better if the authors provide scenarios where this assumption holds and does not hold, and explain why it is reasonable to make the assumption.

Finally, I have a concern regarding the experiments, e.g., the evaluation and interpretation of the results, which is also related to the main claim of the paper. Specifically, the authors claim that OOCDM outperforms prior CDMs in terms of (a) computational efficiency, (b) causal discovery, and (c) generalization of model-based RL. For (a), which is shown in Table 2, it is reasonable but also somewhat obvious since OOCDM leverages the hierarchical structure of the variables and common causal relationships shared with different objects. My major concern is at (b) and (c), which I elaborate on below.

- Evaluation of causal discovery is not thorough. The performance of causal discovery is typically measured as structural hamming distance (SHD), or sometimes AUC and F1 score. The term “accuracy” in table 1 is confusing and details on how it is evaluated are missing.
- OOFULL, which is the ablation of OOCDM without causal discovery, uses fully-connected bipartite graph. In other words, it only leverages the (A) *structural information* (objects belonging to the same class share the same dynamics), without the (B) *causal relationship* between variables. As shown in Table 3-4, OOFULL outperforms other CDMs. Given that CDMs work fairly well in causal discovery (as shown in Table 1), this implies that **(A) structural information (how variables are grouped with objects and classes) is the major contribution** to the performance gain of OOCDM. This poses additional questions, e.g., why and how does OOFULL (which is non-causal) generalize better than CDMs? It seems the assumptions are too strong (i.e., more important than causality). Also, how does the accurate understanding of causality lead to the improvement of OOCDM compared to OOFULL?
- CDMs work fairly well in causal discovery (Table 1), but they fail on o.o.d data, as shown in Table 3. This is contrary to the findings of prior works (CDL, GRADER) and requires explanations.
- In the evaluation of model-based RL (Table 4), the dynamics models are trained using only offline data. However, prior work (GRADER) suggests that it achieves better performance (in terms of both causal discovery and RL) when using online data. I would appreciate it if the authors could provide a justification for this.

**Questions:**

- Attention-based architecture is used to handle varying numbers of objects. In the experiments, does the number of objects vary in the same trajectory? If yes, then how do baselines handle varying input dimensions?
- How is accuracy in Table 1 evaluated?
- Table 1 shows that GRADER performs worse in causal discovery compared to other methods. This is counter-intuitive given that GRADER uses explicit conditional independence tests, unlike others.

**Minor (mostly about notations and typos)**

- Why does it need two separate notations in Def. 2? For example, $C_k\left[U \rightarrow V^{\prime}\right]$ can be written as $C_k\left[C_k \cdot U \rightarrow V^{\prime}\right]$. It would be much easier to follow the paper with simple and consistent notations (maybe just write as $C_l \cdot U \rightarrow C_k \cdot V'$?)
- In eq. 4, $O_p, O_q$ should be $O_x, O_y$
- The notation $f(\cdot\mid \mathbf{O}; \mathbf{U}_{-O}; \mathcal{G})$ in Eq. 5 (as well as Eq. 8-10) is confusing. Isn’t it basically the same as $f(\cdot\mid \mathbf{U}; \mathcal{G})$? Also, notations in Eq. 8-10 are not consistent with Eq. 5.
- In Sec 4.3, it says “$\mathcal{G}_1$ is the full OOCG”. To my understanding, $\mathcal{G}_1$ is a fully-connected bipartite graph and is not OOCG.
- Table 7, description of $O.\mathbf{S}$: action → state
- Ground-truth causal graph provided in the appendix is hard to understand (the actual graph or adjacency matrix would be better).

---

> ### Author Response · Authors · 2023-11-12
>
> Thank you for your valuable comments and opinions.
>
> # Weaknesses
>
> We understand that the notation is a bit heavy, as it is hard to combine the notations of causal inference and object-oriented representation. We will try to improve the clarity of the notations in the revised version.
>
> If we apply identification encoding (each object has an attribute $O.\textrm{ID}$), just like position encoding for Transformer, then Causation Symmetry usually holds.
>
> It's true that OOFULL outperforms CDMs. There are 2 main factors that lead to overfitting:
> 1. Making predictions from **spurious correlations**. CDMs are able to ameliorate this factor by learning the causal structure.
> 2. Even with oracle causal discovery, overfitting can still occur within the structural equations because they are still learned from a **limited amount of data representing the entire space**. CDMs have no capacity to solve this problem, which **explains why CDMs fail on o.o.d. data**. For example, CDL-A performs better than CDL at causal discovery, but generalizes worse than CDL. The reason is that the attention in CDL-A is much stronger than the pooling mechanism in CDL, leading to more overfitting within the structural equations.
>
> One way to reduce the overfitting from the second reason is to increase the number of samples. OOCDM and OOFULL are able to do this because each instance of the class $C$ is treated as a sample for learning the predictor $f_{C.V}$. Thus, the sample size increases significantly. Another reason for OOFULL to generalize better is that causation symmetry limits the VC dimension of the model, which reduces the chance of overfitting.
>
> Based on the above analysis, the generalization ability of OOCDM benefits from 3 sources: causal structure, causation symmetry, and increased sample size. OOFULL benefits from causation symmetry and increased sample size, but not from causal structure. CDMs, on the other hand, only benefits from causal structures. As a result, we observe that the performance is OOCDM > OOFULL > CDMs.
>
> GRADER indeed suggests using online learning to improve its performance. In this paper, we do not specify how OOCDM should be used, and online learning is absolutely acceptable. We experimented with offline data because it is the best way to demonstrate the generalization performance of the model. If OOCDM can handle this challenging case, we have no doubt that it can perform better when online data is available. Moreover, GRADER's theory requires that we repeatedly perform causal discovery with new data. Considering that the causal discovery of GRADER can take hours, we doubt that it is practical to make comparisons with CDM baselines.
>
> # Questions
>
> The number of objects does not vary along the same trajectory. Otherwise, we may need to predict when a new object enters the environment and when an existing object leaves the environment (we leave this to future work). However, the number may vary between episodes. For comparison with baselines in Sections 5.2, 5.3, and 5.4, we fixed the number of objects so that baselines can be applied, and the performance of different numbers of objects are reported separately (Block2, Block5, and Block10). In Section 5.5, when we tested the performance of OO models under different number of objects, we only compared with OOFULL, because other baselines do not apply.
>
> The causal graph accuracy is measured by SHD. We will include this in the main paper.
>
> GRADER employs Fast CIT (FCIT), for which we use the official FCIT package from github. To test whether $X$ and $Y$ are independent conditional on $Z$, the idea of FCIT is to train 2 decision trees -- one predicting $Y$ using only $Z$, and the other predicting $Y$ using both $X$ and $Z$. Then a t-test is performed based on the mean square errors of the two decision trees. To ensure that the test is "fast", the time to train these trees is quite limited, so the precision of the trees is worse than the neural networks used in CDL and OOCDM. It is the low precision of these decision trees that compromises the performance of causal discovery in GRADER.
>
> # About minor problems
>
> Sorry for the typos. We will correct them in the revision.
>
> The expression $C_k[C_k.U \rightarrow V]$ is different from $C_k[U \rightarrow V]$. Assume that $C_k$ is $product$, where $U$ is $price$, and $V$ is $sell$. Then, $Product[Price \rightarrow Sell]$ means that the sell of each product is affected by its own price. $Product[Product.Price \rightarrow Sell]$ means that the sell of each product is influenced by the price of all other products (which is true in a competitive market). Note that $k = l$ is not forbidden in Definition 2. That's why we have to distinguish between notations for local and global casalities. Maybe $C.(U \rightarrow V)$ and $C_k.U \rightarrow C_l.V$ is better here?
>
> We will include adjacency matrices in the appendix.
>
> A fully-connected bipartite graph satisfies the definition of an OOCG, where all causality expressions hold.

---

> > ### Comment · Reviewer_miEU · 2023-11-20
> >
> > Thanks for the thorough response. Most of my concerns are addressed. However, I am still confused about exactly what "The causal graph accuracy is measured by SHD" means. Can you elaborate on exactly how accuracy in Table 1 is evaluated?

---

> ### Author Response · Authors · 2023-11-17
> **We have submitted the revised version.**
>
> Dear reviewer, we have submitted the revised version of our paper. These changes in the revision may be related to your concerns.
>
> 1. In some cases the bold letters (e.g. $\mathbf{S}$) are not distinguishable from the normal letters (e.g. $\mathrm{S}$). Therefore, now a normal letter always carries a subscript (e.g., $\mathrm{A}_i, \mathrm{S}_i$, $\mathrm{S}_j$) so they can be well distinguished.
> 2. The notation $\mathbf{U}$ is no longer used. Instead we will use $(\mathbf{S}, \mathbf{A})$ directly. This way we avoid introducing unfamiliar notations, and $\mathbf{U}$ will not be confused with the letter used in a field $C.U$ or an attribute $O.\mathrm{U}$.
> 3. We have changed the notation of local class-level causality to $C.U \rightarrow V$, and that of glocal class-level causality to $C_l.U \rightarrow C_k.V$. These notations are simpler and more intuitive.
> 4. We simplified equations (5-10) in the old version, and now there are only two equations (5-6) in the revised version to express the same meaning. This reduces the reader's workload and the complexity of the notation.
> 5. The subscripts of objects in Eq. 4 (now Eq. 3) are now corrected.
> 6. in Table 7, "action" is corrected to "state" for the meaning of $O.\mathbf{S}$.
> 7. We now write a field predictor using the form $f_{C.V}(O.\mathrm{V}'|\mathbf{S},\mathbf{A};\mathcal{G})$.
> 8. In Section 5.2, we have explained that the accuracy of causal graphs is measured by Structural Hamming Distance.
> 9. We have visualized the adjacency matrices of the ground truth OOCGs in Appendix G and the discovered OOCGs in Appendix H. These figures are intended to be more direct and understandable than the old presentation of OOCGs.
> 10. In Appendix I, we have provided a simple solution (**with an additional experiment**) to **extend OOCDM to environments where causation symmetry does not hold**. In short, by adding auxiliary identity attributes to the objects, we can ensure causation symmetry and result symmetry in all OOMDPs.

---

> ### Author Response · Authors · 2023-11-20
>
> Thank you for your response.
>
> We limited the computation of accuracy within the edges that a bipartite CDM needs to identify. After causal discovery (both OOCDM and CDM), we represent the causal graph by the adjacency matrix from $(\mathbf{S}, \mathbf{A})$ to $\mathrm{S}_j'$. Then we measure the differences between the true and estimated matrices.
>
> Suppose $\hat{M}_{n_s \times (n_s + n_a)}$ is the estimated adjacency matrix.
>
> Assume that $M_{n_s \times (n_s + n_a)}$ is the true adjacency matrix.
>
> Here, $M_{ji} = 1$ means that the $i$-th variable in $(\mathbf{S}, \mathbf{A})$ is the parent of $\mathrm{S}_j'$.
>
> We count the number of different elements in the two matrices. That is, if $\hat{M}_{ji} \neq {M}_{ji}$, an error is counted. Therefore, each missing edge or redundant edge is considered as an error. Then we express the accuracy as a percentage:
>
> $$ \frac{n_{correct}}{n_s \times (n_s + n_a)} \times 100 $$.
>
> We hope this addresses your doubt, and we will be grateful if the score can be adjusted accordingly.

---

> > ### Comment · Reviewer_miEU · 2023-11-20
> >
> > Thank you for the clarification. Accordingly, I am raising my score from 5 to 6.

---

### Official Review · Reviewer_dZfY · 2023-10-30

**Soundness:** 3 good
**Presentation:** 2 fair
**Contribution:** 3 good
**Rating:** 6
**Confidence:** 2

**Summary:**

The authors study the combination of causality and reinforcement learning by providing an extension to the setup of causal dynamics models through the lens of object oriented programming, which essentially provides a sharing of information based systems where the causal dynamics are shared across objects that belong to the same class. Further, the authors provide a learning mechanism based on key-value attention which enables generalization to arbitrary numbers of objects as long as the underlying classes remain the same. The authors’ experiments highlight that such an object oriented causal dynamics model (OOCDM) outperforms existing approaches along various fronts like causal discovery, performance accuracy and generalization capacity.

**Strengths:**

- The authors are tackling a quite interesting and relevant problem and the approach used is well motivated and cognitively inspired. While it is studied for causality and RL approaches, it has been sparsely leveraged when combining the two fields in trying to model the causal dynamics of the environment itself and hence makes this work a very interesting and worthwhile read.
- It is a natural extension to CDL which also further reduces the computations required owing to sharing of information between objects of the same class.
- The authors test out on a variety of domains ranging from causal discovery and prediction accuracy as well as combining it with planning and highlight that the proposed approach is competitive and lucrative.

**Weaknesses:**

- While the work is quite interesting and very relevant to the field, the writing itself could use some work. The notation is not clearly specified over the paper and the distinction between bold and normal capital letters is hard to understand (eg. for U). For example, in Section 3.1, what does the random variable S represent? What happens if the transitions are time-depdendent?
- The difference between $\mathcal{F}$ and $\mathcal{F}_s$ is not explained at all, and just directly used in Definition 2.
- In Section 4.2, when the authors propose a model for implementing the prediction function, while the approach satisfies causation symmetry, it does not treat the dynamics induced by different kinds of classes in the same manner as opposed to differently. For example, the attributes $C_1.U_1$ should impact the predictor differently from the attributes $C_2.U_2$, while the key-value attention mechanism would imply using the same mechanism. Could the authors clarify this?
- The authors should also take some space to explain the fundamentals of CIT / CMI methodology, and can then point to the Appendix for further clarity. However, without some background on CIT / CMI as well as how gradient based learning can be used to learn the graph structure (Equation 12), it is hard to follow the paper since graphs are discrete objects and gradient ascent based procedures only work on continuous spaces.

For the most part, my main qualms about the work are regarding the clarity and completeness of writing in the main draft. While the Appendix does contain a lot of details, I felt that there was some key information missing in the main draft and the preliminary section which makes the paper fairly dense and hard to read. There are also a few related works that the authors are missing, which I will mention below. I think clarifications on the math and making the notation and related work clearer in the main draft would go a long way, and I would be happy to increase my score if the changes were made.

*Mittal, S., Bengio, Y., & Lajoie, G. (2022). Is a modular architecture enough?. Advances in Neural Information Processing Systems, 35, 28747-28760.*

*Locatello, F., Weissenborn, D., Unterthiner, T., Mahendran, A., Heigold, G., Uszkoreit, J., ... & Kipf, T. (2020). Object-centric learning with slot attention. Advances in Neural Information Processing Systems, 33, 11525-11538.*

*Mittal, S., Lamb, A., Goyal, A., Voleti, V., Shanahan, M., Lajoie, G., ... & Bengio, Y. (2020, November). Learning to combine top-down and bottom-up signals in recurrent neural networks with attention over modules. In International Conference on Machine Learning (pp. 6972-6986). PMLR.*

**Questions:**

- After equation 1, the authors state that they assume the underlying graph is unknown. What about the probability distribution, is that known or also unknown?
- What does it mean that the state variables transit independently? It would be nice if the authors could write it down in math, as it is not immediately clear for people who do not work on RL or the FMDP setup. I understand it is in the appendix but it would be nice to spell it once in the main text.
- In equation 2, does the symbol used represent conditional independence, or the negative of that because it is not clear from the statement under the equation.
- How does the theorem imply that testing every edge requires O(n) complexity?
- In Equation 3, why does the distribution not depend on $\mathbf{O}_{i+2}$ and so on? Further, what is the benefit of using semicolon in describing the conditioning, and why is $O_i.(\mathbf{S}’)$ dependent on $\mathbf{O}_i$ and not $\mathbf{O}_i.\mathbf{S}$?

---

> ### Author Response · Authors · 2023-11-12
>
> Thank you for your comments.
>
> # About weaknesses
>
> First of all, we understand your concern about the readability of the paper. We will try our best to improve this in the revised version.
>
> ## Weakness 1
>
> Regarding notations, you can find a list of all notations in Appendix A. Combining the notations for object-oriented representation and causal inference was more complicated than we thought, so there may be some confusion. We use recommended notations in the ICLR template, but it's true that bold and normal letters are sometimes indistinguishable. We will try to find a way. How about
> $\vec{\textrm{S}}, O.\vec{\textrm{S}}$?
>
> Currently we are considering the FMDP setting. If the transition depends on time, just make "time" a variable (which the predictor is manually given as $t' = t + 1$) in the FMDP.
>
> ## Weakness 2
>
> $\mathcal{F}$ and $\mathcal{F}_s$ are explained in Section 3.2:
> - "$C_k$ specifies a set $\mathcal{F}[C_k]$ of fields that determine the attributes of $O_i$ as well as other instances of $C_k$."
> - We will use $\mathcal{F}_s[C_k]$ to denote the set of state fields of $C_k$ corresponding to state attributes.
>
> Since Definition 2. is a bit far from Section 3.2, it's hard for the reader to catch up. We will add a sentence after Definition 2 about where to find the meaning of $\mathcal{F}$ and $\mathcal{F}_s$.
>
> ## Weakness 3
>
> This is a good question. In our model, we can capture the different influences of classes by using **class-specific encoders**. In other words, each class has its own encoders to encode key and value vectors. See Appendix E for more details. For the predictor $f_{C.V}$, the class uses a key encoder $KEnc_{C_k \rightarrow C.V}$ and a value encoder $VEnc_{C_k \rightarrow C.V}$ to compute the key and value vectors. These class-specific encoders account for the unique mechanism of each class in the dynamics.
>
> ## Weakness 4
>
> We accept your suggestion to briefly explain CIT in the paper. Equation (12) is used to train the predictors in the model instead of learning the causal graph. Then the predictors are used to perform causal discovery through equations (6, 7, 8, 9, 10). Thus, the learning of the causal graph itself is not directly based on gradients. The pseudocode in Appendices E.2. and E.3. further clarifies this process.
>
> # Questions
>
> ## Question 1
>
> The distribution is determined by the domain of the variable, and the domain is given. If the domain is discrete and finite, the distribution is categorical, where the predictor outputs the probability of each category. Otherwise, we assume the distribution is Gaussian and the predictor returns the mean and standard variance.
>
> ## Question 2
> We accept your suggestion about this question and will make the change in the revision.
>
> ## Question 3
> Sorry for the misunderstanding. The symbol (without /) means conditional independence, and the negative (used in eq. 2) means conditional dependence. We will correct this in the revision.
>
> ## Question 4
>
> Consider testing an independence relation between $X$ and $Y$, conditional on $Z$.
> Typically, CIT methods learn two prediction models $\hat{Y} = f(Z)$ and $\hat{Y} = g(X, Z)$ -- that is, we try to predict $Y$ with and without $X$. If the independence holds, $Y$ does not carry information from $X$, so the two prediction models should yield identical results; by comparing the difference between the two prediction models, we can determine whether independence holds. The computational complexity of $f$ and $g$ is bounded by the size of their inputs, which are $O(n)$ in equation (2). Therefore, checking each edge costs at least $O(n)$.
>
> ## Question 5
>
> (a) Sorry for the typo. An ellipsis is missing after $O_{i+1}$. We will correct this in the revision.
>
> (b) We use the semicolon in equation (3) to emphasize the order of the condition variables to which function $p_{C}$ may be sensitive. For example, assuming that $O_1$ and $O_2$ are objects of the same class $C$, based on (3) we have
>
> $$p(O_1.\textrm{S}' | \textrm{U}) = p_{C}(O_1.\textrm{S}' | \textrm{O}_1; \textrm{O}_2, \textrm{O}_3, \cdots, \textrm{O}_N)$$
>
> $$p(O_2.\textrm{S}' | \textrm{U}) = p_{C}(O_2.\textrm{S}' | \textrm{O}_2; \textrm{O}_1, \textrm{O}_3, \cdots, \textrm{O}_N)$$
>
> As we can see, the order of the condition variables in $p_C$ is different in the two equations. Thus, the way $p_C$ handles $\textrm{O}_1$ for $O_1.\textrm{S}'$ is the same as the way it handles $\textrm{O}_2$ for $O_2.\textrm{S}'$. The semicolon can emphasize this positional meaning of condition variables, so it is clear that $\textrm{O}_i$ always takes the first position. Otherwise we would have $p(O_1.\textrm{S}' | \textrm{U}) = p(O_2.\textrm{S}' | \textrm{U})$, which is incorrect.

---

> ### Author Response · Authors · 2023-11-17
> **We have submitted the revised version!**
>
> Dear reviewer, we have submitted the revised version of our paper. These changes in the revision may be related to your concerns.
>
> 1. In some cases the bold letters (e.g. $\mathbf{S}$) are not distinguishable from the normal letters (e.g. $\mathrm{S}$). Therefore, now a normal letter always carries a subscript (e.g., $\mathrm{A}_i, \mathrm{S}_i$, $\mathrm{S}_j$) so they can be well distinguished.
> 2. The notation $\mathbf{U}$ is no longer used. Instead we will use $(\mathbf{S}, \mathbf{A})$ directly. In this way we avoid introducing unfamiliar notations, and $\mathbf{U}$ will not be confused with the letter used in a field $C.U$ or an attribute $O.\mathrm{U}$.
> 3. We have moved the explanation of $\mathcal{F}_s$ into Definition 2 where the notation is used.
> 4. We have added a brief explanation of the Conditional Independence Test (CIT) in Section 3.1. Meanwhile, more information about CIT and how CMI is used for CIT is added in Appendix B.4.
> 5.  The notation of conditional independence in Theorem 1 is now correctly explained. We explicitly use "$\neg$" to avoid confusion.
> 6. We have added the mathematical expression of "independent transition" in Theorem 1.
> 7. We have added the missing ellipsis in Eq. 3 (which is Eq. 2 in the revised version).
> 8. We have added the recommended papers into related works.

---

> > ### Comment · Reviewer_dZfY · 2023-11-20
> > **Official Comment by Reviewer dZfY**
> >
> > Thanks for the thorough response, it has clarified most of my concerns. I would, however like to ask a follow-up question based on the response to the other reviewers. What is the benefit in modeling different classes, as an object $O_1$ of class $C_1$ and an object $O_2$ of Class $C_2$ can both be seen as objects of a shared, common class $C$ such that $O_1.ID = C_1$ and $O_2.ID = C_2$. Have the authors tried performing experiments and testing what kind of performance differences they get in considering the same system in these two different ways?

---

> ### Author Response · Authors · 2023-11-20
>
> Thank you for your response.
>
> We have proposed to introduce "instance ID" to handle the unique property of objects in asymmetric environments. You suggested "class ID" for objects and using the common predictors for all objects, which may work in theory ( provided all classes share the same set of fields). However, we believe that modelling multiple classes has several advantages:
> 1. **Different classes can have different sets of fields**. For example, the class "Human" has a field "Gender", but the class "House" has no such field. By modelling classes, the OO representation is more flexible to such differences.
> 2. The Class ID solution can be problematic in practice. The instinct behind it is that "**objects from the same class should be similar, but objects from different classes are usually very different**", which is a natural concept when people represent the world using the OO technique. Therefore, although instances of the same class may have different dynamics, the difference should be small, so using "instance ID" allows the model to infer the difference. However, **since the dynamic discrepancy between classes is much larger than that between instances of a class, it would be difficult to infer such a discrepancy from class ID using a single neural network.**
> 3. Finally, the use of class ID can severely compromise causal discovery. For example, suppose there are two classes, $Male$ and $Female$, and they have the same set of fields. If we model the two classes differently, we may discover unique causality for each class, e.g. $Female.Beauty \rightarrow Wealth$ but not $Male.Beauty \not\rightarrow Wealth$ (just an example here). However, if we do not treat them differently, we will have $Object.Beauty \rightarrow Wealth$ and $Object.Gender \rightarrow Wealth$ (in this case, gender is the class ID). Then the causal graph does not say that the causality only exists for women. This problem would be much more severe if the difference between classes was large. In fact, the same problem exists in our instance ID solution. However, because objects in the same class are very similar, the problem is tolerable.
>
> It is important to note that humans naturally categorize objects in a way that minimizes the differences within a class and maximizes the differences between classes. This motivates us to model different classes and benefits the performance of OOCDM. The same idea is used in modern object-oriented programming. For example, Python also uses "obj.__class__" as a "class ID", but it also benefits from modeling class-level behavior.
>
> Unfortunately, no further experiments can be done as the rebuttal phase is about to end. If you are still interested, you can leave your email after the final decision and I will send you the results.
>
> We hope this answer clears up your doubts. We would be grateful if the score could be adjusted accordingly.

---

> > ### Comment · Reviewer_dZfY · 2023-11-20
> > **Official Comment by Reviewer dZfY**
> >
> > Thanks to the authors for providing clarifications. However, I am interested in whether this motivation translates to experimental results, since different classes can be embedded into a setting where there is only a single class, and the empty fields for each class can just take a special token, like **Null**, as long as the modeling functions are flexible enough. I understand that the rebuttal phase is about to end, but I would very much appreciate an experiment either in the main draft or the Appendix that presents such results for the camera ready / next iteration depending on if it gets accepted. I will, however, update my score from 5 to 6 as most of my concerns have been addressed; and will look forward to the experiment :)

---

> > > ### Author Response · Authors · 2023-11-20
> > > **Thank you for the kind response**
> > >
> > > We are grateful for your kind comments and for updating the score. In fact, we are interested in extending the model with "class inheritance" in our future study. All classes can be a subclass of some parent class, like "Dog" is the subclass of "Animal", in this way we can treat all subclasses of "Animal" as a single class when discussing the common attributes of animals. Meanwhile, a subclass of "Dog" can have its own unique attributes. We believe this may be a better way to interpret "class merging", and we look forward to digging further in this direction.

---

> > > > ### Comment · Reviewer_dZfY · 2023-11-21
> > > > **Official Comment by Reviewer dZfY**
> > > >
> > > > That would indeed be a cool follow-up, but it would still be an important sanity check just to see if experimentally the benefits hold when considering multiple classes as opposed to a single homogeneous class with an ID attribute.

---

> > > > > ### Author Response · Authors · 2023-11-21
> > > > >
> > > > > We are grateful for your useful suggestion. We will add an additional experiment into the Appendix about that issue in the final version :-)

---

### Official Review · Reviewer_LMvE · 2023-10-31

**Soundness:** 3 good
**Presentation:** 3 good
**Contribution:** 3 good
**Rating:** 5
**Confidence:** 4

**Summary:**

This paper discuss the current progress of learning and applying Causal Dynamics Models in training RL agents. They note that challenges arise in large-scale RL environments with vast number of objects and complex causal dependencies. Inspired by humans' object-oriented (OO) perspective for task perception, the authors developed the Object-Oriented Causal Dynamics Model (OOCDM). OOCDM allows sharing causalities and model parameters among objects from the same class, thus enhancing causal discovery and learning efficiency in RL. The proposed modified Causal Dynamics Learning (CDL) paradigm accommodates varying numbers of objects. Experimental results demonstrate the performance of OOCDM in terms of causal graph accuracy, prediction accuracy, generalization, and computational efficiency.

**Strengths:**

(1) the authors formally defined the problem of object oriented learning in RL from a causal perspective, which paved the way for future work.
(2) the authors identified the core problem of the current causal dynamics model learning algorithms, which is the poor efficiency in the face of mass variables. The proposed method alleviated this problem to some extent with intuitive explanations and thorough justifications.
(3) the writing is clear and easy to follow.

**Weaknesses:**

(1) Dynamics bayesian networks are rather limited rendering them unsuitable to model causal mechanisms. Especially when there is a need for cross layer inference, i.e., identifying the effect of an intervention from observational data. For more information, please see "On Pearl’s hierarchy and the foundations of causal inference." in Probabilistic and Causal Inference: the works of Judea Pearl. In the specific scenario like the one define in this paper, it would be fine (Markovian, no confounders). But still, I would suggest the author use more rigorous causal tools to model the problem.

(2) In section 3.2, when the authors started to define OOMDP, they claim that "following (a Factored MDP paper), we formulate the task as an Object-Oriented MDP (OOMDP)...". The way of presenting the concept totally ignores a seminal previous work that proposed OOMDP already [1]. Interestingly, the authors also referred to this paper in the related works section but seem to forget it here in the main text.

(3) In section 4, the core assumption of this paper, "causation symmetry", which assumes that objects from the same class have the same causal effect on other objects, is too strong to be practical. Consider a system where three objects from the same class are linked one by one by springs in a line (obj1 --spring-- obj2 --spring-- obj3). For the causal effect of pulling obj2/obj3 on obj1, they are not interchangeable due to the existence of springs.

(4) The proposed method requires manual construction of the OOMDP representation (obj classes, attributes fields, etc.). When there are truly a vast number of classes, this would be a new bottleneck for scaling. This fact, to some extent, hinders the authors' original goal of improving the efficiency of current CDMs learning methods. Similarly, when each object belongs to a distinct class, the proposed method might be even worse due to those detailed object fields.

(5) For the experiments, the authors seem to be using a factored state space observation space. If it's not a high-dimensional pixel one, model-free methods are already competitive enough to solve state space observations problems. I doubt if there is really a strong need to introduce the extra complexities of the object oriented learning here. In another word, I do believe there are values in the OO representation and the proposed method even under the strong assumption of "causation symmetry", but it would be better demonstrated via high-dimensional challenging tasks.

[1] Diuk, Carlos, Andre Cohen, and Michael L. Littman. "An object-oriented representation for efficient reinforcement learning." Proceedings of the 25th international conference on Machine learning. 2008.

**Questions:**

(1) Could you elaborate the potential solution if people plan to apply your methods to environments with pixel observations?

(2) Is there a more principled way of defining the object classes and attribute fields?

(3) How would you compare your work with "Causal dynamics learning for task-independent state abstraction." from ICML 22'? Is your proposed method significantly more efficient than theirs?

I will consider change my rating if those questions and weaknesses are properly handled.

---

> ### Author Response · Authors · 2023-11-11
>
> Thank you for your kind opinions and comments.
>
> # Weakness 1
>
> In this paper, we focus on Factored MDPs (FMDPs), so that methods based on FMDPs would benefit from our work. The baselines (CDL, GRADER, TICSA) are all based on the settings of Factored MDP, and their settings also exclude confounders. Thus, we only claim to provide a solution to the current scalability problems of these baselines. With respect to the claim of our paper, we believe that using FMDP does not harm the rigor of this paper.
>
> We agree that the use of FMDP imposes limitations. As you pointed out, there are no confounders and the transition is strictly Markovian, which is not applicable in some real-world problems. We will add this statement about this weakness in Appendix I. Extending our methods to an environment with non-Markovian dynamics, partial observability, or latent confounders is interesting but also challenging.  In our future work, we will certainly investigate such more challenging problems.
>
> # Weakness 2
>
> The reason why [1] is missing in Section 3.2 is that the definition of [1] formulates the dynamics through a set of logical rules (so [1] is a special case for our problem). We do not hope to impose this strong restriction on our setting of the problem, so we avoid citing [1] in Section 3.2. Since "OOMDP" seems to be the most relevant way to name the problem, we have also used it in our paper. We will add a brief explanation of this issue in Section 3.2 in the revision.
>
> # Weakness 3
>
> The problem can be solved by introducing additional attributes in the representation. We can include "identity encoding" in the attributes of objects. For example, we can define $Obj.ID$ fields for these objects. For example
> $$ Obj1.\textrm{ID} = 1, Obj2.\textrm{ID}=2, ... $$
> Then the objects become interchangeable, so causal symmetry holds. Then these IDs will be well handled by our attention-based network, which can naturally assign high attention weights to connected objects.
>
> # Weakness 4
>
> You are right. The key of our method is to share casual structures between objects. But if all objects are from different classes, there is no way to share causal structure. In this case, we have $m$ (the number of fields) equal to $n$ (the number of variables), then the time complexity of causal discovery becomes $O(n^2N)$, where $N$ is the number of objects. In the worst case, we have $N = n$ (each object contains only 1 variable), then the complexity is as bad as for non-object-oriented methods. However, we hope to clarify that our method is not designed to solve such an extreme case, which is against the motivation of our approach.
>
> # Weakness 5 and question 1
>
> The baseline CDMs (GRADER, CDL, TICSA) are all based on factored spaces, so adopting FMDPs in our settings is reasonable for our claim. In fact, we came up with the idea of this paper when trying to learn CDMs in some previous studies, where we found that causal discovery is really slow, and a way to speed up the process is in dire need even in factored environments.
>
> There are studies on object-centric representations (OCR), which can learn homogeneous object representations from image inputs. Thus, OCR methods are compatible with our approach when the number of classes is exactly 1, and some adaptation of the networks and the loss function may be required. However, we believe that the solution should not be limited to single class cases, which does not fully exploit the potential of an OOCDM.
>
> In future work, we may try to extend OCR to multi-class scenarios (e.g., a one-hot vector in the feature of each object to identify its class, and then use different decoders for each class to obtain the attributes). However, much deeper work is needed to ensure that the representation follows causal symmetry in the future.
>
> # Question 2
>
> We tried to give a more rigorous definition in Appendix D.1. (Please ignore some minor typos there, as the submission was in a hurry.)
>
> # Question 3
>
> In fact, the paper in ICML 22' (which we refer to as "CDL") is one of my baselines in the experiments. The original CDL uses a pooling mechanism to aggregate information from causal parents, which we found to be very ineffective in large cases. Therefore, we also considered the attention-based variant of CDL, called "CDL-A". Here are the general conclusions of the comparison:
> 1. The accuracy of causal graphs (Table 1): Ours > CDL-A > CDL
> 2. The accuracy of prediction (Table 3): In the training set, we have Ours > CDL-A > CDL. However, in the in-distribution test set and out-of-distribution test set we have Ours > CDL > CDL-A, so it seems that attention causes more overfitting for CDL.
> 3. Computational complexity (Table 2): Theoretically, Ours <= CDL = CDL-A. Experimentally, Ours <= CDL < CDL-A. Here, Ours = CDL happens only in small-scale environments.
> 4. CDL and CDL-A cannot adapt to varying numbers of objects, but ours can.
>
> In summary, our model is certainly much more efficient than CDL and CDL-A.

---

> > ### Comment · Reviewer_LMvE · 2023-11-21
> >
> > Thank you for your clarifications! Most of my concerns have been addressed. But there are still two questions left.
> >
> > ### Weakness 3
> > Could you elaborate in more details of why adding a unique ID will make objects in such a string system (obj1 --spring-- obj2 --spring-- obj3) "interchangeable"?
> >
> > Say we are interested in the movement of obj1 in the next few time steps. And in the first scenario, we pull obj2 away from obj1 and apply no force to obj3. In the second scenario, we pull obj3 with the same force but apply no force to obj2. If causal symmetry holds, does this indicate that obj1 will move in the same way in these two different scenarios?
> >
> > ### Weakness 4
> > It seems that my question "The proposed method requires manual construction of the OOMDP representation (obj classes, attributes fields, etc.). " hasn't been answered yet. Would you provide more detailed information on this front?

---

> ### Author Response · Authors · 2023-11-17
> **We have submitted the revised version**
>
> Dear reviewer, we have submitted the revised version of our paper. These changes in the revision may be related to your concerns.
>
> 1. In Appendix J (Weaknesses and future works) we have added a discussion about the limitations of using the Factored MDP. In the future, we will explore more complicated scenarios involving partial observability, confounders, and non-Markovian dynamics.
> 2. In Section 3.2, we have added a sentence explaining why our OOMDP does not fully conform to the definition in "An object-oriented representation for efficient reinforcement learning" (Diuk et al).
> 3. In Appendix I, we have provided a simple solution (with an additional experiment) to extend OOCDM to environments where causation symmetry does not hold. In short, by adding auxiliary identity attributes to the objects, we can ensure causation symmetry and result symmetry in all OOMDPs.

---

> ### Author Response · Authors · 2023-11-22
>
> Thank you for your reply.
>
> # About Weakness 3
>
> We have provided a formal proof in Appendix I.1. In short, no matter how we change the order of the objects, the unique IDs allow us to restore the original order. In this way, OOCDM can learn to map the ID to the unique property of the object.
>
> For example, we have $O_i.ID = i$ in the system, and use $O_i.X$ to denote the old attributes of $O_i$ without the ID. Then, in the first scenario you described, we are interested in
> $$p(O_1 | O_2.Force=1, O_2.Id=2, O_3.Force=0, O_3.Id=3)$$
> In the second scenario, we are interested in
> $$p(O_1 | O_2.Force=0, O_2.Id=2, O_3.Force=1, O_3.Id=3)$$
> Checking the definition of causation symmetry, the model does not consider them strictly symmetric, **because the $ID$ attributes are not swapped between O_2 and O_3**. By inserting IDs, causation symmetry should be:
> $$p(O_1 | O_2.Force=1, O_2.Id=2, O_3.Force=0, O_3.Id=3) = p(O_1 | O_2.Force=0, O_2.Id=3, O_3.Force=1, O_3.Id=2),$$
> where the IDs of $O_2$ and $O_3$ are also swapped. Then the OOCDM can easily restore the original order of the objects. In our implementation, the information of the IDs is encoded in the key, value, and query vectors in the attention module, which successfully handles the IDs in the additional experiment.
>
> You can think of the IDs as the "positional encoding" in the Transformer architecture. The key-value attention mechanism is agnostic to the order of the tokens, but the positional encodings allow Transformer to infer spatial information.
>
> # About Weakness 4
>
> It is true that the proposed method requires manual construction of the OOMDP representation. Therefore, we only claimed to learn causal dynamics in object-oriented environments instead of arbitrary environments. Piror's work [1,2], which studies the same problem, is also based on given representations, and we do not claim to remove this requirement. Constructing the OOMDPs from pixel input or raw factorization is still an open problem, and we will certainly investigate this direction in the future. There are potential solutions, such as extending OCR to multi-class domains. However, proposing any efficient solution is a huge breakthrough in the field and enough to support another paper.
>
> In addition, the solution of ID attributes for objects allows causation symmetry and result symmetry to be violated. OOCDM still works well if the violation is not severe. In this way, constructing the OOMDP representation is flexible, as it's not necessary to guarantee the result and causation symmetries.
>
> If there are many classes, we can also **combine similar classes into a superclass**. An auxiliary field specifying the subclass can be inserted into the superclass so that the model can infer the subtle differences of objects in these combined classes. For example, creatures in our world come from countless species, and creating a class for each species is indeed cumbersome. However, we can categorize creatures according to a coarser standard and simply use superclasses like $Animal$ and $Plant$, and the species of the creature is described by an additional attribute $O.\mathrm{Species}$. In our future work we will also investigate more complicated class hierachy, so that the model can better handle scenarios with many classes.
>
> Moreover, we believe that even though the OOCDM requires a known representation, it is still useful:
> 1. There are many environments that are inherently object-oriented. Like StarCraft, every unit is an object. In some multi-agent domains, such as multi-agent partial environments (MPE), each particle is an object.
> 2. Describing the environment in the object-oriented way is quite natural, so that the representation can be constructed from the instinctive understanding of the users (i.e. we don't need a real expert).
>
> [1] C. Diuk, A. Cohen, and M. L. Littman, “An object-oriented representation for efficient reinforcement learning,” in Proceedings of the 25th international conference on Machine learning, 2008.
> [2] C. Guestrin, D. Koller, C. Gearhart, and N. Kanodia, “Generalizing plans to new environments in relational MDPs,” in Proceedings of the 18th International Joint Conference on Artificial Intelligence, 2003.
>
> I sincerely hope these answers adress your concerns, so that the scores can be adjusted accordingly.

---

> > ### Comment · Reviewer_LMvE · 2023-11-22
> >
> > Thank you for your timely response!
> >
> > I can understand your proposed solution of introducing ID attributes. But in this way, it seems that we lose some advantages of being "interchangeable". Clearly by having unique IDs, the network should treat them differently. And this is also related to your claim that "OOCDM still works well if the violation is not severe." Could you elaborate more formally on when this violation is said to be "not severe" and OOCDM still works well?

---

> ### Author Response · Authors · 2023-11-22
>
> Thanks for the response.
>
> First of all, in theory, if we provide the ID attributes, there is always an OOCDM that represents the dynamics. This is formally proven in the appendix. However, **learning this OOCDM may not be tractable if the violation is severe**, because the OOCDM is implemented by neural networks with a limited number of parameters and must be trained with a limited amount of data.
>
> Therefore, "the violation is not severe" means that this dynamics is within the capability of a practical neural network -- that is, **the mapping from the IDs to the individual influence on the dynamics is trainable for our attention-based networks**. Otherwise, we might need a really deep network and a large amount of data to learn this dynamic. For example, in an extreme case, objects of the same class choose completely different dynamics functions based on their IDs, and these functions do not share any common factors. Then the OOCDM has to remember each function independently and learn to choose the correct function based on the ID. In this case, the violation is so strong that the discrepancy between objects of the class is not learnable.
>
> Conceptually, we can write the dynamics for an object $O_j$ as
> $$p(O_j|O_1,...,O_N) = f \big( g(O_1^{-ID},...,O_N^{-ID}), h(O_1,...,O_N), Noise \big)$$
> where $g$ is the shared and interchangeable part of the dynamics, $h$ involves the individual influence of the objects, and $f$ merges the shared and individual influence into the final dynamics. The "large violation" then means that $h$ has a dominant effect and is too complicated to learn. Note that the above fomulation is just an example. The true dynamics may take another form, but it is supposed to have shared and individual parts.
>
> Fortunately, **humans naturally categorize objects in a way that minimizes differences within a class and maximizes differences between classes**. Therefore, even though objects may have individual properties, these properties are easily inferred from their IDs. Therefore, the proposed OOCDM is suitable for most cases because the mapping from IDs to the unique properties of objects is usually learnable.
>
> There is no rigorous mathematical definition of "severe violation", since the implementation of OOCDM can always be augmented with more powerful networks so that the dynamics can always be learned with sufficient data (in theory, neural networks can be arbitrarily strong). In this work, we may do so by increasing the layers of key, value, and query encoders in the predictor networks. However, we are talking about an affordable implementation of OOCDM with high computational efficiency -- if such an implementation exists, we say "**OOCDM works well**", which happens when the mapping from IDs to the individual influence of objects is not difficult to learn.

---

> > ### Comment · Reviewer_LMvE · 2023-11-22
> >
> > Thanks for the detailed explanation! I have adjusted my score. But I would still recommend the authors focus on relaxing the causal symmetry assumptions later on, which would make the submission even stronger.

---

> > > ### Author Response · Authors · 2023-11-22
> > >
> > > Thank you for the extensive discussion. Your advice is extremely valuable. The current solution of relaxing the assumptions (introducing identity attributes) may be useful in relaxing the assumptions, but more effective ways need to be explored (e.g., extracting the representation that ensures symmetry from raw inputs, or modeling relational interactions). These solutions require a lot of work, so we can't guarantee that we'll make the breakthrough before the final submission (if accepted). Nevertheless, we will add some more experiments about relaxing the symmetries by using auxiliary attributes, and try to improve the performance of the current solution :-)

---

### Official Review · Reviewer_mQjQ · 2023-10-31

**Soundness:** 3 good
**Presentation:** 2 fair
**Contribution:** 3 good
**Rating:** 6
**Confidence:** 3

**Summary:**

This manuscript introduces a comprehensive framework for object-oriented reinforcement learning (OORL), utilizing an object-oriented causal dynamic model to capture the intricacies of the OORL environment. The framework adeptly simplifies complex RL scenarios through a decomposition based on object types, ensuring that similar objects share parameters and causality characteristics. This integration of inductive biases such as causal symmetry and result symmetry enhances the model’s capability to represent complex scenarios effectively. The authors provide the framework to learn the corresponding model and the effectiveness of the proposed model is substantiated through a set of empirical experiments.

Overall, this work stands out as an effective and logically sound approach to modeling the OORL environment, with a particular emphasis on causality. It is poised to make significant contributions to the field. Despite these strengths, there are certain elements of the method, presentation, and experimental design that will be further clarified by the authors. Addressing these points is crucial for a comprehensive understanding of the work. Given these considerations, my initial inclination is to recommend a borderline accept.

**Strengths:**

**[Motivation and General Idea]** The manuscript establishes a robust motivation, addressing the critical challenge of inefficiencies prevalent in Causal Dynamic Models (CDMs) within complex reinforcement learning (RL) scenarios. The authors’ choice to navigate RL in the context of multiple objects and categories is judicious and aligns well with the overarching theme of the paper.

**[Proposed Framework]** The architectural design of the proposed framework is simple to follow and technically sound. The framework extends the classical paradigms of Object-Oriented Markov Decision Processes (OO-MDP) and Relational Markov Decision Processes (Relational MDP), incorporating causality as a vital inductive bias. This strategic design choice facilitates an empirically grounded solution, paving the way for efficient causal discovery in settings populated with numerous objects.

**Weaknesses:**

I listed the weaknesses and questions here. I am also a reviewer of the previous version of this paper. Most of my concerns have been addressed by the authors. However, some of them are still a bit unclear to me.

- **[Definition of the local causality]**

When we typically mention local causality, we refer to the case where the causal edges vary, as illustrated in Figure 2 in [1]. In this paper, when the authors mention local causality, it refers more to the causal graph of transitions for individual category objects. I think it is better to give a clarification in the revised version.

- **[About potential relational interactions]**

While the presented work demonstrates a comprehensive approach to object-oriented reinforcement learning, it appears to omit direct consideration of interactions between objects or entities within the framework. Given the prevalence of interactions among objects in real-world reinforcement learning scenarios, particularly in tasks involving object manipulation, this seems to be a significant aspect to address. Could the authors shed light on whether the framework is capable of learning and accounting for these interactions? Insights on how the model might be extended or adapted to incorporate direct interactions among objects would be greatly beneficial, as it would enhance the applicability of the approach to a wider array of real-world scenarios.

- **[Potential to extend to image domains by combing the OCR models]**

In light of works such as [1-3], which integrate Object-Centric Representations (OCR) models into the learning process, the authors might consider incorporating similar OCR models equipped with permutation invariant modules at the beginning of their pipeline. This would enable the extraction of object representations directly from image data, making the method more scalable and applicable to a broader range of real-world scenarios.

*References*

[1] Pitis, Silviu, Elliot Creager, and Animesh Garg. "Counterfactual data augmentation using locally factored dynamics." Advances in Neural Information Processing Systems 33 (2020): 3976-3990.

[2] Yoon, Jaesik, et al. "An investigation into pre-training object-centric representations for reinforcement learning." arXiv preprint arXiv:2302.04419 (2023).

[3] Zadaianchuk, Andrii, Maximilian Seitzer, and Georg Martius. "Self-supervised visual reinforcement learning with object-centric representations." arXiv preprint arXiv:2011.14381 (2020).

[4] Kossen, Jannik, et al. "Structured object-aware physics prediction for video modeling and planning." arXiv preprint arXiv:1910.02425 (2019).

**Questions:**

I listed the weaknesses and questions together in the above section.

---

> ### Author Response · Authors · 2023-11-11
>
> Thank you for your suggestions and comments. It is a great luck to see you again as my reviewers, and the suggestions on the previous version really helped me in improving this work.
>
> # About Weakness 1
>
> The suggestion is very useful here.  In fact, In Section 4.1 we write "The local causality describes shared structures within
> each object" and used Figure 3(a) to further exlain the concept. We will add a footnote with citation to [1] in the revision to reduce any misunderstanding.
>
> # About Weakness 2
>
> It is a interesting topic to consider "relational interactions" in object-oriented dynamics. Here we may come up with a few solutions for that concern.
>
> First of all, our mathod supports an **implicit solution**. We may assign an additional identification attribute called "ID" for each object, then the attention model should be able to capture the relational interaction. High attention weights will be assigned to the objects that take part in the interaction by matching to their IDs. If there are different kinds of relations, we may use multi-head attention in the model. The drawback of this approach is that these relations are implicit in the causal graph.
>
> The second solution is **introducing additional attributes about who the object may be interacting with**. For example, in the starcraft environment, we may add a feild like "Marine.AttackTargetID", which gives the ID of the target object that the marine is attacking in an "attack" relation. A special value for the attribute means that the marine is attacking no one (i.e., not in an attack relation). Then, we probably will have these causations in the causal graph:
> $$ Zerg[Marine.AttackPoint \rightarrow Health], Zerg[ID \rightarrow Health], Zerg[Marine.AttackTarget \rightarrow Health]  $$
>
> The third solution is **introducing addition relation classes for each kind of interaction**. The fields of the class are the IDs of participants of the relation. For example, we may define a class $Attack$ for the attack relation, which has fields $Attack.Attacker, Attack.Target$.  Then, we probably will have these causations in the causal graph:
> $$ Zerg[Attack.Target  \rightarrow Health], Zerg[ID  \rightarrow Health] $$
>
> Note that in the second and third solutions, the participants of the relations become a state variable in the environment and can be predicted by the model.  Moreover, the network may be specially designed to better handle the $ID$ attributes with relational interactions.
>
> # About Weakness 3
>
> Objects in OCR methods are usually homogeneous. Thus, OCR is compatible with our approach when the number of classes is exactly 1, while some adaptation of the network architecture and the loss function may be required. Unfortunately, the rebuttal period is too short to tune these details and perform an additional experiment. In the meantime, a solution limited to a single-class environment does not fully exploit the potential of an OOCDM.
>
> The suggestion to adapt OCR to our work seems to be an interesting and valuable direction. In future work, we may try to extend OCR to multi-class scenarios (e.g., a one-hot vector in the feature of each object to identify its class, and then use different decoders for each class to obtain the attributes). However, much deeper work is needed to ensure that the representation follows causal symmetry.

---

> ### Author Response · Authors · 2023-11-17
> **We have submitted the revised version**
>
> Dear reviewer, we have submitted the revised version of our paper. In Section 4.1, we have added a sentence about the local causality to clear the confusion with respect to the article "Counterfactual Data Augmentation using Locally Factored Dynamics" (Pitis, et al), where they define "local causality" as the causal structure when $(\mathbf{S}, \mathbf{A})$ is confined in a local subset of the state-action space.

---

> ### Comment · Reviewer_mQjQ · 2023-11-20
>
> Thanks for the detailed feedback. Most of the concerns have been addressed and thanks for the clarification on the local causality concepts. I would keep my score for now and be attentive to the discussion with other reviewers.

---

### Official Review · Reviewer_6VtU · 2023-11-02

**Soundness:** 2 fair
**Presentation:** 3 good
**Contribution:** 2 fair
**Rating:** 6
**Confidence:** 3

**Summary:**

This paper considers causal dynamics learning in an object-oriented environment, which is modeled using Object-oriented (OO) MDP. Based on the setting, the authors define OO causal graph that illustrates how attributes of different entities influence other attributes of possibly different entities at the next step. With a measure of conditional dependence (CMI) and flexible model (attention) for conditional probability involving OO, the authors demonstrated the usefulness of OO approach to causal dynamics modeling.

**Strengths:**

- The use of OO concept in causal dynamics learning seems a very well-motivated work where real-world environments are naturally multi-agent setting with heterogeneous players.

- I guess the use of attention to handle a varying number of objects’ attributes seems a clever idea. (Is this idea already adopted in other OO related research?, The authors only left a figure and a single sentence before Section 4.3)

**Weaknesses:**

As a researcher who worked on causal discovery in relational data and causal dynamics learning, this combination seems interesting. However,

- The combination seems a bit not nontrivial in a sense that, while the fomulation is a bit complicated but, at a fundamental level, we just generalize causal dynamics learning to an object-oriented version, and apply the idea of conditional independence. Causal dynamics learning or causal discovery is just a transition probability between the two time steps, and we replace probabilities of random variables to represent probabilities among attributes of objects. We refine variables to consider only the relevant ones.

- This OO causal graph or causal discovery is not a new concept (see Marc Maier and Prof. David Jensen (UMASS)’s work on relational causal discovery), which is a more general setting not just t-1 and t. Temporal version is also proposed (Marazopoulou et al. Learning the Structure of Causal Models with Relational and Temporal Dependence, UAI 2015) . There is no special needs to class-level local/global causality.

**Questions:**

- Regarding “Scalability”, do you mean this part “This causality sharing greatly simplifies causal discovery and improves the readability of CGs in large-scale environments. “ in page 4? If so, isn’t this an already existing result in relational causal discovery? Otherwise, where does the claim of scalability come from?
=========
after the discussion, I see that the paper has certain merits (introducing OO into RL) and raising my rating from 5 to 6.

---

> ### Author Response · Authors · 2023-11-11
>
> Thank you very much for your questions and comments. We are happy to have a professional researcher in causal inference review our paper. We came up with this idea while trying to use causality to address other problems in RL last year, so it is possible that we have neglected some literature in the area of causal inference. Indeed, Relational Causal Discovery (RCD) has similar ideas to ours. Therefore, we will be sure to mention the related papers in the revised version.
>
> # About the question in Strengths
>
> There are other attention-based models in object-oriented environments (usually considering homogeneous objects) using different architectures. In our work, the network is originally designed to compute according to the given causal structure.
>
> # About Weakness 1
>
> This paragraph actually gives a summary of our work -- "generalizing causal dynamics learning to an object-oriented version" is our contribution, and "refining variables to consider only the relevant ones" is our motivation. I am not sure why these statements constitute the weaknesses of our work. The combination seems straightforward, but many issues need to be considered. This paper answers 1) under what condition is the same causal structure shared? 2) Exactly which variables are relevant? 2) what network architecture can be used for inference, and 4) how much does this reduce computational complexity. Meanwhile, we have demonstrated the effectiveness of this combination through experiments.
>
> # About weakness 2
>
> I believe there is some necessity to use class-level causality in our setting. We focus on learning the causal structure in a **Factored MDP** because there have been a large number of studies using this structure to solve their problems, as mentioned in Section 2.1. The OOMDP is just a special case of the FMDP, so our model can directly serve all methods that work on the FMDP, as long as an object-oriented segmentation of the FMDP's variables is available.  **RCD involves another concept called "relations", which is not modeled in FMDPs or OOMDPs -- these relations may exist, but are implicit in the factorization**. Consider the example of students taking courses (Fig. 2 in [1]), the relation "Take" may be implicitly described by two vectors $Alice.Take = (1, 0)$, $Bob.Take = (1, 1)$ in the FMDP. The OOMDP (FMDP) will only model these relationships as attributes (variables) in the state space, so class-level causality is needed to describe the causal graph when relations are not modeled or implicit (e.g., $Course[Student.Take \rightarrow Difficulty]$).
>
> In addition, the classes and objects in OOMDP have already required a lot of domain knowledge about the environment. If the domain knowledge about relations is also required, the approach would be much more limited. Even a layman can instinctively know how to classify objects, but relations sometimes require experts. Meanwhile, RCD requires to learn a unique model for each type of relation, which is also very expensive, while our solution has a lower complexity.
>
> # About Questions
>
> Scalability is embodied in the paper in many ways, not just in the sentence you quote. Of course, that sentence is already covered by the RCD papers, so I will add a reference there in the revised version. However, most of the claim about "scalability" is still original, embodied in the following aspects:
> 1. Part of the scalability comes from the **Causal Symmetry** (Eq. 4), which is not included in RCD. This symmetry allows us to use attention in the model because the input key and value vectors of attention are also **symmetric**. In this way, the model can capture the joint influence of other objects and manage different numbers of objects. In the RCD studies, their regression models do not have this property. As a result, in Fig. 9 of [1], the precision of causal discovery decreases when the number of entities increases from 2 to 3. However, the causal discovery of our model achieves higher precision with more entities.
> 2. RCD only concludes that the scalability of the causal discovery is improved, but we also examine the scalability of the **prediction performance** of the model. Many RL applications require the model to be accurate in predicting the next state. Table 3 in the main paper shows that OOCDM outperforms other causal models.
> 3. RCD does not conclude that scalability about **computational cost**. In some applications, such as model-based policy optimization, causal discovery is performed repeatedly in each iteration with new data collected. In large-scale environments, causal discovery can be excessively time-consuming, making the application of causal models intractable. In such applications, computational cost may be the most important aspect of scalability. In Table 2, we show that our object-oriented version of the model greatly reduces computation time.
>
> [1] Marazopoulou et al. Learning the Structure of Causal Models with Relational and Temporal Dependence, UAI 2015

---

> ### Author Response · Authors · 2023-11-17
> **We have submitted the revised version.**
>
> Dear reviewer, we have submitted the revised version of our paper. We have mentioned these papers in Related Works (Section 2):
> 1. Maier, et al. Learning causal models of relational domains, 2010.
> 2. Marazopoulou, et al. Learning the structure of causal models with relational and temporal dependence, 2015.

---

> > ### Comment · Reviewer_6VtU · 2023-11-20
> > **follow-up**
> >
> > “Part of the scalability comes from the Causal Symmetry (Eq. 4), which is not included in RCD.”
> > It is not true. For example, when considering smoking of friends, RCD does not consider the order of friends, etc.
> >
> > How does causal symmetry “manage different numbers of objects”? I only can see the unorderedness among the objects of the same class but cannot see the equality among different number of objects.
> >
> > “RCD involves another concept called "relations", which is not modeled in FMDPs or OOMDPs” Does this mean that there is a room for designing a new problem and devise new algorithm for “Learning Causal Dynamics Models in Relational Environments” that is not covered by the paper.
> >
> > (BTW, I will soon reflect your response into my rating.)

---

> > > ### Author Response · Authors · 2023-11-20
> > > **Thanks for your response**
> > >
> > > Thank you for your response.
> > >
> > > # 1
> > > Sorry for the confusion. You are right. Looking more closely at the RCD paper, we found that RCD did include causaltion symmetry in the implementation. However, RCD clearly did not make good use of it. It is true that RCD works fine if an oracle d-separation test is available.  Without such an oracle tool, RCD still has to learn regression models and perform t-tests based on the model. RCD (Marazopoulou, et al) uses a linear regression model with average aggregation on the condition terms. Although average aggregation satisfies symmetry, it compromises the scalability of the model. When the number of objects is large, average aggregation may obscure important information. Similarly, CDL (Wang et al.), one of our baselines, uses maximum aggregation, which is also poor in large-scale environments. In this work, we highlight the importane of the causation symmetry, which allows us to use attention-based aggregation with significant improvement on scalability.
> > >
> > > # 2
> > >
> > > Strictly speaking, it is the design of the model architecture that allows it to manipulate different number objects. Thanks to the model architecture, data containing different numbers of objects can be used directly for model training and causal discovery. By collecting data from tasks with different numbers of objects, the model implicitly learns how the number of objects relates to dynamics.
> > >
> > > The function of causation symmetry is to ensure that the cuasal graph is an OOCG, which is the basis of the above model design. The OOCG is defined at the class level, i.e. it is irrelevant to the number of objects. Therefore, we can use only one OOCG to describe the causalities within a family of tasks (each task is an OOMDP).  If the OOMDP family guarantees that the class-level causalities are invariant, then the OOCG is invariant across tasks, which can be learned by our OOCDM. Otherwise, if the OOMDP family contains varying OOCGs, then our approach can learn the union of causalities in all OOMDPs -- i.e., the minimal causal graph that allows a model to infer the dynamics in all tasks in the family.
> > >
> > > # 3
> > >
> > > Yes, there is definitely a room for "learning causal dynamics models in relational environments". Another reviewer also implies that learning causality in a relational interaction should be an interesting direction. Current OOCDM can handle relations by introducing auxiliary attributes or additional relation classes. However, in these solutions, OOCDM still treats relations as variables in the OOMDP. New causal discovery algorithms can be designed to handle relations more efficiently. Special network structures can be designed to improve the performance of the method.

---

### Author Response · Authors · 2023-11-12
**The Revised Version Has Been Submitted!**

We are grateful for all the suggestions and comments, which may greatly improve the quality of our paper. The revised manuscript has already been submitted.  We sincerely hope that these changes will address the concerns of the reviewers so that the scores can be reassessed accordingly. Here is a brief summary of the changes made in the update.

# Clarity of the paper

1. in Section 4.1, we have added a sentence about the local causality to clear the confusion with respect to the article "Counterfactual Data Augmentation using Locally Factored Dynamics" (Pitis, et al), where they define "local causality" as the causal structure when $(\mathbf{S}, \mathbf{A})$ is confined in a local subset of the state-action space.
2. In Section 3.2, we have added a sentence explaining why our OOMDP does not fully conform to the definition in "An object-oriented representation for efficient reinforcement learning" (Diuk et al).
3. We have added the mathematical expression of "independent transition" in Theorem 1.
4. We have added a brief explanation of the Conditional Independence Test (CIT) in Section 3.1. Meanwhile, more information about CIT and how CMI is used for CIT is added in Appendix B.4.
5. We have moved the explanation of $\mathcal{F}_s$ into Definition 2 where the notation is used.
6. In Section 5.2, we have explained that the accuracy of causal graphs is measured by Structural Hamming Distance.
7. In Appendix J (Weaknesses and future works) we have added a discussion about the limitation of using the Factored MDP. In the future, we will explore more complicated scenarios involving partial observability, confounders, and non-Markovian dynamics.
8. We have visualized the adjacency matrices of the ground truth OOCGs in Appendix G and the discovered OOCGs in Appendix H. These figures are intended to be more direct and understandable than the old presentation of OOCGs.

# Typos fixed

These typos found by reviewers have been fixed:

1. The notation of conditional independence in Theorem 1 is now correctly explained. We explicitly use "$\neg$" to avoid confusion.
2. We have added the missing ellipsis in Eq. 3 (which is Eq. 2 in the revised version).
3. The subscripts of objects in Eq. 4 (now Eq. 3) are now corrected.
4. in Table 7, "action" is corrected to "state" for the meaning of $O.\mathbf{S}$.

We will continue to fix typos in the paper for the final version.

# Notation

Some reviewers are concerned about the readability of the notations. So we have made a few changes.
1. In some cases the bold letters (e.g. $\mathbf{S}$) are not distinguishable from the normal letters (e.g. $\mathrm{S}$). Therefore, now a normal letter always carries a subscript (e.g., $\mathrm{A}_i, \mathrm{S}_i$, $\mathrm{S}_j$) so they can be well distinguished.
2. The notation $\mathbf{U}$ is no longer used. Instead we will use $(\mathbf{S}, \mathbf{A})$ directly. This way we avoid introducing unfamiliar notations, and $\mathbf{U}$ will not be confused with the letter used in a field $C.U$ or an attribute $O.\mathrm{U}$.
3. We have changed the notation of local class-level causality to $C.U \rightarrow V$, and that of glocal class-level causality to $C_l.U \rightarrow C_k.V$. These notations are simpler and more intuitive.
4. We simplified equations (5-10) in the old version, and now there are only two equations (5-6) in the revised version to express the same meaning. This reduces the reader's workload and the complexity of the notation.

# Related works

We have mentioned these papers in Related Works (Section 2):
1. Mittal, et al. Learning to Combine Top-Down and Bottom-Up Signals in Recurrent Neural Networks with Attention over Modules, 2020.
2. Mittal, et al. Is a Modular Architecture Enough? 2022.
3. Locatello, et al. Object-Centric Learning with Slot Attention, 2022.
4. Maier, et al. Learning causal models of relational domains, 2010.
5. Marazopoulou, et al. Learning the structure of causal models with relational and temporal dependence, 2015.

# Dealing with asymmetric environments

Some reviewers argue that Causation Symmetry may be too strong for the method to be applied to realistic domains. Therefore, we have provided a simple solution to extend OOCDM to asymmetric environments in Appendix I:
1. A theory of how to ensure result and causation symmetries in OOMDPs.
2. A modification to OOCDM that allows it to handle asymmetric environments without explicitly changing the OOMDP representation.
3. A simple experiment on AsymBlock (the asymmetric version of the Block environment) showing the effectiveness of the solution.

In short, by adding auxiliary identity attributes to the objects, we can ensure causation symmetry and result symmetry in all OOMDPs. If the violation of the symmetries is not severe, OOCDM can still give good performance and discover the minimal OOCG for the environment. In the main paper, we have added a brief discussion of asymmetric environments in Section 4.

---

### Author Response · Authors · 2023-11-22
**Thanks to the reviewers and chairs!**

The discussion phase is successful, and we enjoyed the in-depth conversation with other excellent researchers in the related field. The reviewers' suggestions are very helpful. We will continue to improve the quality of this paper, whether it is accepted or not. Many thanks to all reviewers and chairs for their hard work!

---

### Meta-Review · Area_Chair_zcmA · 2023-12-04

**Metareview:**

The paper tackles the problem of scaling causal dynamics models to larger-scale object-oriented environments. The experiments are quite nice, especially those on the starcraftII environment, exhibiting a much larger visual complexity compared to prior work. The main argument for rejection is the assumption of causal symmetry, with the counterexample of 3 objects in a line connected by springs raised by reviewer LMvE. The issue is that the causal symmetry assumption seems to be the major driver for the performance improvements, and the solution to the counter-example proposed by the authors hinders the original advantages. I suggest the authors try improving on the assumption and find a solution to this issue.

**Justification For Why Not Higher Score:**

The paper has limitations, in particular around the symmetry assumption, which limits the otherwise substantial improvement in visual complexity for object-centric causal scene models. I encourage the authors to fix this issue and resubmit.

**Justification For Why Not Lower Score:**

N/A

---

### Decision · Program_Chairs · 2024-01-16

Reject